# Rectified Flows for Fast Multiscale Fluid Flow Modeling

**Victor Armegioiu**                                    *victor.armegioiu@math.ethz.ch*
*Department of Mathematics, ETH Zürich, Zürich, Switzerland*

**Yannick Ramic**                                        *yannick.ramic@math.ethz.ch*
*Department of Mathematics, ETH Zürich, Zürich, Switzerland*

**Siddhartha Mishra**                                    *siddhartha.mishra@math.ethz.ch*
*Department of Mathematics, ETH Zürich, Zürich, Switzerland*

**Reviewed on OpenReview:** *https://openreview.net/forum?id=2tMD6YXgkp*

## Abstract

We introduce *ReFlow*, a conditional rectified-flow surrogate for PDE forecasting. Given an initial state $u_i$, ReFlow transports Gaussian noise $\xi$ to a sample from the conditional final-state law $p(u_f \mid u_i)$ by integrating a learned deterministic ODE. Unlike diffusion surrogates, which require many stochastic denoising steps, the rectified transport is close to straight in sampling time: on multiscale 2D flow benchmarks, ReFlow matches diffusion-level posterior statistics with as few as 8 ODE steps, compared with $\geq 128$ network evaluations for score-based diffusion.

We also give a law-level analysis for conditional PDE surrogates. We formulate the ideal conditional rectified velocity as a barycentric transport field and show that it pushes the reference law to the target conditional law. At fixed spatial resolution, we decompose the one-step law error into a coverage term, controlled by unresolved high-frequency content via structure functions or spectral tails, and a fit term measuring approximation of the ideal velocity field. We further show that ODE discretization error is governed by the variation of the learned velocity along sampled rectified trajectories. This motivates a curvature-aware sampler that uses an EMA proxy for trajectory-wise velocity variation to stabilize inference, especially out of distribution.

Across incompressible and compressible 2D flows, ReFlow matches diffusion baselines in one-point Wasserstein statistics and energy spectra, preserves fine-scale structure missed by deterministic MSE models, and produces high-resolution conditional samples at substantially lower inference cost.

## 1 Introduction

Partial differential equations (PDEs) Evans (2010) are the backbone of modeling physical phenomena, from atmospheric dynamics to aerodynamics and magnetohydrodynamics. Currently, solutions of PDEs are simulated with high-fidelity numerical solvers such as finite difference, finite element, and spectral methods Quarteroni & Valli (1994). These are often prohibitively expensive for *many–query* tasks Quarteroni et al. (2015) such as uncertainty quantification, optimization, and real-time prediction. In particular, these solvers encounter a significant challenge for classes of non-linear PDEs, whose solutions exhibit extreme sensitivity to initial and boundary data, leading to chaotic behavior and complex multi-scale features. In such settings, one seeks, not a deterministic trajectory, but rather the *statistical solution* of the PDE, i.e., the push-forward of the underlying probability measure on inputs (initial and boundary conditions etc) by the PDE solution operator Fjordholm et al. (2017). Currently, these statistical solutions are sampled with ensemble Monte Carlo method by repeatedly calling high-fidelity PDE solvers Lanthaler et al. (2021a). Given the computational cost of the PDE solver, ensemble methods become infeasible as ensemble sizes grow.

Machine learning and AI based algorithms are increasingly being explored for the data-driven simulation of PDEs Mishra & Townsend (2024). In particular, *neural operators* Kovachki et al. (2023), which directly seek to learn the PDE solution operator from data, are already widely used in a PDE context Mishra & Townsend (2024). Examples of neural operators have been proposed in Li et al. (2020a;b); Lu et al. (2021); Wu et al. (2024); Hao et al. (2023); Li et al. (2023); Alkin et al. (2024) and references therein.

However, these neural operators have been found to be unsuitable for simulating PDE with chaotic, multiscale solutions such as governing equations of fluid dynamics. For instance, recently in Molinaro et al. (2024), it was shown both theoretically and empirically that neural operators, trained to minimize mean absolute (square) errors, will always collapse to the ensemble mean when used for simulating statistics of PDEs with chaotic multiscale solutions.

Given this context, generative AI models—especially score-based and diffusion models—have recently shown remarkable success at learning complex, high-dimensional distributions by progressive denoising score matching Song et al. (2020). In the context of chaotic PDEs, *GenCFD* Molinaro et al. (2024) introduced a conditional diffusion framework that preserves spectral content across fine scales, mitigating the "spectral collapse" of naive MSE-trained surrogates, see also Oommen et al. (2024); Gao et al. (2024a;b;c). However, sampling a score-based diffusion model requires simulating a reverse-time stochastic differential equation (SDE) over 100+ discretization steps, each invoking a large ML model on a moderate-to-high resolution spatial grid, resulting in high runtimes that undermine real-time or resource-constrained applications.

To overcome the inference cost of diffusion-based surrogates, we instead use a *conditional rectified flow* Liu et al. (2022). The key point is to separate the *conditioning variable* from the *transported variable*. In our PDE setting, the initial condition $u_i$ is *not* transported by the rectified flow; rather, it is provided as conditioning information throughout the model. The transported variable is a latent sample initialized from a simple reference law, typically Gaussian noise. Thus, for each fixed initial datum $u_i$, the goal is to learn a time-dependent velocity field whose flow maps a reference sample $\xi \sim \mathcal{N}(0, I)$ to a sample from the conditional output law $p(u_f \mid u_i)$, where $u_f$ denotes the PDE state at the target physical time.

This viewpoint is especially natural for chaotic or strongly multiscale PDEs. When the same macroscopic initial datum is subjected to small unresolved perturbations, the corresponding final state is no longer well represented by a single deterministic predictor; instead, one seeks a conditional *law* over possible outcomes. Conditional diffusion models such as GenCFD already target this law successfully, but do so through a reverse-time stochastic process that typically requires many score-network evaluations. Rectified flow replaces this stochastic sampling procedure by a deterministic ODE in an internal *rectification time* $\tau \in [0, 1]$, chosen so that the learned transport from noise to data is as straight as the conditional law permits. In practice this substantially reduces the number of function evaluations needed to reach the target conditional distribution.

A central theme of this paper is that *straightness in rectification time* matters for computational reasons. If the learned velocity varies little along its own trajectories, then explicit ODE solvers incur small truncation error and can take large steps. Conversely, when the learned velocity oscillates strongly along the sampled path—a phenomenon we observed most clearly in out-of-distribution conditional regimes—few-step sampling becomes fragile. This observation motivated our curvature-aware sampler: an EMA-based controller that tracks short-horizon velocity oscillations along the trajectory and uses them to blend and adapt the step size during inference.

**Statistical evaluation criteria.** Because statistical solutions are probability measures on function spaces, the most robust comparisons are formulated through observables and their induced correlation marginals Lanthaler et al. (2021b). Our main metric in Section 6.2, the *one-point Wasserstein distance*, is a natural $k = 1$ correlation-marginal discrepancy: it compares the pointwise pushforward laws and averages the resulting Wasserstein distances over space. Thus the experimental metrics in Table 1 are directly aligned with the law-level viewpoint used in the theory sections below.

Hence, our main contributions are:

- *Conditional rectified flows for PDE surrogates.* We formulate rectified flow directly at the level of conditional PDE laws: for each fixed input field $u_i$, the model transports a simple reference law on

latent fields to the target conditional law of final states. The key theoretical statement is *correctness at the level of laws*: the ideal conditional barycentric rectified velocity induces an ODE whose flow pushes the reference law to the target conditional law.

- *A law-level decomposition into coverage, fit, and discretization.* At fixed spatial resolution, there is an irreducible approximation floor coming from unresolved Fourier scales. We express this as a *coverage* term controlled by structure functions, equivalently by spectral tails. The remaining continuous-time modeling error is a *fit* term measuring how well the learned velocity approximates the ideal conditional transport field. Separately, we show that discretization error is governed by the variation of the learned velocity *along sampled trajectories* in rectification time.

- *Curvature-aware sampling for OOD-sensitive conditional generation.* Recent work on diffusion-based OOD and anomaly detection has shown that trajectory geometry, score variation, and curvature carry useful information about whether a sample remains in-distribution or moves into poorly calibrated regions of the learned generative dynamics Heng et al. (2024); Abdi et al. (2025a;b); Barkley et al. (2025). Motivated by this perspective, we treat trajectory-wise velocity variation in rectification time as an online signal that the conditional sampler is entering an unstable or OOD regime. We therefore introduce an EMA-based controller that monitors this straightness/curvature proxy and uses it to adaptively blend the velocity and choose the step size during inference on OOD data. Unlike prior works, which use such trajectory information primarily for detection or scoring, we use it *operationally* to stabilize and improve sampling itself.

- *Multi-scale accuracy at substantially lower inference cost.* Across incompressible and compressible 2D benchmarks, ReFlow matches diffusion models in one-point Wasserstein statistics and energy spectra, preserves fine-scale structure missed by deterministic MSE baselines, and achieves these results with dramatically fewer sampling steps. The gain is especially pronounced in the OOD macro–micro regime, where the curvature-aware controller improves the NFE–error tradeoff by reacting to the same trajectory-geometry signals that recent diffusion OOD work has identified as informative under distribution shift Heng et al. (2024); Abdi et al. (2025a); Barkley et al. (2025).

## 2 Approach

**Modeling multi-scale flows and their statistical solutions.** We focus on time-dependent PDEs of the general form

$$\partial_t u(x,t) \, + \, \mathcal{L}\big(u, \nabla_x u, \nabla_x^2 u, \dots\big) \, = \, 0, \quad x \in D \subset \mathbb{R}^d, \ t \in (0,T), \tag{1}$$

subject to boundary conditions $\mathcal{B}(u) = u_b$ on $\partial D \times (0,T)$ and initial data $u(x,0) = \bar{u}(x)$. Here, the solution $u : D \mapsto \mathbb{R}^m$ is evolved in time, with respect to the differential operator $\mathcal{L}$. Concrete examples of (1) are the incompressible Navier-Stokes and compressible Euler equations.

A recurring theme in non-linear, multi-scale fluid flows is *extreme sensitivity* to initial or boundary data, which can lead to chaotic or near-chaotic behavior. Rather than fix a single $u_0$, one may consider an *initial measure* $\mu_0$ to represent uncertain or variable initial states, and then study the *statistical solution* $\mu_t = \mathcal{S}_\#^t \mu_0$ pushed forward by the PDE solution operator $\mathcal{S}^t$. In strongly nonlinear regimes, $\mu_t$ spreads significantly in function space, creating complex, multi-scale distributions of possible flow fields. Accurately approximating this evolving distribution is essential for uncertainty quantification and design under rough conditions, but doing so with high-fidelity PDE solvers can be prohibitively expensive when large ensembles are required.

**High cost of computing statistical solutions.** In practice, characterizing $\mu_t$ directly often requires performing *many* forward simulations of the PDE solver, each initialized with a slightly perturbed $u_0$. As each simulation might require high-resolution discretizations (e.g. millions of degrees of freedom) and multiple time steps, the overall cost for constructing a sizable ensemble rapidly becomes prohibitive. Even with modern high-performance computing platforms, running hundreds or thousands of high-fidelity statistical simulations can be infeasible. Hence, there is a pressing need for more efficient *surrogate models* that can *sample* physically realistic final states without enumerating every realization of the forward PDE solve. Succinctly put, we aim to provide a solution to the following problem

*How can we efficiently capture statistical quantities of interest for the pushforward measure $S_\#^t \mu_0$ for highly sensitive solution operators $S^t$?*

**Observables and correlation marginals (what metrics actually probe).** A classical way to formalize "quantities of interest" for $\mu_t$ is via observables and their induced correlation marginals Lanthaler et al. (2021b). For $k = 1$, the pointwise evaluation map at $x \in D$,

$$\mathrm{ev}_x : \ u \mapsto u(x) \in \mathbb{R}^m,$$

pushes $\mu_t$ to the *one-point marginal* $(\mathrm{ev}_x)_\# \mu_t$ on $\mathbb{R}^m$ – strictly speaking, one studies the pushforwards measures $\mathrm{ev}_x \circ i_\epsilon$, here $(i_\epsilon u) := (\phi_\epsilon * u)$ where $\phi_\epsilon$ is a compactly supported mollifier, so that point-wise evaluation is well-defined; for the sake of brevity, we will not make this distinction explicit in the sequel. Our one-point Wasserstein metric (Section 6.2) is the spatial average of $W_1\big((\mathrm{ev}_x)_\# \mu_t, (\mathrm{ev}_x)_\# \widehat{\mu}_t\big)$. By Kantorovich–Rubinstein duality, controlling this quantity controls the discrepancy of *all* pointwise 1-Lipschitz observables, averaged over space. This is the bridge between Table 1 and the law-level bounds in Section 4.1 and Section 4.3.

**Rectified flows for faster sampling.** To mitigate these inefficiencies, we adopt the framework of *rectified flows* Liu et al. (2022): a deterministic alternative to conditional diffusion designed to reduce the number of required integration steps. In essence, rather than generating samples via small random "diffusion-reversal" increments, rectified flows construct an *ordinary differential equation* (ODE) with *nearly straight* trajectories in function space. This straighter transport map can be integrated more aggressively in time, enabling sampling with far fewer solver steps. Importantly, rectified flows retain the desirable property of matching the same final distributions that standard diffusion models learn, but at a fraction of the runtime cost.

## 3 Related Work

### 3.1 Diffusion Models in Fluid Dynamics

**Conditional diffusion surrogates.** *GenCFD* Molinaro et al. (2024) trains a conditional score-based diffusion model that starts from an initial or low-resolution flow field and iteratively refines it through hundreds of denoising steps, thereby capturing multi-scale turbulent structure. Similarly, the method of Oommen et al. (2024) conditions a diffusion model on a coarse fluid solution (generated by a learned operator) and "super-resolves" the fine-scale flow. Given these similarities, we refer to such diffusion-based solvers collectively as *conditional diffusion PDE solvers*.

**Rectified flow alternative.** Each sample typically needs hundreds of network calls, limiting large-scale or real-time use. Our method replaces the stochastic SDE with a deterministic ODE whose nearly straight trajectories require far fewer steps, achieving similar statistical fidelity at much lower inference cost (see Sections 5.7 and 6). Crucially, we find that rectified flows are actually more efficient from the standpoint of training-time required to reach the same accuracy level with the conditional diffusion approaches; this is also confirmed by earlier work Esser et al. (2024) exploring the scalability of rectified flows with transformer-based backbones.

## 4 Rectified Flows: An ODE-Based Alternative

### 4.1 Mathematical Formulation

**Overview.** Rectified flows provide a *deterministic* mechanism for sampling from a target probability law by integrating an ordinary differential equation in an internal *rectification time* $\tau \in [0, 1]$. In our PDE setting, the target is *conditional*: given an initial condition (and, when relevant, boundary data) $u_i$, we wish to sample from the conditional law of the final state $u_f$ at the forecast time. The key point is that the conditioning datum $u_i$ is *not* itself transported by the rectified flow. Rather, it is provided to the model as side information throughout the trajectory. The transported variable starts from a simple reference law, typically Gaussian noise, and is evolved to a sample from the target conditional law.

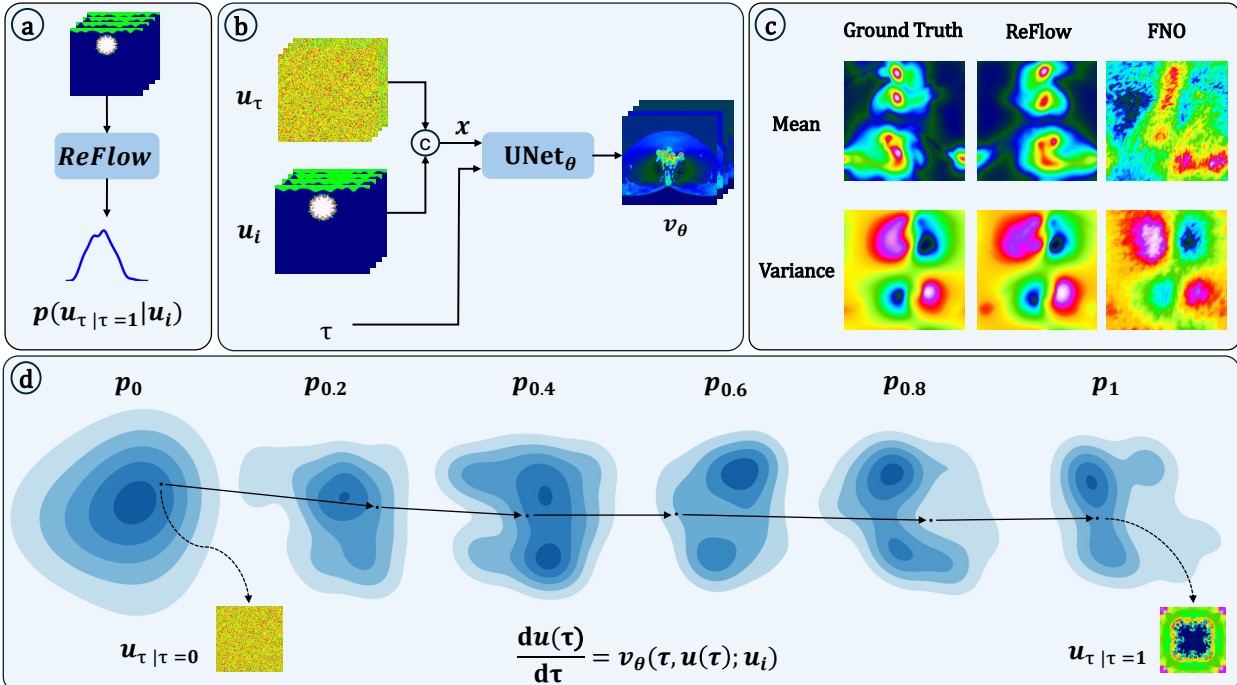

Figure 1: **Overview of ReFlow.** (**a**): ReFlow closely matches high-order statistical moments (in the Wasserstein $W_1$-sense) of fluid flows by learning a neural surrogate to approximate the push-forward of the distribution $p(u_{\tau|\tau=0}|u_i)$, while minimizing inference-time computational costs. (**b**) Macro-architecture overview: A UViT based network predicts the velocity field $v_\theta$, conditioned on Gaussian noise, initial data $u_i$ and boundary conditions, and diffusion time $\tau$. Noise and initial / boundary conditions are concatenated as inputs. (**c**): Results for the cylindrical shear flow dataset. (**d**): The KDE plots monitor the evolution of a down-projected version of the push-forward measure $p_\tau := X_\theta(\tau)_\# p(u_{\tau=0}|u_i)$, where $X_\theta$ is the flow induced by the learned vector field $v_\theta$. Results for the Richtmyer Meshkov dataset showing flow matching between noise ($p_{\tau=0}$) and target ($p_{\tau=1}$) distributions. ReFlow solves an ODE, learning a velocity field whose sampled trajectories are substantially straighter than the corresponding diffusion trajectories (visualized with principal component analysis and kernel density estimation at different diffusion times).

Throughout, the random state fields take values in a separable Hilbert space

$$\mathcal{U} = L^2(\mathcal{D}; \mathbb{R}^m),$$

or, when convenient, in a smoother space such as $H^s(\mathcal{D}; \mathbb{R}^m)$ with $s > d/2$. In computations, all objects are represented on a finite-dimensional Galerkin or grid discretization $\mathbb{R}^{d_N} \subset \mathcal{U}$. The exact flow-map statements below are meant rigorously in this finite-dimensional setting. The Hilbert-space notation records the natural law-level formulation; in infinite dimensions the same identities should be read at the continuity-equation level unless additional well-posedness assumptions are imposed.

**Conditional target law and reference law.** Let $C$ denote the conditioning variable, which in our applications is the initial condition $u_i$ (possibly augmented with boundary data or other known inputs). For each fixed realization $c$ of $C$, let

$$\mu_1^c := \mathrm{Law}(U_f \mid C = c)$$

be the conditional law of the final PDE state, and let

$$\pi_0^c$$

be a simple reference law on $\mathcal{U}$ from which sampling is easy. In practice we take $\pi_0^c = \mathcal{N}(0, I)$, independent of $c$, but allowing a $c$-dependence is conceptually harmless and can be useful in variants with condition-dependent priors.

Thus, for each fixed condition $c$, the goal of rectified flow is to construct a time-dependent velocity field

$$v_\theta(\,\cdot\,, \tau; c) : \mathcal{U} \to \mathcal{U}$$

such that the ODE flow

$$\frac{d}{d\tau} u_\tau = v_\theta(u_\tau, \tau; c), \qquad u_{\tau=0} \sim \pi_0^c, \tag{2}$$

pushes $\pi_0^c$ forward to a law close to $\mu_1^c$ at $\tau = 1$.

**Conditional coupling and rectified interpolation.** Fix a condition $c$. Let

$$\gamma^c \in \mathcal{P}(\mathcal{U} \times \mathcal{U})$$

be any coupling between the reference law $\pi_0^c$ and the target law $\mu_1^c$. We write

$$(\Xi, U_f) \sim \gamma^c,$$

where $\Xi$ is the reference sample and $U_f$ is the target final state. The most basic rectified interpolation is the linear chord

$$U_\tau := (1 - \tau)\,\Xi + \tau\,U_f, \qquad \tau \in [0, 1]. \tag{3}$$

Its conditional law at rectification time $\tau$ is

$$\rho_\tau^c := \mathrm{Law}(U_\tau \mid C = c).$$

This is the population of intermediate states obtained by linearly interpolating between the noise sample and the target sample, under the chosen conditional coupling. The practical training objective used in this paper samples $\xi \sim \pi_0^c$ independently of $u_f \sim \mu_1^c$, and therefore corresponds to the product coupling $\gamma^c = \pi_0^c \otimes \mu_1^c$. We state the construction for a general coupling because the rectified-flow transport identity has this form for any prescribed coupling, but the resulting barycentric velocity depends on that choice.

**The ideal conditional barycentric velocity.** At a fixed condition $c$, many interpolation chords may pass through the same state $u$ at the same rectification time $\tau$. The correct population-level velocity is therefore the average chord direction conditioned on the current state:

$$v^\star(u, \tau; c) := \mathbb{E}[U_f - \Xi \mid U_\tau = u,\ C = c], \qquad \rho_\tau^c\text{-a.e. } u. \tag{4}$$

This is the *conditional barycentric rectified velocity*. It is the natural analogue, in the conditional setting, of the barycentric velocity used in standard rectified-flow and flow-matching theory Liu et al. (2022). For the linear interpolation (3), the chord direction $U_f - \Xi$ is constant along each sample path, so (4) says that the ideal velocity field is obtained by averaging these constant sample-wise directions over all chords compatible with the observed intermediate state.

By the standard $L^2$-projection property of conditional expectations, the field (4) is also the unique $L^2(\rho_\tau^c)$ minimizer of the conditional regression problem

$$\min_{v(\cdot, \tau; c)} \ \mathbb{E}\left[ \left\| v(U_\tau, \tau; c) - (U_f - \Xi) \right\|^2 \,\middle|\, C = c \right].$$

This makes the practical training objective immediate: if one samples

$$c, \quad u_f \sim \mu_1^c, \quad \xi \sim \pi_0^c, \quad \tau \sim \mathrm{Unif}[0, 1],$$

forms

$$u_\tau = (1 - \tau)\xi + \tau u_f, \tag{5}$$

and trains $v_\theta$ against the constant target $(u_f - \xi)$, then the population minimizer is exactly the ideal conditional barycentric field (4). In particular, the transported quantity is the latent variable initialized at $\xi$, while the initial PDE state $c = u_i$ appears only as a condition in the network. With the notation above, the training loss is

$$\min_\theta \ \mathbb{E}\Big[\big\|v_\theta(u_\tau, \tau; c) - (u_f - \xi)\big\|^2\Big], \qquad u_\tau = (1 - \tau)\xi + \tau u_f, \tag{6}$$

where the expectation is taken over the joint sampling of $(c, u_f)$ from the dataset, $\xi \sim \pi_0^c$, and $\tau \sim \mathrm{Unif}[0, 1]$. This is the conditional rectified-flow objective used throughout the paper.

**Sampling from the learned conditional law.** Once trained, the model generates a conditional sample at a prescribed initial datum $c = u_i$ by drawing

$$\xi \sim \pi_0^{u_i}$$

and integrating the conditional transport ODE

$$\frac{d}{d\tau} u_\tau = v_\theta(u_\tau, \tau; u_i), \qquad u_0 = \xi. \tag{7}$$

The final state $u_1$ is then the model sample from the learned approximation to the conditional law $\mu_1^{u_i}$.

**Rectification time versus physical time.** The variable $\tau \in [0, 1]$ is an *internal* transport parameter and should not be confused with the physical PDE time. The physical dynamics have already been absorbed into the conditional target law $p(u_f \mid u_i)$ through the training data. Rectification time only parameterizes the learned transport from the reference law to that target conditional law. The benefit of rectified flow is not that the learned velocity is literally constant, but that for a well-trained model it can vary substantially less along sampled trajectories than the score field underlying diffusion-based samplers. This matters because the numerical error of explicit ODE solvers depends on how rapidly

$$\tau \mapsto v_\theta(u_\tau, \tau; u_i)$$

changes along the realized trajectory. When this variation is small, few large steps suffice; when it becomes oscillatory, sampling becomes more delicate. In our experiments, such oscillations are especially pronounced in out-of-distribution conditional regimes, which motivates the curvature-aware controller described next.

**Curvature-aware control: monitoring velocity variation along the trajectory.** During sampling, at step $t$ with state $u_t$ and rectification time $\tau_t$, write

$$v_t := v_\theta(u_t, \tau_t; u_i),$$

and maintain an exponential moving average

$$v_t^{\mathrm{ema}} = \lambda\, v_{t-1}^{\mathrm{ema}} + (1 - \lambda)\, v_t, \qquad \lambda \in (0, 1). \tag{8}$$

We use the deviation

$$s_t := \big\|v_t - v_t^{\mathrm{ema}}\big\|_2 \tag{9}$$

as a computable proxy for short-horizon variation of the velocity along the sampled path. Large values of $s_t$ indicate that the trajectory is entering a regime where the learned transport is bending or oscillating in rectification time, precisely where aggressive explicit stepping becomes less reliable.

**Blending and adaptive step size.** When the straightness proxy is large, we replace $v_t$ by a regularized velocity

$$\tilde{v}_t = (1 - \alpha_t)v_t + \alpha_t v_t^{\mathrm{ema}}, \qquad \alpha_t \in [0, 1],$$

where $\alpha_t$ is chosen as an increasing function of $s_t$. At the same time, we choose the step size inversely proportional to the estimated local variation, for example

$$\Delta\tau_t \propto \frac{1}{\sqrt{\kappa_1 s_t + \kappa_2}}, \qquad \Delta\tau_t \in [\Delta\tau_{\mathrm{min}}, \Delta\tau_{\mathrm{max}}]. \tag{10}$$

The resulting update is

$$u_{t+1} = u_t + \Delta\tau_t \, \tilde{v}_t.$$

This controller is dormant when the transport is straight and active only when the velocity becomes oscillatory. In practice, the clipping interval $[\Delta\tau_{\min}, \Delta\tau_{\max}]$ and the controller constants are selected once on a held-out macro–micro validation sweep by minimizing the cost–accuracy criterion Rel-$L^2 \times$ NFE, and are then fixed for the reported test evaluations. The robustness sweep in Section 7.3 shows that performance is not tied to a narrow tuning region. In Section 5 we connect this construction to the truncation error of explicit ODE integration and explain why such a controller is particularly helpful in the OOD regimes where velocity oscillations become more pronounced.

## 4.2 Error Decomposition and Bounds

There are two distinct sources of error once the spatial resolution is fixed. The first is a *continuous-time modeling error*: even if we could integrate the learned ODE exactly, the learned velocity field $v_\theta$ may differ from the ideal conditional barycentric field $v^\star$. The second is a *numerical discretization error*: even if $v_\theta$ were fixed, solving the learned ODE with finitely many steps introduces truncation error. These two effects should not be conflated. In particular, if $v_\theta = v^\star$, then the continuous-time modeling error must vanish; the remaining error is purely numerical.

**Ideal, learned, and discretized conditional flows.** Fix a condition $c$. Let $\Xi \sim \pi_0^c$, and let

$$\dot{U}_\tau = v^\star(U_\tau, \tau; c), \qquad U_0 = \Xi, \tag{11}$$

denote the *ideal* conditional rectified flow, and let

$$\dot{\widehat{U}}_\tau = v_\theta(\widehat{U}_\tau, \tau; c), \qquad \widehat{U}_0 = \Xi, \tag{12}$$

denote the *learned continuous-time* conditional flow driven by the neural velocity field. Finally, let $\widehat{U}_1^{(N)}$ denote the output obtained by numerically integrating (12) with $N$ explicit steps. Writing

$$\widehat{\mu}_1^{c,\mathrm{cts}} := \mathrm{Law}(\widehat{U}_1 \mid C = c), \qquad \widehat{\mu}_1^{c,N} := \mathrm{Law}(\widehat{U}_1^{(N)} \mid C = c),$$

the conditional law error satisfies the elementary decomposition

$$W_2\big(\mu_1^c, \widehat{\mu}_1^{c,N}\big) \ \leq \ W_2\big(\mu_1^c, \widehat{\mu}_1^{c,\mathrm{cts}}\big) \ + \ W_2\big(\widehat{\mu}_1^{c,\mathrm{cts}}, \widehat{\mu}_1^{c,N}\big). \tag{13}$$

The first term is the *fit* error of the learned velocity field; the second is the *discretization* error of the sampler. The exact-transport statement proved later in Proposition 5.1 identifies $\mu_1^c$ as the terminal law of the ideal flow (11).

**Theorem 4.1** (Continuous-time fit error for the learned conditional flow). *Fix a condition $c$ and assume that $v_\theta(\cdot, \tau; c)$ is $L$-Lipschitz in $u$, uniformly in $\tau \in [0,1]$. Let $U_\tau$ and $\widehat{U}_\tau$ solve (11) and (12) with the same initial sample $\Xi \sim \pi_0^c$. Then*

$$\Big(\mathbb{E}\big[\|\widehat{U}_1 - U_1\|^2 \,\big|\, C = c\big]\Big)^{1/2} \ \leq \ e^L \int_0^1 \Big(\mathbb{E}\big[\|v_\theta(U_\tau, \tau; c) - v^\star(U_\tau, \tau; c)\|^2 \,\big|\, C = c\big]\Big)^{1/2} d\tau. \tag{14}$$

*In particular, defining*

$$\varepsilon_{\mathrm{fit}}^2(c) := \int_0^1 \mathbb{E}\big[\|v_\theta(U_\tau, \tau; c) - v^\star(U_\tau, \tau; c)\|^2 \,\big|\, C = c\big] \, d\tau, \tag{15}$$

*we obtain*

$$\Big(\mathbb{E}\big[\|\widehat{U}_1 - U_1\|^2 \,\big|\, C = c\big]\Big)^{1/2} \ \leq \ e^L \, \varepsilon_{\mathrm{fit}}(c). \tag{16}$$

Consequently, by coupling the ideal and learned flows with the same initial sample,

$$W_2\big(\mu_1^c, \widehat{\mu}_1^{c,\mathrm{cts}}\big) \le e^L \varepsilon_{\mathrm{fit}}(c),$$

where $\widehat{\mu}_1^{c,\mathrm{cts}} = \mathrm{Law}(\widehat{U}_1 \mid C = c)$.

*Proof sketch.* Set $E_\tau := \widehat{U}_\tau - U_\tau$. Then

$$\dot{E}_\tau = v_\theta(\widehat{U}_\tau, \tau; c) - v^\star(U_\tau, \tau; c) = \big(v_\theta(\widehat{U}_\tau, \tau; c) - v_\theta(U_\tau, \tau; c)\big) + \big(v_\theta(U_\tau, \tau; c) - v^\star(U_\tau, \tau; c)\big).$$

The Lipschitz assumption yields

$$\|\dot{E}_\tau\| \le L\|E_\tau\| + \|v_\theta(U_\tau, \tau; c) - v^\star(U_\tau, \tau; c)\|.$$

Applying the standard Grönwall stability argument for ODEs pathwise, then taking conditional $L^2$ norms and using Minkowski's inequality in time, gives (14). The compact form (16) follows from Cauchy–Schwarz in $\tau$. $\qquad\square$

**Lemma 4.2** (Local truncation error for explicit Euler on the learned conditional ODE). *Assume that $v_\theta$ is $C^2$ in $(u, \tau)$ on the region traversed by the learned trajectory $\widehat{U}_\tau$, and define*

$$g(\tau) := v_\theta(\widehat{U}_\tau, \tau; c).$$

*Then one explicit-Euler step of size $\Delta\tau$ from time $\tau$ along the learned ODE (12) satisfies*

$$\mathrm{LTE}(\tau; \Delta\tau) := \left\| \widehat{U}_{\tau+\Delta\tau} - \big(\widehat{U}_\tau + \Delta\tau\, v_\theta(\widehat{U}_\tau, \tau; c)\big) \right\|$$

$$\le \frac{(\Delta\tau)^2}{2} \left\| \partial_\tau v_\theta(\widehat{U}_\tau, \tau; c) + J_u v_\theta(\widehat{U}_\tau, \tau; c)\, v_\theta(\widehat{U}_\tau, \tau; c) \right\| + C\,(\Delta\tau)^3. \qquad (17)$$

*If, in addition,*

$$\|\partial_\tau v_\theta(u, \tau; c)\| \le M_\tau, \qquad \|J_u v_\theta(u, \tau; c)\| \le L, \qquad \|v_\theta(u, \tau; c)\| \le M$$

*on the relevant region, then*

$$\mathrm{LTE}(\tau; \Delta\tau) \le \frac{(\Delta\tau)^2}{2}\,(M_\tau + LM) + \mathcal{O}\big((\Delta\tau)^3\big). \qquad (18)$$

*Proof sketch.* Taylor-expand the exact trajectory $\widehat{U}_{\tau+\Delta\tau}$ around $\tau$. Since $\dot{\widehat{U}}_\tau = v_\theta(\widehat{U}_\tau, \tau; c)$, the second derivative is

$$\ddot{\widehat{U}}_\tau = \frac{d}{d\tau} v_\theta(\widehat{U}_\tau, \tau; c) = \partial_\tau v_\theta(\widehat{U}_\tau, \tau; c) + J_u v_\theta(\widehat{U}_\tau, \tau; c)\, \dot{\widehat{U}}_\tau,$$

and substituting $\dot{\widehat{U}}_\tau = v_\theta(\widehat{U}_\tau, \tau; c)$ gives (17). The bound (18) is immediate. $\qquad\square$

**Corollary 4.3** (Global Euler discretization error). *Assume the hypotheses of Lemma 4.2, and assume again that $v_\theta(\cdot, \tau; c)$ is $L$-Lipschitz in $u$ uniformly in $\tau$. Let $\widehat{U}^{(N)}$ denote the explicit-Euler approximation of (12) with uniform step $\Delta\tau = 1/N$. Then there exists a constant $C(L) > 0$ such that*

$$\sup_{0 \le n \le N} \Big( \mathbb{E}\big[\|\widehat{U}_{\tau_n}^{(N)} - \widehat{U}_{\tau_n}\|^2 \,\big|\, C = c\big] \Big)^{1/2} \le C(L)\,\Delta\tau \int_0^1 \Big( \mathbb{E}\big[\|\dot{g}(\tau)\|^2 \,\big|\, C = c\big] \Big)^{1/2} d\tau, \qquad (19)$$

*where*

$$\dot{g}(\tau) = \partial_\tau v_\theta(\widehat{U}_\tau, \tau; c) + J_u v_\theta(\widehat{U}_\tau, \tau; c)\, v_\theta(\widehat{U}_\tau, \tau; c).$$

*Equivalently, defining*

$$\varepsilon_{\mathrm{disc}}(c; N) := \frac{1}{N} \int_0^1 \Big( \mathbb{E}\big[\|\dot{g}(\tau)\|^2 \,\big|\, C = c\big] \Big)^{1/2} d\tau, \qquad (20)$$

*we have*

$$\sup_{0 \le n \le N} \Big( \mathbb{E}\big[\|\widehat{U}_{\tau_n}^{(N)} - \widehat{U}_{\tau_n}\|^2 \,\big|\, C = c\big] \Big)^{1/2} \le C(L)\,\varepsilon_{\mathrm{disc}}(c; N). \qquad (21)$$

*Proof sketch.* Using the integral Taylor remainder along the learned trajectory, one has

$$\mathrm{LTE}(\tau_n; \Delta\tau) \leq \Delta\tau \int_{\tau_n}^{\tau_{n+1}} \|\dot{g}(s)\| \, ds.$$

Propagating these one-step errors through the $L$-Lipschitz flow stability and applying discrete Grönwall gives (19). $\qquad\square$

Corollary 4.3 shows that the quantity governing step-count requirements is the *total derivative of the learned velocity along the sampled trajectory*,

$$\frac{d}{d\tau} v_\theta(\widehat{U}_\tau, \tau; c) = \partial_\tau v_\theta(\widehat{U}_\tau, \tau; c) + J_u v_\theta(\widehat{U}_\tau, \tau; c) \, v_\theta(\widehat{U}_\tau, \tau; c).$$

This is the precise mathematical meaning of "curvature" or "straightness" in rectification time for our sampler. It is not merely the partial derivative $\partial_\tau v_\theta$ at fixed state, but the full variation of the velocity seen along the realized transport path.

**Connection to the EMA controller.** The practical proxy

$$s_t = \|v_t - v_t^{\mathrm{ema}}\|_2, \qquad v_t = v_\theta(u_t, \tau_t; c),$$

used in Section 4.1, is intended to track this trajectory-wise variation over a short time window. Large values of $s_t$ indicate that the map $\tau \mapsto v_\theta(u_\tau, \tau; c)$ is oscillating, precisely the regime where (17) predicts larger truncation error for aggressive explicit stepping. Empirically, we observed these oscillations to be much more pronounced in out-of-distribution conditional regimes, which is why the EMA-based blending and adaptive step size are most useful there.

The discussion above isolates *fit* and *discretization* at a fixed spatial resolution. The remaining error source is *coverage*: even a perfect bandlimited surrogate cannot represent unresolved high-frequency content. We quantify that irreducible law-level floor in the next subsection through structure functions and spectral tails.

## 4.3 Spectral Capacity, Structure Functions, and Law-Level Coverage

**Functional setting.** We work on the periodic box $\mathcal{D} = \mathbb{T}^d$ and consider $L^2$ fields $\mathcal{H} := L^2(\mathcal{D}; \mathbb{R}^m)$ with norm $\|\cdot\|_2$. For incompressible velocity fields (e.g. SL2D, see Section 6), one may equivalently restrict to divergence-free subspaces and adopt the statistical-solution framework of Lanthaler et al. (2021b), which characterizes laws via correlation identities, admissible observables, and an energy inequality. In that framework the natural distance on laws is the $L^2$-based Wasserstein metric $W_2$, and small-scale regularity is expressed in terms of *structure functions* and their time-averages. The numerical ensembles used in our experiments fit this setting: they are generated by dissipative solvers on $\mathbb{T}^2$, have uniformly bounded energy, and exhibit empirical structure-function scaling (equivalently, power-law spectra) consistent with the assumptions below.

### 4.3.1 Bandlimited capacity and strain constraints

Any $u \in \mathcal{H}$ admits a Fourier series $u(x) = \sum_{k \in \mathbb{Z}^d} \hat{u}(k) e^{ik\cdot x}$. We write $P_{\leq K}$ for the projector onto modes $\{k : |k| \leq K\}$ and $P_{>K} = I - P_{\leq K}$. On a grid of size $N^d$, the Nyquist frequency satisfies $K_{\max} \simeq N$, so every numerically represented field is effectively bandlimited to $|k| \lesssim K_{\max}$. Architecturally, downsampling and anti-aliasing may enforce an even stricter cutoff $K_c \leq K_{\max}$.

The next lemma is a standard Bernstein-type inequality: bandlimited fields have a uniform Lipschitz cap.

**Lemma 4.4** (Bernstein upper bound for bandlimited fields)**.** *Let $u : \mathbb{T}^d \to \mathbb{R}^m$ satisfy $u = P_{\leq K} u$ for some integer $K \geq 1$. Then there exists a constant $C_d > 0$ depending only on the dimension $d$ such that*

$$\|\nabla_x u\|_{L^\infty(\mathbb{T}^d)} \leq C_d K^{1+\frac{d}{2}} \|u\|_{L^2(\mathbb{T}^d)}. \tag{22}$$

*In particular, on an energy shell $\|u\|_{L^2} \leq E^{1/2}$ one has*

$$\|\nabla_x u\|_{L^\infty} \leq C_d K^{1+\frac{d}{2}} E^{1/2}. \tag{23}$$

*Proof sketch.* Write $u(x) = \sum_{|k| \leq K} \hat{u}(k) e^{ik \cdot x}$, so $\nabla_x u(x) = \sum_{|k| \leq K} (ik) \hat{u}(k) e^{ik \cdot x}$ and $|\nabla_x u(x)| \leq \sum_{|k| \leq K} |k| |\hat{u}(k)|$. Cauchy–Schwarz and a lattice-counting bound $\#\{|k| \leq K\} \lesssim K^d$ give $\|\nabla u\|_{L^\infty} \lesssim (\sum_{|k| \leq K} |k|^2)^{1/2} \|\hat{u}\|_{\ell^2} \lesssim K^{1+d/2} \|u\|_{L^2}$. Full details are deferred to the theory appendix. $\square$

Upper bounds alone do not say what happens if a surrogate *misses* an entire high-frequency annulus. The next lemma gives a complementary lower bound.

**Lemma 4.5** (Annulus lower bound)**.** *Let $f = P_{[K,2K]}u$ for some $u \in L^2(\mathbb{T}^d)$, where $P_{[K,2K]} := P_{\leq 2K} - P_{\leq K}$ and $K \geq 1$. Then there exists $c > 0$ such that*

$$\|\nabla_x f\|_{L^2} \geq c K \|f\|_{L^2}. \tag{24}$$

*Proof sketch.* By Plancherel and the fact that the Fourier support of $f$ is contained in an annulus with $|k| \simeq K$,

$$\|\nabla f\|_{L^2}^2 = \sum_{K \leq |k| \leq 2K} |k|^2 |\hat{f}(k)|^2 \geq cK^2 \sum_{K \leq |k| \leq 2K} |\hat{f}(k)|^2 = cK^2 \|f\|_{L^2}^2.$$

$\square$

**Corollary 4.6** (Capacity requirement for typical strain levels)**.** *Suppose the target flows at time $t$ have typical energy $E$ and typical rate-of-strain $\|\nabla_x u\|_{L^\infty} \simeq L_\star$. Let a surrogate produce only bandlimited outputs $u = P_{\leq K_c} u$ with $\|u\|_{L^2} \leq E^{1/2}$. Then necessarily*

$$C_d K_c^{1+\frac{d}{2}} E^{1/2} \gtrsim L_\star, \tag{25}$$

*and hence a minimal cutoff*

$$K_c \gtrsim \left( \frac{L_\star}{E^{1/2}} \right)^{\frac{2}{2+d}}. \tag{26}$$

*Moreover, if the true flow has nontrivial energy in an annulus above $K_c$, then any surrogate restricted to modes $|k| \leq K_c$ necessarily misses a high-frequency gradient contribution of size at least $cK\|P_{[K,2K]}u\|_{L^2}$ for some $K > K_c$.*

*Remark* 4.7*.* Corollary 4.6 formalises the intuition that spatial capacity ($K_c$) is a hard constraint: at fixed energy scale $E$, there is a minimum cutoff needed to represent typical strain levels $L_\star$. In our experiments both GenCFD and ReFlow operate at the same grid resolution, so they share the same effective $K_c$; rectification does not invent new small scales but makes the trajectories inside this bandlimited class straight and easy to integrate.

### 4.3.2 Structure functions and small-scale coverage

We now quantify how much of the *law* of the PDE solution can be captured at a given cutoff $K_c$. For $u : \mathbb{T}^d \to \mathbb{R}^m$ and $r > 0$ define the second-order structure function

$$S_r^2(u) := \frac{1}{|B_r|} \int_{|h| \leq r} \|u(\cdot + h) - u(\cdot)\|_{L^2(\mathbb{T}^d)}^2 \, dh,$$

and for a probability law $\mu$ on $\mathcal{H}$ set

$$S_r^2(\mu) := \int S_r^2(u) \, d\mu(u).$$

In homogeneous turbulence $S_r^2(\mu)$ is directly related to the energy spectrum; power-law behaviour $S_r^2(\mu) \sim r^{\zeta_2}$ at small $r$ encodes the scaling of the cascade. In our experiments, Figure 3 visualizes this connection directly: matching the high-wavenumber tail in the spectrum is equivalent to capturing the small-$r$ behaviour of structure functions, and therefore to controlling the coverage term below.

Guided by the statistical-solution framework of Fjordholm et al. (2017), we assume there exists a modulus of continuity $\omega : (0, 1] \to [0, \infty)$ with $\omega(r) \downarrow 0$ as $r \downarrow 0$ such that

$$S_r^2(\mu_t) \leq \omega(r) \quad \text{for all } r \in (0, 1], \ t \in [0, T]. \tag{27}$$

In practice, $\omega$ can be taken as a piecewise power law $Cr^{2\alpha}$ calibrated from the empirical spectra of the numerical ensembles. The next lemma shows that (27) controls the high-frequency tail of the law.

**Lemma 4.8** (Structure function controls Fourier tail)**.** *Let $\mu$ be a law on $L^2(\mathbb{T}^d)$ satisfying $S_r^2(\mu) \leq \omega(r)$ for all $r \in (0,1]$, where $\omega(r) \downarrow 0$ as $r \downarrow 0$. Then there exist constants $c_d, C_d > 0$ such that for all integers $K \geq 1$,*

$$\int \|P_{>K}u\|_{L^2}^2 \, d\mu(u) \; \leq \; C_d \, \omega\big(c_d/K\big). \tag{28}$$

*Proof sketch.* Expand $u$ in Fourier series and observe $\|u(\cdot + h) - u(\cdot)\|_{L^2}^2 = \sum_k |\hat{u}(k)|^2 |e^{ik\cdot h} - 1|^2$. Averaging over $h \in B_r$ yields $S_r^2(u) = \sum_k |\hat{u}(k)|^2 \Psi_r(k)$, where $\Psi_r(k) \gtrsim \min\{(|k|r)^2, 1\}$ uniformly in $k, r$. For $|k| \gtrsim 1/r$, one has $\Psi_r(k) \gtrsim 1$, hence $S_r^2(u) \gtrsim \|P_{>c/r}u\|_{L^2}^2$. Integrate over $u \sim \mu$, apply (27), and set $r \sim 1/K$. Full constants and the Littlewood–Paley formulation are deferred to the theory appendix. $\square$

**Lemma 4.9** (Bandlimited projection in $W_2$)**.** *For any law $\mu$ on $L^2(\mathbb{T}^d)$ and cutoff $K_c \geq 1$,*

$$W_2\big(\mu, \, (P_{\leq K_c})_{\#}\mu\big) \; \leq \; \left(\int \|P_{>K_c}u\|_{L^2}^2 \, d\mu(u)\right)^{1/2}. \tag{29}$$

*Proof.* Couple $u \sim \mu$ and $\tilde{u} = P_{\leq K_c}u$. Then $\tilde{u} \sim (P_{\leq K_c})_{\#}\mu$ and $W_2^2(\mu, (P_{\leq K_c})_{\#}\mu) \leq \mathbb{E}\|u - \tilde{u}\|_{L^2}^2 = \mathbb{E}\|P_{>K_c}u\|_{L^2}^2$. $\square$

Combining Lemmas 4.8 and 4.9 gives a clean coverage bound.

**Corollary 4.10** (Coverage term from the structure-function modulus)**.** *Let $\mu_t$ be the statistical solution law at time $t$ and assume (27). Then for any cutoff $K_c \geq 1$,*

$$W_2\big(\mu_t, \, (P_{\leq K_c})_{\#}\mu_t\big) \; \lesssim \; \omega\big(c_d/K_c\big)^{1/2}, \tag{30}$$

*with $c_d$ as in Lemma 4.8.*

### 4.3.3  Capacity–coverage–fit decomposition for one-step RF

Finally, we combine the coverage term with the training error of a one-step rectified-flow surrogate. Let $S_{\Delta t} : \mathcal{H} \to \mathcal{H}$ denote the PDE solution operator over a physical step $\Delta t$, and write

$$\mu_{t+\Delta t} := (S_{\Delta t})_{\#}\mu_t.$$

Assume the rectified-flow propagator $\mathcal{T}_{\Delta t}^\theta$ aims at the bandlimited target $P_{\leq K_c}(S_{\Delta t}u)$ in mean-square. Concretely, sample $u \sim \mu_t$ and an internal noise seed $\xi \sim \mathcal{N}(0, I)$, and define the RF output $\hat{u} := \mathcal{T}_{\Delta t}^\theta(u, \xi)$. We measure the training fit by

$$\varepsilon_{\text{train}}^2(t) := \mathbb{E}_{u \sim \mu_t} \mathbb{E}_{\xi \sim \mathcal{N}(0, I)} \big\| P_{\leq K_c}(S_{\Delta t}u) - \hat{u} \big\|_{L^2}^2. \tag{31}$$

Finally, we define the one-step RF law by

$$\hat{\mu}_{t+\Delta t} := \text{Law}(\hat{u}).$$

**Proposition 4.11** (One-step law-level error: capacity, coverage, and fit)**.** *Under the structure-function modulus assumption (27), the one-step $W_2$ error between the PDE law and the RF law satisfies*

$$W_2\big(\mu_{t+\Delta t}, \hat{\mu}_{t+\Delta t}\big) \; \leq \; C_d \, \omega\big(c_d/K_c\big)^{1/2} + \varepsilon_{\text{train}}(t), \tag{32}$$

*where the first term is a* coverage *error determined by the structure-function modulus and the cutoff $K_c$, and the second is a* fit *error determined by training.*

*Proof sketch.* Couple $u \sim \mu_t$ and $\widehat{u}$ using the same input/noise realization. Then

$$W_2(\mu_{t+\Delta t}, \widehat{\mu}_{t+\Delta t}) \leq \left( \mathbb{E} \| S_{\Delta t} u - \widehat{u} \|_{L^2}^2 \right)^{1/2}.$$

Insert the resolved target $P_{\leq K_c} S_{\Delta t} u$ and use Minkowski's inequality:

$$\left( \mathbb{E} \| S_{\Delta t} u - \widehat{u} \|_{L^2}^2 \right)^{1/2} \leq \left( \mathbb{E} \| P_{>K_c} S_{\Delta t} u \|_{L^2}^2 \right)^{1/2} + \left( \mathbb{E} \| P_{\leq K_c} S_{\Delta t} u - \widehat{u} \|_{L^2}^2 \right)^{1/2}.$$

The first term is bounded by Lemma 4.8 applied at time $t + \Delta t$, and the second term is $\varepsilon_{\text{train}}(t)$. $\qquad\square$

**Interpretation and experimental validation.** At a fixed resolution (fixed $K_c$), Proposition 4.11 says that even a perfectly trained rectified flow cannot beat the coverage term $\omega(c_d/K_c)^{1/2}$ dictated by small-scale physics. This is precisely why spectrum matching (Figure 3) is a non-negotiable diagnostic on multiscale flows: if the surrogate loses the high-frequency tail, it is *necessarily* paying a coverage penalty in law. The remaining error is controlled by the training fit and the straightness properties of the learned velocity (via the continuous-time bounds in Section 4.1). Rectified flows improve efficiency precisely by reducing the curvature term in these bounds, so that a given coverage+fit accuracy can be reached with many fewer ODE steps than a diffusion-based surrogate such as GenCFD (Section 5.7).

## 5 Correctness and Curvature-Controlled Discretization

**Organization.** The previous subsection isolated the two dynamical ingredients that matter once the spatial resolution is fixed: the *fit* of the learned velocity field to the ideal conditional rectified velocity, and the *discretization error* incurred when the learned ODE is solved with finitely many steps. We now add three complementary ingredients. First, we state the law-level correctness result for the ideal conditional barycentric velocity. Second, because our EMA/blending controller was introduced specifically to stabilize *OOD sampling*, we state a separate OOD-focused theorem for the blended sampler; this theorem is distinct in nature from the generic correctness statement and should be read as a structural analysis of the regime studied empirically in Section 7. Third, we combine the finite-resolution coverage term from Section 4.3, the continuous-time fit bound, and the discretization bound into a single conditional law-level decomposition:

$$\text{coverage} + \text{fit} + \text{discretization}.$$

Straightness enters the generic theory only through the discretization term. The EMA theorem below addresses the additional, practically important OOD regime in which the learned field develops oscillatory errors along rectification time.

### 5.1 Exact transport under the conditional barycentric rectified velocity

**Proposition 5.1** (Exact conditional transport under the barycentric velocity)**.** *Fix a condition c. Let $\pi_0^c, \mu_1^c \in \mathcal{P}(\mathcal{U})$ be the reference and target conditional laws, let $\gamma^c$ be a coupling of $(\pi_0^c, \mu_1^c)$, and define the rectified interpolation*

$$U_\tau = (1 - \tau) \Xi + \tau U_f, \qquad (\Xi, U_f) \sim \gamma^c.$$

*Let*

$$\rho_\tau^c := \text{Law}(U_\tau \mid C = c)$$

*and define the conditional barycentric velocity*

$$v^\star(u, \tau; c) = \mathbb{E}[U_f - \Xi \mid U_\tau = u, C = c], \qquad \rho_\tau^c\text{-a.e. } u.$$

*Assume, on the finite-dimensional grid/Galerkin discretization used by the surrogate, that the continuity equation*

$$\partial_\tau \rho_\tau^c + \nabla \cdot \left( \rho_\tau^c \, v^\star(\cdot, \tau; c) \right) = 0 \tag{33}$$

*admits a unique flow-map $X_\tau^c$ associated to*

$$\dot{u}_\tau = v^\star(u_\tau, \tau; c).$$

*Then*

$$(X_1^c)_\# \pi_0^c = \mu_1^c.$$

*In other words, the ideal conditional barycentric rectified velocity transports the reference law exactly to the target conditional law.*

*Proof sketch.* By construction, $\rho_\tau^c$ is the law of the random interpolation $U_\tau = (1-\tau)\Xi + \tau U_f$. For every smooth test function $\varphi$,

$$\frac{d}{d\tau}\, \mathbb{E}[\varphi(U_\tau) \mid C = c] = \mathbb{E}[\nabla\varphi(U_\tau) \cdot (U_f - \Xi) \mid C = c].$$

Conditioning further on $U_\tau$ gives

$$\frac{d}{d\tau}\, \mathbb{E}[\varphi(U_\tau) \mid C = c] = \mathbb{E}[\nabla\varphi(U_\tau) \cdot v^\star(U_\tau, \tau; c) \mid C = c],$$

which is exactly the weak form of (33). Since $\rho_0^c = \pi_0^c$ and uniqueness holds for the continuity equation in the chosen finite-dimensional setting, it follows that $\rho_\tau^c = (X_\tau^c)_\# \pi_0^c$ for all $\tau$, hence $\rho_1^c = (X_1^c)_\# \pi_0^c = \mu_1^c$. $\square$

Proposition 5.1 is the law-level correctness statement underlying the method. It is the standard rectified-flow continuity-equation argument applied conditionally: in the finite-dimensional surrogate space, the ideal conditional rectified-flow ODE transports the reference law to the target conditional law exactly. In the infinite-dimensional notation, the same computation should be read as a weak continuity-equation identity unless a separate flow-map well-posedness theory is imposed. The only remaining issues are therefore approximation of the ideal velocity by $v_\theta$, numerical discretization of the learned ODE, and the unavoidable loss of unresolved high frequencies at finite spatial resolution.

## 5.2 A compact interpretation of straightness

The local truncation error result in Lemma 4.2 shows that the relevant notion of straightness is not a purely geometric property of the sample path $u_\tau$ itself, but the variation of the learned *velocity field along that path*:

$$\frac{d}{d\tau} v_\theta(\widehat{U}_\tau, \tau; c) = \partial_\tau v_\theta(\widehat{U}_\tau, \tau; c) + J_u v_\theta(\widehat{U}_\tau, \tau; c)\, v_\theta(\widehat{U}_\tau, \tau; c).$$

This is the precise quantity that determines how many explicit steps are required to integrate the learned transport stably. When this trajectory-wise variation is small, the learned flow behaves almost like "chord shooting" and very few ODE steps suffice. When it oscillates strongly, the sampler must reduce the step size or regularize the velocity direction, which is exactly the role of the EMA-based controller introduced earlier.

**The OOD regime.** In the out-of-distribution conditional settings studied in our experiments, the learned velocity field can encounter regions of state space that were less thoroughly represented during training. Empirically this manifests as stronger oscillations of $\tau \mapsto v_\theta(u_\tau, \tau; c)$, even when the overall conditional sample quality remains good. The EMA-based straightness proxy is designed precisely for this regime: it does not change the learned law in the continuous-time limit, but it makes the numerical integration of the learned ODE substantially more stable when these oscillations appear.

## 5.3 OOD stabilization by EMA blending

The EMA-based controller used in our sampler was not introduced to establish generic correctness of rectified flow. Its practical purpose is narrower: in the out-of-distribution macro–micro regime studied in Section 7, the learned velocity field can become substantially more oscillatory from one rectification step to the next, which makes explicit integration brittle. The role of EMA/blending is to suppress this oscillatory component while preserving the slowly varying part of the transport. Accordingly, the theorem below is not a replacement for the exact-fit comparison of Theorem 4.1; it is an additional structural result for the OOD stabilization mechanism.

**Setup.** Fix a condition $c$ and let $U_\tau$ solve the ideal conditional rectified ODE

$$\dot{U}_\tau = v^\star(U_\tau, \tau; c), \qquad U_0 \sim \pi_0^c. \tag{34}$$

Along this ideal path, define

$$g(\tau) := v^\star(U_\tau, \tau; c), \tag{35}$$

and the learned-field error

$$\varepsilon(\tau) := v_\theta(U_\tau, \tau; c) - g(\tau). \tag{36}$$

Thus

$$v_\theta(U_\tau, \tau; c) = g(\tau) + \varepsilon(\tau).$$

Let $E_\lambda$ denote the initialized causal exponential smoother in rectification time. Equivalently, for a trajectory-valued signal $f$, set $E_\lambda f = m$, where

$$\lambda \dot{m}(\tau) = f(\tau) - m(\tau), \qquad m(0) = f(0). \tag{37}$$

Thus

$$(E_\lambda f)(\tau) = e^{-\tau/\lambda} f(0) + \frac{1}{\lambda} \int_0^\tau e^{-(\tau-s)/\lambda} f(s) \, ds.$$

This initialization preserves constants, which is essential for the bias estimate below.

We measure the total approximation error along the ideal path by

$$\varepsilon_{\text{approx}}^2(c) := \int_0^1 \mathbb{E}\big[\|\varepsilon(\tau)\|^2 \,\big|\, C = c\big] \, d\tau. \tag{38}$$

We also define the total along-trajectory variation of the ideal conditional velocity by

$$S_{*,\text{tot}}^2(c) := \int_0^1 \mathbb{E}\big[\|\dot{g}(\tau)\|^2 \,\big|\, C = c\big] \, d\tau, \tag{39}$$

where

$$\dot{g}(\tau) = \partial_\tau v^\star(U_\tau, \tau; c) + J_u v^\star(U_\tau, \tau; c) \, v^\star(U_\tau, \tau; c). \tag{40}$$

This is exactly the same total along-trajectory derivative that drives the local truncation error in Lemma 4.2.

**Oscillatory-error regime.** The EMA controller is useful only when the learned error has a genuinely oscillatory component in rectification time. We encode this directly through the smoothing ratio

$$\sigma^2(c, \lambda) := \frac{\int_0^1 \mathbb{E}\big[\|E_\lambda \varepsilon(\tau)\|^2 \,\big|\, C = c\big] \, d\tau}{\int_0^1 \mathbb{E}\big[\|\varepsilon(\tau)\|^2 \,\big|\, C = c\big] \, d\tau}. \tag{41}$$

The structural OOD assumption is that $\sigma(c, \lambda) < 1$, meaning that the smoother removes a nontrivial part of the learned error along the rectification trajectory. If $\varepsilon_{\text{approx}}(c) = 0$, the raw approximation error already vanishes and we use the harmless convention $\sigma(c, \lambda) = 0$. For the corresponding whole-line exponential filter, the transfer function is

$$H_\lambda(\omega) = \frac{1}{1 + i\lambda\omega},$$

so temporal components with $|\omega| \gtrsim 1/\lambda$ are damped. This motivates the condition $\sigma(c, \lambda) < 1$ in regimes where the OOD error has substantial high-frequency content in rectification time. The actual sampler in our experiments uses discrete EMA updates and an adaptive blend weight $\alpha_t$. To isolate the mechanism analytically, we consider the continuous-time fixed-$\alpha$ extended-state surrogate

$$\dot{\widetilde{U}}_\tau = (1 - \alpha) v_\theta(\widetilde{U}_\tau, \tau; c) + \alpha m_\tau, \tag{42}$$

$$\lambda \dot{m}_\tau = v_\theta(\widetilde{U}_\tau, \tau; c) - m_\tau, \qquad m_0 = v_\theta(\widetilde{U}_0, 0; c), \tag{43}$$

with $\widetilde{U}_0 = U_0$ and $\alpha \in [0, 1]$. This idealization captures the same bias–variance tradeoff as the practical adaptive controller: smoothing suppresses oscillatory error but introduces a bias proportional to how much the *ideal* field itself varies along rectification time.

**Theorem 5.2** (EMA stabilization in the oscillatory OOD regime)**.** *Assume the compared ODE dynamics satisfy the following stability estimate on the region traversed by the trajectories: if two trajectories start from the same initial condition and their velocity fields differ along the comparison path by a perturbation $r$, then*

$$\left(\mathbb{E}\big[\|\widetilde{U}_1 - U_1\|^2 \mid C = c\big]\right)^{1/2} \leq C_{\mathrm{stab}}\,\|r\|_{L^2_\tau L^2_\Omega(c)}. \tag{44}$$

*Assume also that $\sigma(c, \lambda) < 1$ in (41). Then there exists a universal constant $C_{\mathrm{ema}} > 0$ such that*

$$\left(\mathbb{E}\big[\|\widetilde{U}_1 - U_1\|^2 \mid C = c\big]\right)^{1/2} \leq C_{\mathrm{stab}}\Big[\big(1 - \alpha(1 - \sigma(c, \lambda))\big)\,\varepsilon_{\mathrm{approx}}(c) + C_{\mathrm{ema}}\,\alpha\,\lambda\,S_{*,\mathrm{tot}}(c)\Big]. \tag{45}$$

*The corresponding raw-field bound is $C_{\mathrm{stab}}\varepsilon_{\mathrm{approx}}(c)$. Consequently, EMA blending gives a strictly sharper upper bound whenever*

$$\big(1 - \sigma(c, \lambda)\big)\,\varepsilon_{\mathrm{approx}}(c) \; > \; C_{\mathrm{ema}}\,\lambda\,S_{*,\mathrm{tot}}(c). \tag{46}$$

*Proof sketch.* The stability estimate (44) reduces the terminal error to the size of the velocity perturbation along the ideal path. For the raw learned field this perturbation is $r_{\mathrm{raw}} = \varepsilon$. For the EMA-blended field, along the ideal path,

$$(1 - \alpha)v_\theta(U_\tau, \tau; c) + \alpha E_\lambda[v_\theta(U., \cdot; c)](\tau) - g(\tau) = (1 - \alpha)\varepsilon(\tau) + \alpha E_\lambda\varepsilon(\tau) + \alpha(E_\lambda g(\tau) - g(\tau)).$$

By the definition of $\sigma(c, \lambda)$,

$$\|(1 - \alpha)\varepsilon + \alpha E_\lambda\varepsilon\|_{L^2_\tau L^2_\Omega(c)} \leq \big(1 - \alpha + \alpha\sigma(c, \lambda)\big)\varepsilon_{\mathrm{approx}}(c).$$

For the initialized exponential smoother, constants are preserved and the standard filter estimate gives

$$\|E_\lambda g - g\|_{L^2_\tau L^2_\Omega(c)} \leq C_{\mathrm{ema}}\lambda\|\dot{g}\|_{L^2_\tau L^2_\Omega(c)} = C_{\mathrm{ema}}\lambda S_{*,\mathrm{tot}}(c).$$

Combining the two bounds yields (45); the improvement criterion follows by comparing with the raw-field bound. $\square$

The bound (45) has the exact interpretation needed for the OOD sampler study:

- The factor $(1 - \sigma)\varepsilon_{\mathrm{approx}}$ is the *oscillatory error removed by EMA smoothing*.

- The term $\lambda S_{*,\mathrm{tot}}$ is the *bias paid for smoothing*, and it is controlled by how much the *ideal conditional rectified velocity* varies along its own trajectories in rectification time.

Thus EMA blending is beneficial precisely when the learned OOD error is sufficiently oscillatory while the ideal rectified transport remains slowly varying in rectification time.

**Experimental considerations.** This is exactly the situation targeted in Section 7. There the model is frozen and only the sampler is varied. In the OOD macro–micro regime, the learned field develops pronounced step-to-step oscillations, which increases $\varepsilon_{\mathrm{approx}}$ and makes the high-frequency component of the error non-negligible, hence $\sigma < 1$. At the same time, the ideal conditional rectified transport remains comparatively straight in rectification time, because the rectified-flow objective is designed to regress onto an approximately $\tau$-stationary chord direction. The practical effect is the one seen in Tables 2–4: EMA-based smoothing stabilizes the sampler and improves the NFE–error tradeoff without harming conditional diversity.

### 5.4 Master bound

We now combine the finite-resolution coverage term from Section 4.3, the continuous-time fit bound from Theorem 4.1, and the discretization bound from Corollary 4.3. The fit comparison must be made against the same resolved target law that appears in the coverage decomposition.

Fix a condition $c$, and let $\mu_1^c$ denote the target conditional law at the forecast time. Let $K_c$ be the effective spatial cutoff of the surrogate class and define the resolved target law

$$\mu_{1,\leq K_c}^c := (P_{\leq K_c})_{\#}\mu_1^c.$$

Let $v_{\leq K_c}^\star(\cdot, \tau; c)$ denote the conditional barycentric rectified velocity associated with a coupling between $\pi_0^c$ and $\mu_{1,\leq K_c}^c$, and let $U_\tau^{\leq K_c}$ be its ideal flow. By Proposition 5.1,

$$\mathrm{Law}(U_1^{\leq K_c} \mid C = c) = \mu_{1,\leq K_c}^c.$$

Let $\widehat{\mu}_1^{c,\mathrm{cts}}$ be the law of the learned *continuous-time* rectified flow initialized with the same source sample, and let $\widehat{\mu}_1^{c,N}$ be the law of the learned *discretized* rectified flow obtained with $N$ explicit steps.

**Theorem 5.3** (Conditional law-level error decomposition). *Assume that the target conditional law $\mu_1^c$ satisfies the structure-function modulus hypothesis of Corollary 4.10, so that*

$$W_2\big(\mu_1^c, \mu_{1,\leq K_c}^c\big) \ \lesssim\ \omega(c_d/K_c)^{1/2}. \tag{47}$$

*Assume also that the learned velocity field satisfies the hypotheses of Theorem 4.1 and Corollary 4.3. Define the resolved fit error*

$$\varepsilon_{\mathrm{fit},K_c}(c)^2 = \int_0^1 \mathbb{E}\big[\|v_\theta(U_\tau^{\leq K_c}, \tau; c) - v_{\leq K_c}^\star(U_\tau^{\leq K_c}, \tau; c)\|^2 \,\big|\, C = c\big]\, d\tau.$$

*Then there exists a constant $C(L) > 0$ such that*

$$W_2\big(\mu_1^c, \widehat{\mu}_1^{c,N}\big) \ \lesssim\ \underbrace{\omega(c_d/K_c)^{1/2}}_{coverage} + \underbrace{e^L \varepsilon_{\mathrm{fit},K_c}(c)}_{fit} + \underbrace{C(L)\, \varepsilon_{\mathrm{disc}}(c; N)}_{discretization}, \tag{48}$$

*where*

$$\varepsilon_{\mathrm{disc}}(c; N) = \frac{1}{N}\int_0^1 \Big(\mathbb{E}\big[\|\partial_\tau v_\theta(\widehat{U}_\tau, \tau; c) + J_u v_\theta(\widehat{U}_\tau, \tau; c)\, v_\theta(\widehat{U}_\tau, \tau; c)\|^2 \,\big|\, C = c\big]\Big)^{1/2} d\tau.$$

*Proof sketch.* Insert the resolved target law $\mu_{1,\leq K_c}^c$ and the learned continuous-time law $\widehat{\mu}_1^{c,\mathrm{cts}}$ as intermediate terms:

$$W_2(\mu_1^c, \widehat{\mu}_1^{c,N}) \leq W_2(\mu_1^c, \mu_{1,\leq K_c}^c) + W_2(\mu_{1,\leq K_c}^c, \widehat{\mu}_1^{c,\mathrm{cts}}) + W_2(\widehat{\mu}_1^{c,\mathrm{cts}}, \widehat{\mu}_1^{c,N}).$$

The first term is the coverage term (47). The second term is controlled by Theorem 4.1 applied to the resolved ideal flow $U_\tau^{\leq K_c}$ and the resolved barycentric field $v_{\leq K_c}^\star$. The third term is controlled by Corollary 4.3. □

Equation (48) matches the empirical instantiation in Section 6. The spectrum plots diagnose the coverage term by revealing whether the surrogate preserves high-wavenumber energy. The one-point Wasserstein and moment statistics diagnose overall conditional-law fidelity, which reflects both coverage and fit. Finally, the speed advantage of ReFlow is a discretization phenomenon: because the learned transport is substantially straighter along rectification time, the same conditional-law accuracy is reached with far fewer evaluations.

### 5.5 Optional regularity interpretation via structure functions

For completeness, we record the standard heuristic that structure-function control corresponds to Besov regularity and therefore to a quantitative high-frequency tail bound.

**Lemma 5.4** (Structure functions and Besov smoothness). *Let $0 < s < 1$ and $u \in L^2(\mathbb{T}^d)$. Suppose that*

$$S_r^2(u) \le Cr^{2s} \qquad \text{for all } r \in (0, 1].$$

*Then $u \in B_{2,\infty}^s(\mathbb{T}^d)$, with*

$$\|u\|_{B_{2,\infty}^s} \lesssim \|u\|_{L^2} + C^{1/2}.$$

*Equivalently, for $0 < s < 1$,*

$$\sup_{0 < |h| \le 1} \frac{\|u(\cdot + h) - u(\cdot)\|_{L^2}}{|h|^s} \lesssim \|u\|_{L^2} + C^{1/2}.$$

*At the law level, the consequence used in our coverage estimates is the averaged tail bound*

$$S_r^2(\mu) \lesssim r^{2s} \implies \int \|P_{>K}u\|_{L^2}^2 \, d\mu(u) \lesssim K^{-2s}.$$

*Remark* 5.5. In turbulence one often observes inertial-range power laws

$$S_r^2(\mu) \sim r^{\zeta_2}.$$

Heuristically this corresponds to $s \approx \zeta_2/2$, so the coverage term in Theorem 5.3 can be interpreted as a direct law-level manifestation of inertial-range scaling.

## 5.6 Implementation Details for PDEs

Our model is a *conditional noise-to-solution rectified flow*. Given an input/conditioning field $u_i$ (and, when relevant, boundary data or other known inputs), we learn a velocity field $v_\theta(x, \tau; u_i)$, where:

- $u_i$ is the *conditioning variable*,

- $x_\tau$ is the *noise-initialized output state* transported in rectification time,

- $\tau \in [0, 1]$ is the internal rectification-time variable.

The key point is that $u_i$ is *not* the state being evolved by the rectified flow. It remains fixed throughout the trajectory and enters the network only as side information.

**Training pairs.** We assume access to paired samples

$$(u_i, u_f), \qquad u_f = \mathcal{S}_{\Delta t}(u_i),$$

where $\mathcal{S}_{\Delta t}$ denotes the PDE solution operator over the forecast horizon of interest. In the stochastic/statistical setting considered in this paper, repeated perturbations around the same macro input $u_i$ induce a conditional law of possible outputs; the purpose of the generator is to sample from this conditional output law.

**Rectified interpolation used in training.** For each training pair $(u_i, u_f)$, we sample source noise

$$\xi \sim \mathcal{N}(0, I)$$

and rectification time

$$\tau \sim \text{Unif}[0, 1].$$

We then form the linear rectified interpolation

$$x_\tau = (1 - \tau)\xi + \tau u_f. \tag{49}$$

With this choice, the target velocity along the interpolation path is simply the constant chord direction

$$\dot{x}_\tau = u_f - \xi. \tag{50}$$

This is exactly the conditional rectified-flow objective introduced in Section 4.1: the network is trained to predict the transport direction from the current interpolated state $x_\tau$ and the condition $u_i$.

---

**Algorithm 1** Conditional Rectified-Flow Training (PDE Setting)

---

**Require:** Dataset of paired states $\{(u_i, u_f)\}$ with $u_f = \mathcal{S}_{\Delta t}(u_i)$; velocity model $v_\theta$

1: **for** each training iteration **do**
2:      Sample a training pair $(u_i, u_f)$
3:      Sample $\xi \sim \mathcal{N}(0, I)$
4:      Sample $\tau \sim \mathcal{U}(0, 1)$
5:      Form the rectified interpolation
$$x_\tau = (1 - \tau)\xi + \tau u_f$$
6:      Compute the conditional rectified-flow loss
$$\mathcal{L}(\theta) = \left\| v_\theta(x_\tau, \tau; u_i) - (u_f - \xi) \right\|^2$$
7:      Update $\theta$ by stochastic gradient descent / Adam
8: **end for**

---

**Network parameterization.** The neural network takes as input the transported state $x_\tau$, the conditioning field $u_i$, and a learned embedding of rectification time:

$$v_\theta(x_\tau, \tau; u_i) = \mathrm{UNet}_\theta\big(x_\tau, \, u_i, \, \Gamma(\tau)\big), \tag{51}$$

where

$$\Gamma(\tau) = \mathrm{MLP}\big(\mathrm{PE}(\tau)\big) \in \mathbb{R}^{d_{\mathrm{emb}}}$$

is the learned embedding of the scalar rectification time, where PE denotes a sinusoidal positional encoding. Concretely, $x_\tau$ and $u_i$ are concatenated channelwise at the input, while each network block receives its own learned affine projection of the shared vector $\Gamma(\tau)$, for example as scale and shift parameters in feature-wise modulation. Thus the embedding vector is globally shared across the network, while its effect on features is block-specific.

**Architecture overview.** Our implementation uses a U-Net / UViT-style backbone with residual blocks, attention, downsampling to a bottleneck, and symmetric upsampling with skip connections; see the Supplementary Material for the exact architecture. The output head predicts the velocity field itself, i.e. an estimate of the chord direction $u_f - \xi$ conditioned on the current transported state and on the PDE input. This is in contrast to score-based diffusion models, which predict either a score or a denoising residual inside a stochastic reverse-time sampler.

**Sampling procedure.** At inference time, the condition $u_i$ is fixed. We draw

$$\xi \sim \mathcal{N}(0, I),$$

set

$$x_0 = \xi,$$

and integrate the learned conditional ODE

$$\frac{d}{d\tau} x_\tau = v_\theta(x_\tau, \tau; u_i), \qquad x_{\tau=0} = \xi. \tag{52}$$

The final state $x_1$ is then interpreted as a sample from the learned approximation to the conditional output law associated with $u_i$.

The learned rectified transport is often straight enough that a small fixed number of Euler steps already gives excellent performance. The EMA/blending controller is introduced specifically to stabilize sampling in the more delicate OOD regime studied in Section 7, where the learned velocity exhibits stronger oscillations across rectification time. In that regime, the controller acts as a low-pass filter on the oscillatory component of the learned field and improves the NFE–error tradeoff without changing the underlying trained model.

---

**Algorithm 2** Conditional Rectified-Flow Sampling with Optional Curvature-Aware Control

---

**Require:** Condition/input field $u_i$; trained velocity model $v_\theta$; number of steps $N$ or adaptive controller
1: Sample $\xi \sim \mathcal{N}(0, I)$ and set $x_0 = \xi$
2: Initialize $\tau_0 = 0$
3: **if** using fixed-step integration **then**
4:     **for** $t = 0, \ldots, N-1$ **do**
5:         $v_t \leftarrow v_\theta(x_t, \tau_t; u_i)$
6:         $\Delta\tau \leftarrow 1/N$
7:         $x_{t+1} \leftarrow x_t + \Delta\tau\, v_t$
8:         $\tau_{t+1} \leftarrow \tau_t + \Delta\tau$
9:     **end for**
10: **else**
11:     Initialize EMA state $m_0$
12:     **for** $t = 0, 1, \ldots$ until $\tau_t \geq 1$ **do**
13:         $v_t \leftarrow v_\theta(x_t, \tau_t; u_i)$
14:         Update EMA:
$$m_t \leftarrow \lambda m_{t-1} + (1-\lambda)v_t$$
15:         Compute straightness proxy:
$$s_t \leftarrow \|v_t - m_t\|_2$$
16:         Choose blend weight $\alpha_t$ and step size $\Delta\tau_t$ from $s_t$
17:         Form blended velocity:
$$\widetilde{v}_t \leftarrow (1-\alpha_t)v_t + \alpha_t m_t$$
18:         Update state:
$$x_{t+1} \leftarrow x_t + \Delta\tau_t\, \widetilde{v}_t$$
19:         $\tau_{t+1} \leftarrow \min\{1, \tau_t + \Delta\tau_t\}$
20:     **end for**
21: **end if**
22: **return** $x_1$

---

**Relation to the theory.** This implementation matches the conditional theory exactly:

- the source law is Gaussian noise,

- the transported state is $x_\tau$,

- the condition $u_i$ is fixed throughout sampling,

- the ideal target velocity is the conditional barycentric transport direction,

- and the EMA/blending mechanism is an *additional OOD stabilization device*, not part of the generic exactness statement.

**Practical efficiency.** Because rectified-flow training promotes transports whose velocity varies comparatively slowly along rectification time, accurate sampling often requires only a handful of ODE evaluations. This is the computational advantage that distinguishes ReFlow from diffusion-based PDE surrogates: instead of integrating a long reverse-time stochastic process, we solve a short deterministic conditional transport ODE.

## 5.7 Computational Efficiency Comparison

Conditional diffusion surrogates such as GenCFD generate samples by integrating a reverse-time stochastic process whose trajectory typically requires many network evaluations before reaching the target conditional law. Even with accelerated samplers, this procedure usually remains expensive because each evaluation acts only as a local refinement step. In our setting, this cost is especially significant since every network call operates on a moderate-to-high resolution spatial field.

ReFlow replaces this reverse-time stochastic sampling procedure by a deterministic *conditional transport ODE*. For a fixed input field $u_i$, we draw source noise $\xi$ and solve

$$\frac{d}{d\tau} x_\tau = v_\theta(x_\tau, \tau; u_i), \qquad x_0 = \xi,$$

returning $x_1$ as the generated sample. The computational advantage does not come from changing the conditioning information or the target conditional law, but from changing the *sampling geometry*: the learned rectified transport varies substantially less along rectification time than the stochastic diffusion trajectory, so explicit integration can be performed with far fewer evaluations.

This distinction is important. The benefit of rectified flow is not that it introduces new spatial capacity or accesses higher frequencies than diffusion at the same grid resolution. Both ReFlow and GenCFD operate under the same effective spectral cutoff imposed by resolution and architecture. Rather, rectification makes the *path from source noise to conditional output law easier to integrate numerically*. In the terminology of Section 5, rectified flow improves the *discretization* term: straighter learned transports require fewer function evaluations to reach the same law-level accuracy.

The curvature-aware sampler further improves this advantage in the regime where the learned velocity becomes oscillatory along rectification time. This is particularly relevant in the OOD macro–micro setting of Section 7, where we keep the trained model fixed and vary only the numerical sampler. There, the adaptive controller suppresses high-frequency oscillations of the learned velocity and allocates smaller steps only where needed, while retaining large steps on the straight portions of the transport.

Figure 2 quantifies the practical consequence. Across model sizes, ReFlow reaches the target conditional-law accuracy regime with only 4–8 ODE steps, whereas GenCFD typically requires 64–128 stochastic denoising steps. Since each step invokes a network of comparable scale, this reduction in neural function evaluations translates directly into lower inference cost.

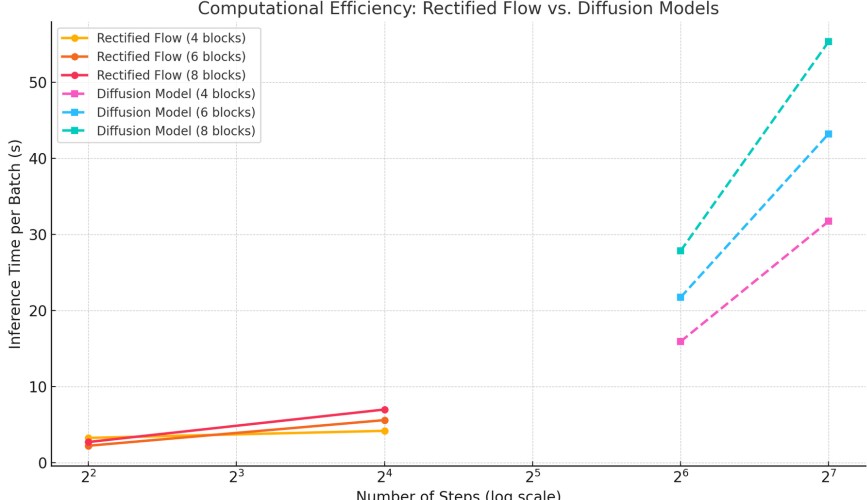

Figure 2: Inference cost versus model size. Both ReFlow and GenCFD use comparable backbone families, but ReFlow reaches the target conditional-law accuracy regime with substantially fewer neural function evaluations. In our experiments, ReFlow typically requires 4–8 ODE steps, whereas GenCFD requires 64–128 stochastic denoising steps. U-Net depth varies from 4 to 8 blocks (roughly 5.5M to 10.2M parameters).

## 6 Experiments

### 6.1 Experimental Setup

We benchmark our method, termed henceforth as *ReFlow*, against four neural surrogate models. *GenCFD* and *GenCFD ∘ FNO* are conditional diffusion surrogates following the framework introduced earlier in this section, whereas FNO and UViT serve as one-shot deterministic baselines. Detailed descriptions of all baselines are provided in **SM** L.

**Datasets for Training and Evaluation.** We consider three canonical 2D multi-scale PDE problems (see the Supplementary Material for precise mathematical formulations and solver details):

**SL2D** *Shear Layer in 2D.* This setup models the evolution of an incompressible shear layer, initially concentrated near a horizontal interface, which rolls up into intricate vortical structures over time. We generate approximate solutions by numerically solving the 2D Navier–Stokes system (with periodic boundaries) at a resolution of $512^2$, then downsample to $128^2$ for training and testing. The data has been generated with the spectral viscosity code Tadmor (1989); Rohner & Mishra (2024).

**CS2D** *Cloud-Shock Interaction in 2D.* In this popular compressible-flow test, a shock wave moves through a higher-density "cloud" region, triggering complex wave interactions and small-scale turbulent mixing in the wake. We solve the compressible Euler system at an effective resolution of $256^2$ using a GPU-optimized finite-volume ALSVINN code of Lye (2020); Fjordholm et al. (2020), then also downsample to $128^2$.

**RM2D** *Richtmeyer–Meshkov in 2D.* In this case, a shock wave interacts with a density interface separating two fluids, depositing vorticity and triggering vigorous mixing. We numerically solve the compressible Euler system on a $256^2$ grid, then downsample to $128^2$ to create the training and test data. The data has been generated with a high-resolution finite-volume ALSVINN code of Fjordholm et al. (2020)

**Training and evaluation protocol.** All models are trained on paired samples

$$(u_i, u_f), \qquad u_f = \mathcal{S}_{\Delta t}(u_i),$$

where $u_i$ is the input / conditioning field and $u_f$ is the PDE state at the forecast horizon. For the probabilistic models (GenCFD and ReFlow), the goal is to learn the *conditional output law*

$$p(u_f \mid u_i),$$

rather than only a single-point prediction. During training, however, the models are only given paired realizations $(u_i, u_f)$ from the dataset; the conditional law is learned implicitly from the collection of such pairs.

**Macro–micro evaluation and conditional laws.** Our main evaluation target is the conditional law associated with a *fixed* macroscopic input. Concretely, for macro index $a$ we choose a nominal macro/input state $\widehat{u}_i^{(a)}$. This macro state is constructed by the dataset-specific initial-condition generator, either from the matched family or from the shifted parameter range used for OOD evaluation. We then generate microscopic perturbations

$$u_i^{(a,b)} = \widehat{u}_i^{(a)} + \delta u_i^{(a,b)}, \qquad b = 1, \ldots, N_{\mathrm{micro}},$$

and evolve them with the high-fidelity solver to obtain

$$u_f^{(a,b)} = \mathcal{S}_{\Delta t}\big(u_i^{(a,b)}\big).$$

The solver ensemble $\{u_f^{(a,b)}\}_{b=1}^{N_{\mathrm{micro}}}$ is our empirical reference approximation to the output law at fixed macro condition $\widehat{u}_i^{(a)}$. The model-generated law $\widehat{p}(\cdot \mid \widehat{u}_i^{(a)})$ is obtained by holding $\widehat{u}_i^{(a)}$ fixed and sampling the model's latent noise.

Thus, throughout the experiments, the conditioning/input field is held fixed at the macro level, while diversity appears at the micro level through the induced spread of outputs. This is exactly the statistical object that ReFlow and GenCFD are designed to model.

**Matched and shifted conditional regimes.** We report two evaluation regimes:

1. **Matched regime.** The macro/input state $\widehat{u}_i$ is drawn from the same broad family as the training inputs. This is the easier setting, although the evaluation is still conditional because the model is asked to generate an ensemble at a fixed macro condition.

2. **Shifted / OOD regime.** The macro/input state $\widehat{u}_i$ is drawn from a different family or parameter range than those seen during training. This probes whether the learned conditional sampler remains statistically faithful when queried out of distribution.

In the rebuttal-driven experiments of Section 7, the macro–micro protocol is made explicit: the model is frozen, the macro conditions are chosen in an OOD regime, and only the numerical sampler is varied. This is precisely the setting targeted by the EMA-based stabilization analysis in Section 5.3.

**Role of deterministic baselines.** FNO and UViT are deterministic predictors and therefore do not directly define a conditional output law. To make them comparable on statistical metrics, we apply the same macro–micro perturbation protocol at the input level and aggregate their outputs across perturbed inputs into an empirical ensemble. This does *not* turn them into intrinsically probabilistic models; rather, it provides a controlled way to compare how much conditional variability they preserve under perturbations of the macro input. By contrast, ReFlow and GenCFD generate diversity intrinsically through noise sampling at fixed $u_i$.

**Training budget and architectural parity.** To keep the comparison computationally controlled, we train each model under a fixed optimization budget and report the best checkpoint achieved within that budget. Under the budgets used here, ReFlow and GenCFD are compared with matched backbone families and similar parameter counts, so the main difference is not representational scale but sampling mechanism. In particular, ReFlow's advantage should be read as a gain in *conditional-law accuracy per neural function evaluation*, rather than as a trivial consequence of larger models.

## 6.2 Evaluation Metrics

We evaluate the generative performance of each method using metrics that probe conditional-law fidelity rather than only pointwise prediction quality.

**Mean error.** Let $\bar{u}_{\mathrm{model}}$ and $\bar{u}_{\mathrm{ref}}$ denote the empirical means of the model-generated and reference conditional ensembles for a fixed macro/input state. We report the relative $L^2$ discrepancy between these means. This measures whether the surrogate captures the large-scale conditional average correctly.

**Standard-deviation error.** Let $\sigma_{\mathrm{model}}$ and $\sigma_{\mathrm{ref}}$ denote the pointwise empirical standard deviations of the generated and reference conditional ensembles. We report the normalized $L^2$ discrepancy between them. This measures whether the model reproduces the correct spatial pattern and magnitude of conditional variability.

**One-point Wasserstein distance.** To compare the local output distributions themselves, we use the spatially averaged one-point Wasserstein-1 distance. Let $\mu$ and $\widehat{\mu}$ be two laws on fields $u : D \to \mathbb{R}^m$. For each spatial point $x \in D$, define the one-point marginals

$$\mu_x := (\mathrm{ev}_x)_\# \mu, \qquad \widehat{\mu}_x := (\mathrm{ev}_x)_\# \widehat{\mu}.$$

Our reported metric is

$$W_1^{(1)}(\mu, \widehat{\mu}) := \int_D W_1(\mu_x, \widehat{\mu}_x)\, dx. \tag{53}$$

By Kantorovich–Rubinstein duality, each $W_1(\mu_x, \widehat{\mu}_x)$ controls the discrepancy of all pointwise 1-Lipschitz observables at the spatial location $x$. Thus $W_1^{(1)}$ is a natural spatially averaged $k = 1$ correlation-marginal metric in the statistical-solution sense Lanthaler et al. (2021b).

These three metrics probe complementary aspects of the conditional law:

- mean error measures whether the conditional average is correct,

- standard-deviation error measures whether the spread is correct,

- one-point Wasserstein distance measures whether the local conditional output distributions themselves are correct.

Together with the spectral diagnostics reported below, they provide a law-level view of whether a surrogate captures both large-scale structure and fine-scale conditional variability.

**Baselines and evaluation settings.** All results are reported under a fixed training budget per model, and the best checkpoint within that budget is retained. Under this budget, ReFlow is compared primarily against GenCFD as the corresponding conditional probabilistic baseline, using matched backbone families and similar parameter counts. Deterministic baselines such as FNO and UViT are included as references for one-shot prediction quality, but they should not be interpreted as intrinsically modeling the conditional output law.

At inference time, ReFlow requires only **8** rectified-flow ODE steps to reach the reported accuracy regime, whereas diffusion baselines such as GenCFD typically require **128** denoising steps. This is the central efficiency comparison of the paper: both methods target the same conditional-law prediction problem, but ReFlow reaches that regime with far fewer neural function evaluations.

For deterministic baselines, ensemble statistics are estimated by applying the macro–micro perturbation protocol at the input level and aggregating the resulting outputs. For ReFlow and GenCFD, diversity is generated intrinsically by sampling latent noise at fixed conditioning input $u_i$.

Since FNO, and UViT produce *deterministic* predictions, we approximate their spread by an *ensemble perturbation* strategy (**SM** K.2), where each baseline is fed slightly perturbed inputs and outputs are aggregated to form a sample set. In contrast, both the original GenCFD (baseline) and our *ReFlow* are inherently *probabilistic*, requiring no auxiliary procedure to generate diverse samples.

Table 1: Comparison of models across datasets CS2D, SL2D, and RM2D, in terms of mean relative error ($e_\mu$), standard deviation relative error ($e_\sigma$), and Wasserstein-1 distance ($W_1$). Best-performing model is colored in Blue and second-best performing model in Orange.

| Model | Metric | CS2D | | | | SL2D | | RM2D | | | |
|---|---|---|---|---|---|---|---|---|---|---|---|
| | | $\rho$ | $m_x$ | $m_y$ | $E$ | $u_x$ | $u_y$ | $\rho$ | $m_x$ | $m_y$ | $E$ |
| ReFlow | $e_\mu$ | 0.0477 | 0.0332 | 0.041 | 0.015 | 0.034 | 0.189 | 0.018 | 0.032 | 0.020 | 0.021 |
| | $e_\sigma$ | 0.1005 | 0.072 | 0.078 | 0.0584 | 0.071 | 0.077 | 0.032 | 0.046 | 0.046 | 0.085 |
| | $W_1$ | 0.0091 | 0.0107 | 0.0124 | 0.0143 | 0.0214 | 0.0164 | 0.0033 | 0.0036 | 0.0028 | 0.0017 |
| GenCFD | $e_\mu$ | 0.0830 | 0.0619 | 0.127 | 0.0280 | 0.039 | 0.267 | 0.0049 | 0.0010 | 0.0025 | 0.0025 |
| | $e_\sigma$ | 0.195 | 0.197 | 0.244 | 0.169 | 0.092 | 0.072 | 0.0351 | 0.0254 | 0.0435 | 0.0435 |
| | $W_1$ | 0.0151 | 0.0184 | 0.0185 | 0.0247 | 0.0276 | 0.0213 | 0.0092 | 0.0016 | 0.0049 | 0.0049 |
| UViT | $e_\mu$ | 0.111 | 0.078 | 0.127 | 0.0479 | 0.233 | 0.623 | 0.070 | 0.070 | 0.071 | 0.052 |
| | $e_\sigma$ | 0.415 | 0.343 | 0.412 | 0.296 | 0.28 | 0.178 | 0.425 | 0.570 | 0.570 | 0.617 |
| | $W_1$ | 0.087 | 0.085 | 0.158 | 0.172 | 0.270 | 0.256 | 0.060 | 0.055 | 0.060 | 0.036 |
| FNO | $e_\mu$ | 0.0989 | 0.0748 | 0.0930 | 0.0383 | 0.058 | 0.408 | 0.085 | 0.070 | 0.063 | 0.067 |
| | $e_\sigma$ | 0.2573 | 0.2779 | 0.3188 | 0.2777 | 0.137 | 0.110 | 0.366 | 0.443 | 0.436 | 0.450 |
| | $W_1$ | 0.1455 | 0.1660 | 0.1879 | 0.1884 | 0.184 | 0.150 | 0.096 | 0.051 | 0.056 | 0.062 |
| GenCFD ∘ FNO | $e_\mu$ | 0.0638 | 0.0454 | 0.0603 | 0.0204 | 0.055 | 0.393 | 0.040 | 0.027 | 0.039 | 0.041 |
| | $e_\sigma$ | 0.1099 | 0.0969 | 0.1245 | 0.0974 | 0.131 | 0.100 | 0.091 | 0.075 | 0.091 | 0.090 |
| | $W_1$ | 0.0467 | 0.0534 | 0.0774 | 0.0743 | 0.166 | 0.135 | 0.0269 | 0.0158 | 0.0293 | 0.0273 |

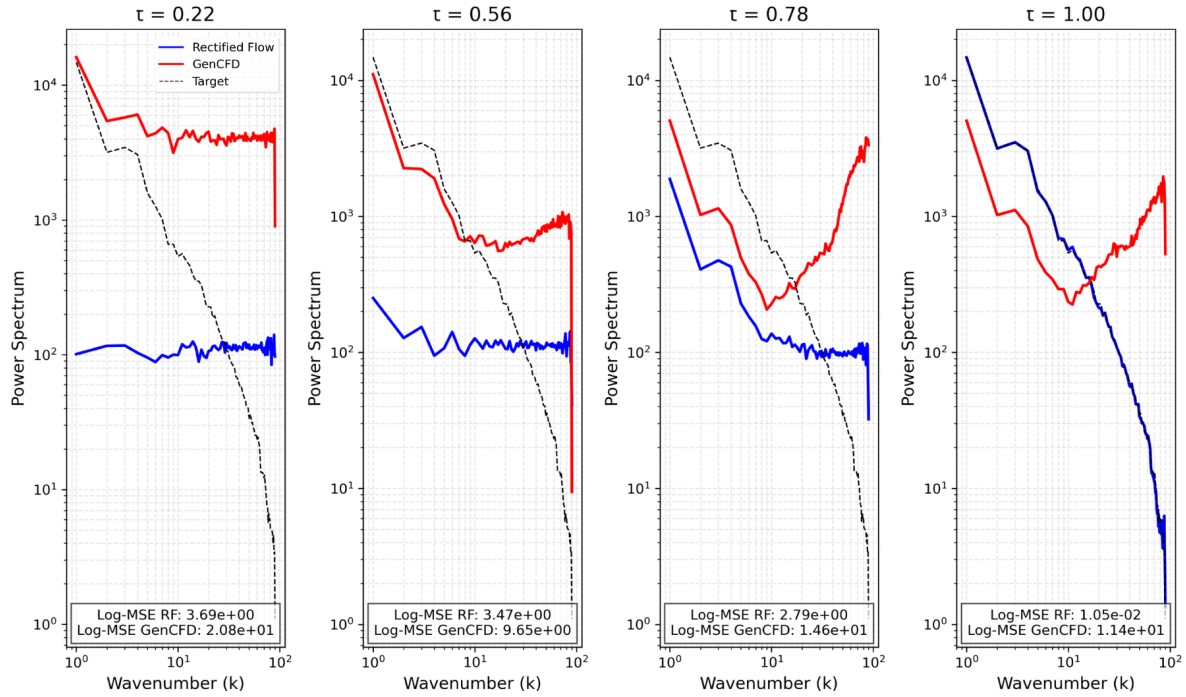

Figure 3: Energy-spectrum evolution for a random cloud–shock initial datum. ReFlow (blue) tracks high-wavenumber energy faster and more accurately than GenCFD (red).

## 6.3 Results.

Table 1 compares five surrogate models (ReFlow, together with the four baselines) on the three PDEs learning tasks using the mean error ($e_\mu$), spread error or *standard deviation* ($e_\sigma$) and the distribution-level Wasserstein-1 distance ($W_1$). On the cloud–shock and shear-layer datasets (CS2D, SL2D) the rectified-flow model (ReFlow) attains the lowest values on all three metrics, indicating that its samples most closely match both the mean behaviour and the higher-order statistics of the ground truth. The hybrid *GenCFD∘FNO*

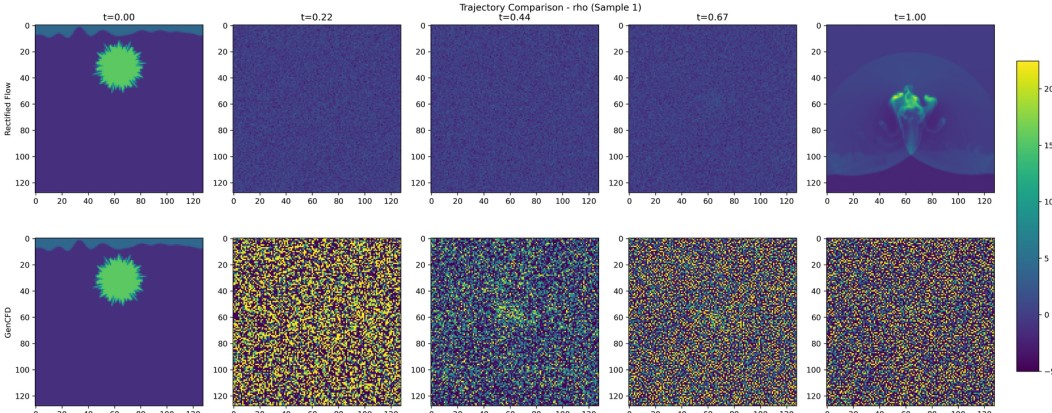

Figure 4: Density trajectories for Sample 1: ReFlow (top) versus GenCFD (bottom).

reduces the gap to ReFlow relative to its GenCFD component, but still records noticeably larger errors, whereas the one-shot deterministic baselines (FNO, UViT) show the largest deviations, especially in $e_\sigma$ and $W_1$, confirming their difficulty in capturing output variability. The RM2D benchmark is more mixed: GenCFD gives lower mean errors on several variables, while ReFlow gives the best one-point Wasserstein distances on most variables and uses far fewer function evaluations. We therefore interpret RM2D as supporting the efficiency and distributional-fidelity claim, not as evidence of uniform metric-wise dominance over diffusion. Under identical training budgets, ReFlow achieves lower errors on CS2D and SL2D compared to all baselines, including GenCFD with the same architecture and model size. This highlights ReFlow's superior convergence efficiency despite parity in model capacity and training iterations.

Moreover as visualized in the **SM**, ReFlow generates samples of very high quality, matching the qualitative features of the ground truth and capturing features such as (interacting) vortices as well as propagating shock waves very accurately. Another aspect where ReFlow really shines in being able to generate the correct point pdfs. Although already indicated in the very low 1-point Wasserstein distance errors from Table 1, we present 1-point pdfs in the **SM** to illustrate this observation.

Another key indicator of the quality of approximation of multiscale PDE solutions is the energy spectrum Molinaro et al. (2024). In Figure 3, we plot the (log) spectra for the energy variable of a randomly chosen Cloud Shock initial datum at four different Diffusion times, $\tau \in \{0.25, 0.50, 0.75, 1.00\}$ to observe that i) the solution is indeed highly multiscale, with spectra decaying as a power law over a range of frequencies ii) ReFlow approximates the spectra accurately within a few (5) steps and iii) on the other hand, for these few steps in solving the underlying reverse SDE, a score-based Diffusion model such as GenCFD, struggles to approximate the relevant small scales. It will require much more sampling steps to do so. These results also showcase how ReFlow is able to infer the correct statistical solutions with significantly less ODE solve steps and is hence, much faster, than a diffusion based model such as GenCFD. This gain in inference speed is also reinforced from Figure 4, where we compare ReFlow (RF) and GenCFD reconstructions of the density field $\rho$ for the Cloud-Shock interaction over five diffusion times $\tau \in \{0.00, 0.25, 0.50, 0.75, 1.00\}$. We see that ReFlow inpaints fine-scale features in far fewer steps, whereas GenCFD remains visibly much more noisy at these small number of steps. See **SM** M.7 for further visualizations of this effect.

## 7 Additional empirical analysis (integrators, robustness, and diversity)

This section consolidates additional experiments conducted with the goal of making the computational trade-offs and the role of the curvature-aware controller explicit. The emphasis is on (i) solver choices for the learned rectified-flow ODE, (ii) robustness of the controller to its hyperparameters, and (iii) distributional fidelity / diversity under the macro–micro evaluation protocol.

Table 2: **Fixed-step ODE integrators** at sampling time (ShearLayer2D, OOD macro–micro; 10 macros $\times$ 20 micros). We fix the trained rectified-flow model and vary only the integrator. NFE is the number of velocity evaluations; lower is faster. Cost$\times$Err is defined in (55).

| Method | Avg. NFE $\downarrow$ | Rel. $L^2$ Error $\downarrow$ (mean $\pm$ std) | Cost$\times$Err $\downarrow$ |
|---|---|---|---|
| Model-Baseline (built-in sampler) | 32.0 | $0.2533 \pm 0.0382$ | 8.11 |
| `torchdiffeq` Midpoint | 32.0 | $0.2533 \pm 0.0382$ | 8.11 |
| Euler (16 steps) | 16.0 | $0.2489 \pm 0.0371$ | 3.98 |
| RK2 (16 steps; 32 NFEs) | 32.0 | $0.2533 \pm 0.0379$ | 8.11 |
| RK4 (16 steps; 64 NFEs) | 64.0 | $0.2540 \pm 0.0381$ | 16.26 |

### 7.1 Protocol recap: macro–micro ensembles, OOD macros, and metrics

**Macro–micro conditional ensembles.** To evaluate *statistical fidelity* rather than only pointwise accuracy, we follow a macro–micro protocol. For each dataset, we select $N_{\mathrm{macro}}$ macroscopic states ("macros") and for each macro generate $N_{\mathrm{micro}}$ microscopic perturbations ("micros") yielding a reference ensemble of size $N_{\mathrm{macro}} \times N_{\mathrm{micro}}$. This makes diversity concrete: for each fixed macro, the ground-truth solver produces an empirical conditional distribution over micros, and the model produces its own empirical conditional distribution. We compare these distributions directly via Wasserstein distances (below), which is the relevant question for statistical surrogates.

**OOD macro selection.** Unless otherwise stated, we report results in an out-of-distribution (OOD) regime by choosing $N_{\mathrm{macro}} = 10$ macros from shifted initial-condition families / parameter ranges, and $N_{\mathrm{micro}} = 20$ micros per macro (200 conditional samples per dataset). All models are frozen; only the sampling/integration procedure is varied.

**Metrics.** We report: (i) *relative $L^2$ error* against the PDE solution,

$$\text{Rel-}L^2(\widehat{u}, u) := \frac{\|\widehat{u} - u\|_{L^2}}{\|u\|_{L^2}}, \tag{54}$$

(ii) *NFE* (neural function evaluations), i.e. the number of velocity-field evaluations per generated sample, and (iii) a combined cost–accuracy scalar

$$\text{Cost} \times \text{Err} := (\text{Rel-}L^2) \times (\text{NFE}), \tag{55}$$

which is convenient for summarizing the NFE–error plane (lower is better). Distributional fidelity is measured via empirical $W_1$ between conditional ensembles; we report both aggregated and per-variable $W_1$ (Section 7.5).

### 7.2 Integrator study at sampling time: fixed-step baselines vs. curvature-aware control

A recurring concern is whether the gains are merely a consequence of choosing a particular numerical solver. To isolate this, we fix the trained rectified-flow model and vary *only* the integrator used to solve the rectified-flow ODE at sampling time, starting from noise and integrating from rectified time $\tau = 0$ to $\tau = 1$. We include a reference implementation from `torchdiffeq` and standard explicit fixed-step schemes. We report the aggregated results on *ShearLayer2D* in the OOD macro–micro regime; the second benchmark exhibits the same qualitative trends and is reported in the appendix.

**Observation: Euler has a tuned "sweet spot" but remains brittle.** Among fixed-step schemes, Euler with 16 steps is the strongest baseline in this regime, improving on 32/64-NFE schemes in Cost$\times$Err. However, in multiscale mildly stiff regimes, fixed-step performance is highly sensitive to the chosen step budget: slightly fewer steps can degrade accuracy sharply, while increasing steps can *also* worsen error due to accumulated discretization bias along curved trajectories. Thus, even simple Euler is not a drop-in robust choice: it requires per-dataset/per-regime tuning to hit its efficient operating point.

Table 3: **Hyperparameter robustness** of the curvature-aware adaptive sampler (ShearLayer2D, OOD macro–micro). Across 20 controller configurations, Avg. NFE stays in a narrow band and relative errors vary by $< 1\%$, indicating the controller is not brittle and does not rely on a narrow tuning "sweet spot".

| Config | Avg. NFE ↓ | Rel. $L^2$ Error ↓ (mean ± std) | Cost×Err ↓ |
|---|---|---|---|
| baseline | 11.0 | 0.2374 ± 0.0389 | 2.62 |
| **ema_0.25** | **10.7** | **0.2369 ± 0.0386** | **2.54** |
| ema_0.45 | 11.3 | 0.2361 ± 0.0384 | 2.67 |
| alpha_0.05 | 11.0 | 0.2373 ± 0.0389 | 2.62 |
| alpha_0.12 | 11.0 | 0.2373 ± 0.0389 | 2.62 |
| alpha_0.20 | 11.0 | 0.2374 ± 0.0389 | 2.62 |
| gamma_1.5 | 11.0 | 0.2374 ± 0.0389 | 2.62 |
| gamma_2.0 | 11.0 | 0.2374 ± 0.0389 | 2.62 |
| gamma_2.5 | 11.0 | 0.2374 ± 0.0389 | 2.62 |
| gate_0.50 | 11.0 | 0.2374 ± 0.0389 | 2.62 |
| gate_0.70 | 11.0 | 0.2374 ± 0.0389 | 2.62 |
| calib_0.70_0.90 | 11.1 | 0.2371 ± 0.0387 | 2.63 |
| calib_0.80_0.98 | 10.9 | 0.2372 ± 0.0387 | 2.60 |
| adapt_1.5_0.75 | 12.8 | 0.2374 ± 0.0378 | 3.04 |
| adapt_2.0_0.85 | 11.2 | 0.2360 ± 0.0376 | 2.63 |
| no_ortho_filter | 11.0 | 0.2374 ± 0.0389 | 2.62 |
| damp_0.05 | 11.0 | 0.2373 ± 0.0389 | 2.62 |
| damp_0.10 | 11.0 | 0.2373 ± 0.0389 | 2.61 |
| damp_0.20 | 11.0 | 0.2370 ± 0.0389 | 2.60 |
| damp_0.30 | 11.0 | 0.2368 ± 0.0389 | 2.60 |

### 7.3 Curvature-aware sampler: hyperparameter robustness and Pareto dominance

We next evaluate the curvature-based adaptive controller under a broad sweep of hyperparameters while holding the model fixed. The controller uses (i) an EMA-based straightness/curvature proxy on the learned velocity field, (ii) a mild velocity regularization/blending step triggered by curvature, and (iii) adaptive step sizing. We vary EMA time scale, thresholds, gating, damping, and calibration parameters across 20 configurations.

**Observation: robust controller, automatic step allocation.** Across all 20 configurations, Avg. NFE remains in a tight range (10.7–12.8) and the mean relative error stays within $[0.2360, 0.2374]$ (sub-1% relative variation), indicating that the controller does not require fragile tuning. This directly contrasts with fixed-step Euler, where efficiency is achieved only at a tuned step budget and can degrade rapidly outside it.

### 7.4 Best fixed-step baseline vs. best adaptive configurations

For clarity, Table 4 compares the strongest fixed-step baseline (Euler-16) against the top adaptive configurations on ShearLayer2D OOD.

**Interpretation in the NFE–error plane.** These comparisons summarize a consistent pattern: the adaptive curvature-aware sampler lies on (or very near) the Pareto frontier in the NFE–error plane, while fixed-step baselines are strictly dominated in the regimes tested. This is the practical meaning of the curvature theory: the controller allocates steps only where the learned transport bends, while retaining large steps when the flow is straight.

Table 4: **Best fixed-step baseline vs. top adaptive configs** (ShearLayer2D, OOD macro–micro). At roughly one third fewer velocity evaluations, the adaptive sampler improves both error and Cost×Err.

| Method / Config | Avg. NFE | Rel. $L^2$ Err | Cost×Err | $\Delta$NFE vs Euler | $\Delta$Err vs Euler | $\Delta$(Cost×Err) |
|---|---|---|---|---|---|---|
| **Euler (16 steps)** | 16.0 | 0.2489 | 3.98 | 0% | 0% | 0% |
| **ema_0.25** | 10.7 | 0.2369 | 2.54 | $-33.1\%$ | $-4.8\%$ | $-36.2\%$ |
| calib_0.80_0.98 | 10.9 | 0.2372 | 2.60 | $-31.9\%$ | $-4.7\%$ | $-34.7\%$ |
| damp_0.30 | 11.0 | 0.2368 | 2.60 | $-31.3\%$ | $-4.9\%$ | $-34.7\%$ |

## 7.5 Distributional fidelity and diversity: ensemble $W_1$ under macro–micro evaluation

A common concern for deterministic probability-flow ODE sampling is potential under-dispersion or reduced diversity. Our evaluation is designed to test this directly in the conditional setting where diversity is well-defined.

**Empirical $W_1$ between conditional ensembles.** For each macro, let $\{\widehat{u}^{(j)}\}_{j=1}^{N_{\mathrm{micro}}}$ be model samples and $\{u^{(j)}\}_{j=1}^{N_{\mathrm{micro}}}$ be reference samples. We compute a one-point (spatially averaged) empirical $W_1$ between the corresponding micro-ensembles, and then average across macros. We also report per-variable $W_1$ (density, momenta, energy) and an aggregated score.

**Key empirical outcome.** Across benchmarks, our rectified-flow model matches or improves upon the diffusion baseline in ensemble $W_1$ while using substantially fewer function evaluations. In particular, the macro–micro protocol provides a direct quantitative check against mode collapse: matching $W_1$ at fixed macro conditions indicates that the conditional output distribution is captured rather than collapsed. (Complete per-variable $W_1$ tables and additional datasets are reported in the appendix.)

## 7.6 Practical takeaway: speed requires both straight transports and curvature-aware integration

The integrator study separates two effects: (i) the rectified-flow training objective yields a transport that can be traversed with far fewer evaluations than score-based diffusion, and (ii) within this ODE setting, curvature-aware control is a robust way to realize the efficiency in stiff multiscale regimes without per-problem solver tuning. Empirically, this is exactly what Tables 2–4 show: the controller achieves lower error at lower NFE than the strongest tuned fixed-step baseline, and its performance is stable across wide hyperparameter ranges.

**Appendix pointers.** We provide: (a) the same integrator/robustness tables on the second benchmark, (b) NFE–error scatter plots (Pareto frontiers), and (c) detailed per-variable $W_1$ breakdowns and additional OOD regimes.

# 8 Conclusion and Future Work

We have demonstrated that rectified flows provide an efficient surrogate for sampling multi-scale PDE solutions: on a suite of challenging incompressible and compressible problems, our method matches the statistical fidelity of conditional diffusion models while requiring an order of magnitude fewer solver steps—yielding up to 22× faster inference in practice. Across matched and shifted initial-condition regimes, rectified flows consistently reproduce energy spectra, vorticity distributions, and one-point Wasserstein distances within sampling error, confirming their robustness and multi-scale accuracy.

**Key Takeaways.** On account of its straightening bias, ReFlow requires significantly fewer ODE steps at inference-time (8-10) than its conditional-diffusion-based counterparts (more than 128) to achieve the same level of performance on our statistical metrics – consult **SM** M for more details. Furthermore, ReFlow is much more training efficient, requiring up to 120,000 iterations in reaching the same level of performance, across

all experiments, as our other conditional diffusion baselines (which typically take 500-600,000 iterations), see **SM** M.6. Our results further demonstrate that straightening effects remain evident even when projecting down to a 2D PCA subspace (see **SM** M.7).

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

# A    Theory appendix: conditional rectified flows for multiscale PDE surrogates

**Purpose and scope.**    This appendix provides the mathematical backbone for the theory statements made in the main text. The key points are the following.

1. Our model is a *conditional noise-to-solution rectified flow*: for a fixed input field $u_i$, it transports a simple reference sample $\xi$ to a sample from the conditional output law of the PDE.

2. The ideal conditional barycentric rectified velocity is *correct at the level of laws*: its flow maps the reference law exactly to the target conditional law.

3. Once the effective spatial resolution is fixed, the total conditional-law error splits into three distinct pieces:
$$\text{coverage} + \text{fit} + \text{discretization}.$$
Here coverage is an irreducible finite-resolution floor, fit measures approximation of the ideal conditional transport field by the learned neural field, and discretization measures the error incurred by numerically integrating the learned ODE.

4. The EMA/blending mechanism used in our sampler is *not* part of generic correctness. It is an additional stabilization device for the *OOD sampling regime*, where the learned velocity develops oscillatory errors along rectification time.

**How to read this appendix.**    Section B fixes the conditional law-level setting and clarifies what the experiments probe. Section C states the exact conditional transport result for the ideal barycentric rectified velocity and the $L^2$ projection property targeted by the training loss. Section D gives the continuous-time fit theorem for the learned conditional flow. Section E derives the discretization bounds and identifies the total derivative along trajectories as the relevant straightness quantity. Section F gives a separate OOD stabilization result for the EMA-blended sampler. Section G quantifies the finite-resolution coverage floor through structure functions and Fourier tails. Finally, Section H aggregates the pieces into the master law-level bound used in the main text, and Section I explains how this law-level viewpoint relates to the reported one-point metrics.

# B    Setup: conditional laws of PDE outputs and what the experiments measure

**State space and finite-dimensional surrogate.**    We work on the periodic domain $\mathcal{D} = \mathbb{T}^d$ and consider state fields taking values in
$$\mathcal{U} = L^2(\mathcal{D}; \mathbb{R}^m),$$
or, in the incompressible case, in the divergence-free subspace when appropriate. In the actual implementation all fields are represented on a finite-dimensional grid / Galerkin discretization
$$\mathbb{R}^{d_N} \subset \mathcal{U}.$$

All flow-map arguments below are rigorous in that finite-dimensional setting; the Hilbert-space notation is retained because it matches the law-level viewpoint of statistical solutions. In infinite-dimensional spaces, the exact-transport computation should be read as a weak continuity-equation identity unless an additional well-posedness theory for the associated flow is imposed.

**Conditional target laws.**    Let $C$ denote the conditioning variable, which in our applications is the input field $u_i$ (possibly together with boundary data or other known parameters). For a fixed realization $c$ of $C$, let
$$\mu_1^c := \text{Law}(U_f \mid C = c)$$
be the conditional law of the PDE output at the forecast horizon. Let
$$\pi_0^c$$
be a simple reference law on $\mathcal{U}$; in the experiments we take $\pi_0^c = \mathcal{N}(0, I)$, independent of $c$.

**Finite-resolution target law.** At an effective cutoff $K_c$, the best law a bandlimited surrogate can hope to match is not the full conditional law $\mu_1^c$ but its projected version

$$\bar{\mu}_1^c := (P_{\leq K_c})_{\#}\mu_1^c. \tag{56}$$

Accordingly, the exact conditional transport result below is stated for a generic target conditional law $\nu^c$, and in the finite-resolution master theorem we apply it with

$$\nu^c = \bar{\mu}_1^c.$$

**What the experiments probe.** Our empirical evaluation is conditional. At test time, a macro/input state $\widehat{u}_i$ is fixed, and both the model and the reference PDE solver are used to generate ensembles of outputs associated with that fixed macro condition. The reported statistics—mean error, spread error, and one-point Wasserstein distance—therefore probe whether the model reproduces the *conditional law* of outputs at fixed input, rather than only an unconditional average across inputs.

**Law-level versus marginal viewpoints.** From the abstract measure-theoretic perspective, a natural global metric on output laws is Wasserstein distance on $\mathcal{U}$, especially $W_2$ because it interacts cleanly with $L^2$-based structure functions and Fourier tails. The experiments report one-point $W_1$ averaged over space because it is computationally accessible and directly interpretable in terms of $k = 1$ correlation marginals. The mechanisms isolated below—coverage, fit, and discretization—are not specific to one particular metric; they explain why a learned transport that is accurate and easy to integrate can match the relevant output statistics with few function evaluations.

## C   Exact conditional transport under the barycentric rectified velocity

### C.1   Conditional couplings and rectified interpolation

Fix a condition $c$. Let

$$\pi_0^c, \nu^c \in \mathcal{P}(\mathcal{U})$$

be a source/reference conditional law and a target conditional law, and let

$$\gamma^c \in \mathcal{P}(\mathcal{U} \times \mathcal{U})$$

be any coupling of $(\pi_0^c, \nu^c)$. The practical training objective in the paper uses the product coupling because the source noise and target sample are drawn independently; the theoretical identity below is stated for a general prescribed coupling, with the resulting barycentric velocity depending on that choice. Write

$$(\Xi, Y) \sim \gamma^c,$$

where $\Xi$ is the source sample and $Y$ is the target sample.

For $\tau \in [0, 1]$, define the linear rectified interpolation

$$U_\tau := (1 - \tau)\Xi + \tau Y, \qquad \rho_\tau^c := \mathrm{Law}(U_\tau \mid C = c). \tag{57}$$

Thus $\rho_\tau^c$ is the conditional law of intermediate states obtained by moving linearly from the source sample $\Xi$ to the target sample $Y$.

### C.2   Conditional barycentric velocity

**Definition C.1** (Conditional barycentric rectified velocity)**.** For each fixed condition $c$ and each rectification time $\tau \in [0, 1]$, define

$$v^\star(u, \tau; c) := \mathbb{E}[Y - \Xi \mid U_\tau = u,\ C = c], \qquad \rho_\tau^c\text{-a.e. } u. \tag{58}$$

**Interpretation.** Along a single interpolation chord, the velocity is simply the constant vector $Y - \Xi$. At a population level, many such chords may pass through the same state $u$ at the same rectification time $\tau$. The field $v^\star(u, \tau; c)$ is the conditional version of the barycentric velocity used in standard rectified-flow and flow-matching arguments Liu et al. (2022): it averages the compatible chord directions, conditioned both on the current state and on the input variable $c$.

**Proposition C.2** (Exact conditional transport). *Assume we work on a finite-dimensional surrogate $\mathbb{R}^{d_N}$ of $\mathcal{U}$, and assume the continuity equation associated with $v^\star(\cdot, \tau; c)$ admits a unique classical flow map $X_\tau^c$ solving*

$$\frac{d}{d\tau} X_\tau^c(u) = v^\star(X_\tau^c(u), \tau; c), \qquad X_0^c(u) = u.$$

*Then*

$$(X_\tau^c)_\# \pi_0^c = \rho_\tau^c \qquad \text{for all } \tau \in [0, 1],$$

*and in particular*

$$(X_1^c)_\# \pi_0^c = \nu^c.$$

*Proof.* Let $(\Xi, Y) \sim \gamma^c$ and define $U_\tau = (1 - \tau)\Xi + \tau Y$, so that $\mathrm{Law}(U_\tau \mid C = c) = \rho_\tau^c$. Fix a smooth compactly supported test function $\varphi$ on $\mathbb{R}^{d_N}$. Differentiating under the conditional expectation gives

$$\frac{d}{d\tau} \mathbb{E}[\varphi(U_\tau) \mid C = c] = \mathbb{E}\big[\nabla\varphi(U_\tau) \cdot (Y - \Xi) \mid C = c\big].$$

Condition further on $U_\tau$:

$$\mathbb{E}\big[\nabla\varphi(U_\tau) \cdot (Y - \Xi) \mid C = c\big] = \mathbb{E}\big[\nabla\varphi(U_\tau) \cdot v^\star(U_\tau, \tau; c) \mid C = c\big].$$

Hence $\rho_\tau^c$ satisfies the weak continuity equation

$$\frac{d}{d\tau} \int \varphi(u)\, \rho_\tau^c(du) = \int \nabla\varphi(u) \cdot v^\star(u, \tau; c)\, \rho_\tau^c(du).$$

On the other hand, the pushforward curve $(X_\tau^c)_\# \pi_0^c$ satisfies the same weak equation by the method of characteristics, with the same initial condition at $\tau = 0$. By uniqueness in the finite-dimensional setting, the two curves coincide, so $(X_\tau^c)_\# \pi_0^c = \rho_\tau^c$ for all $\tau$. At $\tau = 1$, this yields $(X_1^c)_\# \pi_0^c = \nu^c$. $\qquad\square$

## C.3 The training loss as an $L^2$ projection

The practical rectified-flow loss is the conditional regression objective

$$\mathcal{L}(v) = \mathbb{E}\Big[\|v(U_\tau, \tau; C) - (Y - \Xi)\|^2\Big], \tag{59}$$

where the expectation is taken over the joint sampling of $C$, $(\Xi, Y) \sim \gamma^C$, and $\tau \sim \mathrm{Unif}[0, 1]$.

**Proposition C.3** (Population minimizer of the conditional rectified-flow loss). *For each fixed $\tau$ and condition $c$, the unique $L^2(\rho_\tau^c)$ minimizer of*

$$v \longmapsto \mathbb{E}\Big[\|v(U_\tau, \tau; c) - (Y - \Xi)\|^2 \,\big|\, C = c\Big]$$

*is the conditional barycentric field $v^\star(\cdot, \tau; c)$ from (58).*

*Proof.* This is the standard characterization of conditional expectation as the orthogonal projection in $L^2$. For fixed $c$ and $\tau$, write $Z = Y - \Xi$ and $X = U_\tau$. Then

$$v^\star(X, \tau; c) = \mathbb{E}[Z \mid X, \ C = c].$$

For any square-integrable measurable $v(X, \tau; c)$,

$$\mathbb{E}\big[\|Z - v(X, \tau; c)\|^2 \mid C = c\big] = \mathbb{E}\big[\|Z - \mathbb{E}[Z \mid X, C = c]\|^2 \mid C = c\big] + \mathbb{E}\big[\|\mathbb{E}[Z \mid X, C = c] - v(X, \tau; c)\|^2 \mid C = c\big],$$

which proves minimality and uniqueness. $\qquad\square$

Proposition C.2 is the law-level correctness statement: it is the standard rectified-flow continuity-equation argument applied conditionally. If the learned velocity matches the ideal conditional barycentric field for the relevant target law, then exact integration transports the source law to that target law in the finite-dimensional surrogate space. Proposition C.3 explains why the training loss is the natural conditional regression objective.

## D  Continuous-time fit error for the learned conditional flow

For the rest of the appendix, fix a condition $c$ and a target conditional law $\nu^c$ (in the finite-resolution application we will set $\nu^c = \bar{\mu}_1^c$). Let $v^\star(\cdot, \tau; c)$ be the corresponding conditional barycentric field.

Let $U_\tau$ solve the ideal conditional rectified ODE

$$\dot{U}_\tau = v^\star(U_\tau, \tau; c), \qquad U_0 \sim \pi_0^c. \tag{60}$$

By Proposition C.2,

$$\mathrm{Law}(U_1 \mid C = c) = \nu^c.$$

Let $\widehat{U}_\tau$ solve the learned continuous-time conditional ODE

$$\dot{\widehat{U}}_\tau = v_\theta(\widehat{U}_\tau, \tau; c), \qquad \widehat{U}_0 = U_0. \tag{61}$$

**Theorem D.1** (Continuous-time fit error). *Assume $v_\theta(\cdot, \tau; c)$ is $L$-Lipschitz in $u$, uniformly in $\tau \in [0,1]$. Then*

$$\left(\mathbb{E}\big[\|\widehat{U}_1 - U_1\|^2 \mid C = c\big]\right)^{1/2} \le e^L \int_0^1 \left(\mathbb{E}\big[\|v_\theta(U_\tau, \tau; c) - v^\star(U_\tau, \tau; c)\|^2 \mid C = c\big]\right)^{1/2} d\tau. \tag{62}$$

*In particular, if we define*

$$\varepsilon_{\mathrm{fit}}^2(c) := \int_0^1 \mathbb{E}\big[\|v_\theta(U_\tau, \tau; c) - v^\star(U_\tau, \tau; c)\|^2 \mid C = c\big]\, d\tau, \tag{63}$$

*then*

$$\left(\mathbb{E}\big[\|\widehat{U}_1 - U_1\|^2 \mid C = c\big]\right)^{1/2} \le e^L\, \varepsilon_{\mathrm{fit}}(c). \tag{64}$$

*Consequently,*

$$W_2\big(\nu^c, \widehat{\mu}_1^{c,\mathrm{cts}}\big) \le e^L\, \varepsilon_{\mathrm{fit}}(c), \qquad \widehat{\mu}_1^{c,\mathrm{cts}} := \mathrm{Law}(\widehat{U}_1 \mid C = c). \tag{65}$$

*Proof.* Set $E_\tau := \widehat{U}_\tau - U_\tau$. Then

$$\dot{E}_\tau = v_\theta(\widehat{U}_\tau, \tau; c) - v^\star(U_\tau, \tau; c) = \big(v_\theta(\widehat{U}_\tau, \tau; c) - v_\theta(U_\tau, \tau; c)\big) + \big(v_\theta(U_\tau, \tau; c) - v^\star(U_\tau, \tau; c)\big).$$

Hence

$$\|\dot{E}_\tau\| \le L\|E_\tau\| + \|v_\theta(U_\tau, \tau; c) - v^\star(U_\tau, \tau; c)\|.$$

Since $E_0 = 0$, the standard Grönwall stability argument for ODEs yields

$$\|E_1\| \le e^L \int_0^1 \|v_\theta(U_\tau, \tau; c) - v^\star(U_\tau, \tau; c)\|\, d\tau.$$

Take conditional $L^2$ norms, use Minkowski's integral inequality, and obtain (62). Equation (64) follows from Cauchy–Schwarz in $\tau$. Finally, (65) follows because the coupling given by the common initial sample $U_0 = \widehat{U}_0$ implies

$$W_2^2(\nu^c, \widehat{\mu}_1^{c,\mathrm{cts}}) \le \mathbb{E}\big[\|\widehat{U}_1 - U_1\|^2 \mid C = c\big].$$

$\square$

**Sanity check.** The continuous-time fit bound (64) vanishes when

$$v_\theta(\cdot, \tau; c) = v^\star(\cdot, \tau; c) \qquad \text{for a.e. } \tau,$$

which is the correct exactness property. This is precisely where the old fit+curvature theorem failed: a generic "straightness" term should *not* survive when the learned continuous-time field equals the ideal one.

## E Discretization: why straightness controls step count

We now fix the learned field $v_\theta(\cdot, \tau; c)$ and study only the numerical error incurred when integrating the learned ODE

$$\dot{\widehat{U}}_\tau = v_\theta(\widehat{U}_\tau, \tau; c).$$

### E.1 Local truncation error and the total derivative along trajectories

**Lemma E.1** (Local truncation error for explicit Euler). *Assume $v_\theta$ is $C^2$ in $(u, \tau)$ on the region traversed by the learned trajectory $\widehat{U}_\tau$, and define*

$$g(\tau) := v_\theta(\widehat{U}_\tau, \tau; c).$$

*Then one explicit-Euler step of size $\Delta\tau$ from time $\tau$ satisfies*

$$\begin{aligned}
\mathrm{LTE}(\tau; \Delta\tau) &:= \left\| \widehat{U}_{\tau+\Delta\tau} - \left( \widehat{U}_\tau + \Delta\tau \, v_\theta(\widehat{U}_\tau, \tau; c) \right) \right\| \\
&\leq \frac{(\Delta\tau)^2}{2} \left\| \partial_\tau v_\theta(\widehat{U}_\tau, \tau; c) + J_u v_\theta(\widehat{U}_\tau, \tau; c) \, v_\theta(\widehat{U}_\tau, \tau; c) \right\| + C \, (\Delta\tau)^3.
\end{aligned} \tag{66}$$

*If, in addition,*

$$\|\partial_\tau v_\theta(u, \tau; c)\| \leq M_\tau, \qquad \|J_u v_\theta(u, \tau; c)\| \leq L, \qquad \|v_\theta(u, \tau; c)\| \leq M$$

*on the relevant region, then*

$$\mathrm{LTE}(\tau; \Delta\tau) \leq \frac{(\Delta\tau)^2}{2}(M_\tau + LM) + \mathcal{O}((\Delta\tau)^3). \tag{67}$$

*Proof.* Differentiate the learned ODE along the trajectory:

$$\ddot{\widehat{U}}_\tau = \frac{d}{d\tau} v_\theta(\widehat{U}_\tau, \tau; c) = \partial_\tau v_\theta(\widehat{U}_\tau, \tau; c) + J_u v_\theta(\widehat{U}_\tau, \tau; c) \, \dot{\widehat{U}}_\tau.$$

Since $\dot{\widehat{U}}_\tau = v_\theta(\widehat{U}_\tau, \tau; c)$, this becomes

$$\ddot{\widehat{U}}_\tau = \partial_\tau v_\theta(\widehat{U}_\tau, \tau; c) + J_u v_\theta(\widehat{U}_\tau, \tau; c) \, v_\theta(\widehat{U}_\tau, \tau; c).$$

Now Taylor-expand $\widehat{U}_{\tau+\Delta\tau}$ around $\tau$:

$$\widehat{U}_{\tau+\Delta\tau} = \widehat{U}_\tau + \Delta\tau \, v_\theta(\widehat{U}_\tau, \tau; c) + \frac{(\Delta\tau)^2}{2} \ddot{\widehat{U}}_\tau + \mathcal{O}((\Delta\tau)^3).$$

Subtract the Euler update and take norms to obtain (66). The bound (67) follows from the triangle inequality. $\square$

**Interpretation.** Lemma E.1 identifies the relevant straightness quantity:

$$\frac{d}{d\tau} v_\theta(\widehat{U}_\tau, \tau; c) = \partial_\tau v_\theta(\widehat{U}_\tau, \tau; c) + J_u v_\theta(\widehat{U}_\tau, \tau; c) \, v_\theta(\widehat{U}_\tau, \tau; c). \tag{68}$$

This is the quantity that governs numerical step-count requirements. It is *not* merely the partial derivative $\partial_\tau v_\theta$ at fixed state.

### E.2 Global explicit-Euler error

**Theorem E.2** (Global Euler discretization error). *Assume the hypotheses of Lemma E.1, and assume moreover that $v_\theta(\cdot, \tau; c)$ is L-Lipschitz in u, uniformly in $\tau \in [0, 1]$. Let $\widehat{U}^{(N)}$ denote the explicit-Euler approximation of the learned flow with uniform step $\Delta\tau = 1/N$:*

$$\widehat{U}^{(N)}_{\tau_{n+1}} = \widehat{U}^{(N)}_{\tau_n} + \Delta\tau \, v_\theta(\widehat{U}^{(N)}_{\tau_n}, \tau_n; c), \qquad \tau_n = n/N.$$

*Then there exists a constant $C(L) > 0$ such that*

$$\sup_{0 \leq n \leq N} \left( \mathbb{E}\left[ \|\widehat{U}^{(N)}_{\tau_n} - \widehat{U}_{\tau_n}\|^2 \mid C = c \right] \right)^{1/2} \leq C(L) \, \Delta\tau \int_0^1 \left( \mathbb{E}\left[ \|\dot{g}(\tau)\|^2 \mid C = c \right] \right)^{1/2} d\tau. \tag{69}$$

*Equivalently, defining*

$$\varepsilon_{\text{disc}}(c; N) := \frac{1}{N} \int_0^1 \left( \mathbb{E}\left[ \|\dot{g}(\tau)\|^2 \mid C = c \right] \right)^{1/2} d\tau, \tag{70}$$

*we have*

$$\sup_{0 \leq n \leq N} \left( \mathbb{E}\left[ \|\widehat{U}^{(N)}_{\tau_n} - \widehat{U}_{\tau_n}\|^2 \mid C = c \right] \right)^{1/2} \leq C(L) \, \varepsilon_{\text{disc}}(c; N). \tag{71}$$

*Proof.* Let $e_n := \widehat{U}^{(N)}_{\tau_n} - \widehat{U}_{\tau_n}$. Standard one-step perturbation gives

$$\|e_{n+1}\| \leq (1 + L\Delta\tau)\|e_n\| + \text{LTE}(\tau_n; \Delta\tau).$$

Using the integral Taylor remainder along the learned trajectory,

$$\text{LTE}(\tau_n; \Delta\tau) \leq \Delta\tau \int_{\tau_n}^{\tau_{n+1}} \|\dot{g}(s)\| \, ds.$$

Iterating the recursion and using $(1 + L\Delta\tau)^N \leq e^L$ yields

$$\sup_{0 \leq n \leq N} \|e_n\| \leq C(L)\Delta\tau \int_0^1 \|\dot{g}(s)\| \, ds.$$

Taking conditional $L^2$ norms and applying Minkowski's inequality gives (69). $\square$

**Sanity check.** At fixed learned field, the discretization error (71) vanishes as $N \to \infty$. Thus, if the learned field is exact and we integrate it exactly, there is no residual discretization term. This is again where the old theory went wrong: straightness should influence *how many steps are needed*, not create an extra continuous-time approximation error.

## F  OOD stabilization by EMA blending

The EMA-based controller used in the sampler is not part of the generic correctness theorem. Its purpose is more specific: in the OOD macro–micro regime studied in the experiments, the learned velocity can become oscillatory along rectification time, which makes explicit integration brittle. The role of EMA/blending is to suppress this oscillatory component while preserving the slowly varying part of the ideal transport.

**Idealized continuous-time surrogate.** Fix a condition $c$ and consider the ideal path $U_\tau$ from (60). Define along this path

$$g(\tau) := v^\star(U_\tau, \tau; c), \tag{72}$$

and the learned-field error

$$\varepsilon(\tau) := v_\theta(U_\tau, \tau; c) - g(\tau). \tag{73}$$

Thus

$$v_\theta(U_\tau, \tau; c) = g(\tau) + \varepsilon(\tau).$$

### F.1 An abstract smoothing assumption

Let $E_\lambda$ be a causal smoothing operator in rectification time, with bandwidth parameter $\lambda > 0$. We assume the following two properties along the ideal path:

1. **Contraction on the oscillatory error component.** There exists $\sigma(c, \lambda) \in [0, 1)$ such that

$$\|E_\lambda \varepsilon\|_{L^2_\tau L^2_\Omega(c)} \leq \sigma(c, \lambda) \|\varepsilon\|_{L^2_\tau L^2_\Omega(c)}, \tag{74}$$

   where

$$\|f\|^2_{L^2_\tau L^2_\Omega(c)} := \int_0^1 \mathbb{E}\big[\|f(\tau)\|^2 \mid C = c\big] \, d\tau.$$

2. **Small bias on slowly varying signals.** There exists a universal constant $C_{\mathrm{ema}} > 0$ such that

$$\|E_\lambda g - g\|_{L^2_\tau L^2_\Omega(c)} \leq C_{\mathrm{ema}} \lambda \|\dot{g}\|_{L^2_\tau L^2_\Omega(c)}. \tag{75}$$

**Remark on exponential moving averages.** For the continuous-time exponential smoother defined by

$$m'(\tau) = \frac{1}{\lambda}\big(f(\tau) - m(\tau)\big), \qquad m(0) = f(0),$$

the bias estimate (75) holds with $C_{\mathrm{ema}} = 1$. Indeed, the error $e = m - f$ satisfies

$$e'(\tau) + \frac{1}{\lambda} e(\tau) = -f'(\tau), \qquad e(0) = 0,$$

so

$$e(\tau) = -\int_0^\tau e^{-(\tau - s)/\lambda} f'(s) \, ds,$$

and Young's inequality yields

$$\|e\|_{L^2(0,1)} \leq \lambda \|f'\|_{L^2(0,1)}.$$

A sufficient condition for (74) is that a nontrivial fraction of the error energy lies above the smoothing cutoff frequency; in the OOD regime this is precisely the structural behavior we empirically observe.

### F.2 Idealized extended-state blended flow

For a fixed blend weight $\alpha \in [0, 1]$, we analyze the continuous-time extended-state surrogate

$$\dot{\widetilde{U}}_\tau = (1 - \alpha) v_\theta(\widetilde{U}_\tau, \tau; c) + \alpha m_\tau, \tag{76}$$

$$\lambda \dot{m}_\tau = v_\theta(\widetilde{U}_\tau, \tau; c) - m_\tau, \qquad m_0 = v_\theta(\widetilde{U}_0, 0; c), \tag{77}$$

with $\widetilde{U}_0 = U_0$. This is the continuous analogue of the discrete EMA/blending sampler and should be viewed as a non-Markovian sampler on $\widetilde{U}$ or, equivalently, a Markovian system on the extended state $(\widetilde{U}, m)$.

Define

$$\varepsilon_{\mathrm{approx}}(c) := \|\varepsilon\|_{L^2_\tau L^2_\Omega(c)}. \tag{78}$$

If $\varepsilon_{\mathrm{approx}}(c) = 0$, the raw approximation error already vanishes; in that case we use the harmless convention $\sigma(c, \lambda) = 0$.

Define also

$$S_{*,\mathrm{tot}}(c) := \|\dot{g}\|_{L^2_\tau L^2_\Omega(c)}. \tag{79}$$

**Theorem F.1** (EMA stabilization in the oscillatory OOD regime)**.** *Assume the compared ODE dynamics satisfy the following stability estimate on the region traversed by the trajectories: if two trajectories start from the same initial condition and their velocity fields differ along the comparison path by a perturbation $r$, then*

$$\left(\mathbb{E}\big[\|\widetilde{U}_1 - U_1\|^2 \mid C = c\big]\right)^{1/2} \le C_{\text{stab}} \, \|r\|_{L^2_\tau L^2_\Omega(c)}. \tag{80}$$

*Assume also the smoothing estimates* (74)–(75)*. Then*

$$\left(\mathbb{E}\big[\|\widetilde{U}_1 - U_1\|^2 \mid C = c\big]\right)^{1/2} \le C_{\text{stab}}\Big[\big(1 - \alpha(1 - \sigma(c,\lambda))\big)\,\varepsilon_{\text{approx}}(c) + C_{\text{ema}}\,\alpha\,\lambda\,S_{*,\text{tot}}(c)\Big]. \tag{81}$$

*The corresponding unsmoothed bound is $C_{\text{stab}}\varepsilon_{\text{approx}}(c)$. Consequently, EMA blending yields a strictly smaller upper bound whenever*

$$(1 - \sigma(c,\lambda))\,\varepsilon_{\text{approx}}(c) > C_{\text{ema}}\,\lambda\,S_{*,\text{tot}}(c). \tag{82}$$

*Proof.* The stability estimate (80) reduces the terminal error to the $L^2_\tau L^2_\Omega(c)$ norm of the velocity perturbation along the ideal path. For the unsmoothed learned field this perturbation is $r_{\text{raw}} = \varepsilon$. For the EMA-blended field, along this path,

$$(1-\alpha)v_\theta(U_\tau,\tau;c) + \alpha E_\lambda[v_\theta(U.,\cdot;c)](\tau) - g(\tau) = (1-\alpha)\varepsilon(\tau) + \alpha E_\lambda\varepsilon(\tau) + \alpha(E_\lambda g(\tau) - g(\tau)).$$

By (74),

$$\|(1-\alpha)\varepsilon + \alpha E_\lambda\varepsilon\|_{L^2_\tau L^2_\Omega(c)} \le \big(1 - \alpha + \alpha\sigma(c,\lambda)\big)\varepsilon_{\text{approx}}(c).$$

By (75),

$$\|E_\lambda g - g\|_{L^2_\tau L^2_\Omega(c)} \le C_{\text{ema}}\lambda S_{*,\text{tot}}(c).$$

Combining the two estimates yields (81). The improvement criterion follows by comparing with the unsmoothed bound. □

**Interpretation.** The factor

$$(1 - \sigma(c,\lambda))\,\varepsilon_{\text{approx}}(c)$$

is the amount of oscillatory error removed by smoothing, while

$$\lambda\,S_{*,\text{tot}}(c)$$

is the bias paid for smoothing the *ideal* transport itself. Theorem F.1 therefore captures exactly the intended OOD mechanism: EMA helps when the learned error is sufficiently oscillatory, while the ideal conditional rectified transport remains slowly varying in rectification time.

**Relation to the practical sampler.** The actual algorithm uses a discrete-time EMA and an adaptive blend weight $\alpha_t$. The theorem above is a continuous-time fixed-$\alpha$ surrogate analysis of the same bias–variance tradeoff; it should be read as a structural explanation of the OOD sampler behavior rather than as an exact theorem for every implementation detail.

## G Coverage: structure functions control unresolved law tails

### G.1 Coverage term

At fixed spatial resolution, every surrogate operates under an effective cutoff $K_c$. Therefore, even a perfect model cannot represent energy beyond that cutoff. The best it can do is match the projected law

$$(P_{\le K_c})_{\#}\mu_1^c.$$

The discrepancy between the full conditional law and this projected conditional law is the *coverage term.*

## G.2 Structure functions and tail control

For $u : \mathbb{T}^d \to \mathbb{R}^m$, define the second-order structure function

$$S_r^2(u) := \frac{1}{|B_r|} \int_{|h| \leq r} \|u(\cdot + h) - u(\cdot)\|_{L^2(\mathbb{T}^d)}^2 \, dh, \qquad B_r := \{h \in \mathbb{R}^d : |h| \leq r\}. \tag{83}$$

For a law $\mu$ on $L^2(\mathbb{T}^d)$, set

$$S_r^2(\mu) := \int S_r^2(u) \, d\mu(u).$$

Assume that the target conditional law $\mu_1^c$ satisfies the modulus condition

$$S_r^2(\mu_1^c) \leq \omega_c(r) \qquad \text{for all } r \in (0, 1], \tag{84}$$

where $\omega_c(r) \downarrow 0$ as $r \downarrow 0$.

**Lemma G.1** (Structure function controls Fourier tail). *There exist constants $c_d, C_d > 0$ depending only on the dimension such that for every integer $K \geq 1$,*

$$\int \|P_{>K} u\|_{L^2}^2 \, d\mu_1^c(u) \leq C_d \, \omega_c(c_d/K). \tag{85}$$

*Proof.* For fixed $u$, expand

$$\|u(\cdot + h) - u(\cdot)\|_{L^2}^2 = \sum_{k \in \mathbb{Z}^d} |\hat{u}(k)|^2 \, |e^{ik \cdot h} - 1|^2.$$

Average over $|h| \leq r$ to obtain

$$S_r^2(u) = \sum_{k \in \mathbb{Z}^d} |\hat{u}(k)|^2 \Psi_r(k), \qquad \Psi_r(k) := \frac{1}{|B_r|} \int_{|h| \leq r} |e^{ik \cdot h} - 1|^2 \, dh.$$

A standard oscillation estimate gives

$$\Psi_r(k) \geq c \, \min\{(|k|r)^2, 1\}$$

for a dimensional constant $c > 0$. Hence $\Psi_r(k) \geq c_0$ for all $|k| \geq c'/r$, so

$$S_r^2(u) \geq c_0 \sum_{|k| \geq c'/r} |\hat{u}(k)|^2 = c_0 \|P_{>c'/r} u\|_{L^2}^2.$$

Integrate with respect to $\mu_1^c$ and choose $r = c_d/K$. $\qquad\square$

**Lemma G.2** (Projection estimate in $W_2$). *For any law $\mu$ on $L^2(\mathbb{T}^d)$ and cutoff $K_c \geq 1$,*

$$W_2\big(\mu, (P_{\leq K_c})_\# \mu\big) \leq \left( \int \|P_{>K_c} u\|_{L^2}^2 \, d\mu(u) \right)^{1/2}. \tag{86}$$

*Proof.* Couple $u \sim \mu$ with $\tilde{u} = P_{\leq K_c} u$. Then $\tilde{u} \sim (P_{\leq K_c})_\# \mu$ and

$$W_2^2\big(\mu, (P_{\leq K_c})_\# \mu\big) \leq \mathbb{E}\|u - \tilde{u}\|_{L^2}^2 = \mathbb{E}\|P_{>K_c} u\|_{L^2}^2.$$

$\qquad\square$

**Corollary G.3** (Coverage bound). *Under* (84),

$$W_2\big(\mu_1^c, \bar{\mu}_1^c\big) = W_2\big(\mu_1^c, (P_{\leq K_c})_\# \mu_1^c\big) \lesssim \omega_c(c_d/K_c)^{1/2}. \tag{87}$$

**Interpretation.** Corollary G.3 is the finite-resolution floor. Even a perfectly trained and exactly integrated surrogate cannot beat this term at cutoff $K_c$.

### G.3 Annulus lower bound and Besov interpretation

The following elementary inverse estimate is the only annulus lower bound used in the paper.

**Lemma G.4** (Annulus lower bound). *Let $f = P_{[K,2K]}u$ with $K \geq 1$. Then*

$$\|\nabla f\|_{L^2} \geq cK\|f\|_{L^2}.$$

*Proof.* By Plancherel,

$$\|\nabla f\|_{L^2}^2 = \sum_{K \leq |k| \leq 2K} |k|^2 |\hat{f}(k)|^2 \geq cK^2 \sum_{K \leq |k| \leq 2K} |\hat{f}(k)|^2 = cK^2 \|f\|_{L^2}^2.$$

$\square$

For $0 < s < 1$, the structure-function scaling also has the usual Besov interpretation. If

$$S_r^2(u) \leq Cr^{2s} \qquad 0 < r \leq 1,$$

then the Fourier proof of Lemma G.1, applied dyadically with $r \simeq 2^{-j}$, gives

$$\|\Delta_j u\|_{L^2} \lesssim C^{1/2} 2^{-js},$$

and hence $u \in B_{2,\infty}^s(\mathbb{T}^d)$. At the law level, the consequence used in the coverage estimates is the averaged Fourier-tail bound

$$S_r^2(\mu) \lesssim r^{2s} \implies \int \|P_{>K}u\|_{L^2}^2 \, d\mu(u) \lesssim K^{-2s}.$$

### G.4 One-step capacity–coverage–fit bound

Let $S_{\Delta t}$ denote the PDE solution operator over one physical step. Suppose a one-step RF surrogate returns $\widehat{u} = \mathcal{T}_{\Delta t}^\theta(u, \xi)$ and is trained against the resolved target $P_{\leq K_c} S_{\Delta t} u$ with

$$\varepsilon_{\text{train}}^2(t) := \mathbb{E}_{u \sim \mu_t} \mathbb{E}_\xi \|P_{\leq K_c} S_{\Delta t} u - \widehat{u}\|_{L^2}^2.$$

Then

$$W_2\big((S_{\Delta t})_\# \mu_t, \text{Law}(\widehat{u})\big) \leq C_d \omega(c_d/K_c)^{1/2} + \varepsilon_{\text{train}}(t), \tag{88}$$

where the structure-function modulus is applied to the target law $(S_{\Delta t})_\# \mu_t$.

*Proof.* Use the coupling given by the same input $u$ and noise seed $\xi$. Then

$$W_2\big((S_{\Delta t})_\# \mu_t, \text{Law}(\widehat{u})\big) \leq \Big(\mathbb{E}\|S_{\Delta t}u - \widehat{u}\|_{L^2}^2\Big)^{1/2}.$$

Insert $P_{\leq K_c} S_{\Delta t} u$ and apply Minkowski:

$$\Big(\mathbb{E}\|S_{\Delta t}u - \widehat{u}\|_{L^2}^2\Big)^{1/2} \leq \Big(\mathbb{E}\|P_{>K_c} S_{\Delta t}u\|_{L^2}^2\Big)^{1/2} + \varepsilon_{\text{train}}(t).$$

The first term is controlled by Lemma G.1. $\square$

# H  Master law-level bound: coverage + fit + discretization

We now combine the projected-target exact transport, the continuous-time fit error, and the discretization error.

Fix a condition $c$. Let

$$\bar{\mu}_1^c = (P_{\leq K_c})_{\#} \mu_1^c$$

be the projected target conditional law. Let $v_{\leq K_c}^{\star}(\cdot, \tau; c)$ be the conditional barycentric field associated with a coupling between $\pi_0^c$ and $\bar{\mu}_1^c$, and let $U_{\tau}^{\leq K_c}$ solve

$$\dot{U}_{\tau}^{\leq K_c} = v_{\leq K_c}^{\star}(U_{\tau}^{\leq K_c}, \tau; c), \qquad U_0^{\leq K_c} \sim \pi_0^c.$$

By Proposition C.2,

$$\mathrm{Law}(U_1^{\leq K_c} \mid C = c) = \bar{\mu}_1^c.$$

Let $\widehat{U}_{\tau}$ solve the learned continuous-time ODE

$$\dot{\widehat{U}}_{\tau} = v_{\theta}(\widehat{U}_{\tau}, \tau; c), \qquad \widehat{U}_0 = U_0^{\leq K_c},$$

and let $\widehat{U}^{(N)}$ be the explicit-Euler approximation of the same learned ODE with $N$ steps. Write

$$\widehat{\mu}_1^{c,\mathrm{cts}} := \mathrm{Law}(\widehat{U}_1 \mid C = c), \qquad \widehat{\mu}_1^{c,N} := \mathrm{Law}(\widehat{U}_1^{(N)} \mid C = c).$$

Define

$$\varepsilon_{\mathrm{fit},K_c}^2(c) := \int_0^1 \mathbb{E}\big[\|v_{\theta}(U_{\tau}^{\leq K_c}, \tau; c) - v_{\leq K_c}^{\star}(U_{\tau}^{\leq K_c}, \tau; c)\|^2 \mid C = c\big]\, d\tau, \tag{89}$$

and

$$\varepsilon_{\mathrm{disc}}(c; N) := \frac{1}{N} \int_0^1 \Big(\mathbb{E}\big[\|\partial_{\tau} v_{\theta}(\widehat{U}_{\tau}, \tau; c) + J_u v_{\theta}(\widehat{U}_{\tau}, \tau; c)\, v_{\theta}(\widehat{U}_{\tau}, \tau; c)\|^2 \mid C = c\big]\Big)^{1/2}\, d\tau. \tag{90}$$

**Theorem H.1** (Master conditional law-level bound). *Assume the hypotheses of Theorem D.1, Theorem E.2, and Corollary G.3. Then*

$$W_2\big(\mu_1^c, \widehat{\mu}_1^{c,N}\big) \lesssim \underbrace{\omega_c(c_d/K_c)^{1/2}}_{coverage} + \underbrace{e^L\, \varepsilon_{\mathrm{fit},K_c}(c)}_{fit} + \underbrace{C(L)\, \varepsilon_{\mathrm{disc}}(c; N)}_{discretization}. \tag{91}$$

*More explicitly,*

$$W_2\big(\mu_1^c, \widehat{\mu}_1^{c,N}\big) \leq W_2(\mu_1^c, \bar{\mu}_1^c) + W_2(\bar{\mu}_1^c, \widehat{\mu}_1^{c,\mathrm{cts}}) + W_2(\widehat{\mu}_1^{c,\mathrm{cts}}, \widehat{\mu}_1^{c,N})$$
$$\leq W_2(\mu_1^c, \bar{\mu}_1^c) + e^L\, \varepsilon_{\mathrm{fit},K_c}(c) + C(L)\, \varepsilon_{\mathrm{disc}}(c; N). \tag{92}$$

*Proof.* Insert the projected target law and the learned continuous-time law as intermediate terms:

$$W_2(\mu_1^c, \widehat{\mu}_1^{c,N}) \leq W_2(\mu_1^c, \bar{\mu}_1^c) + W_2(\bar{\mu}_1^c, \widehat{\mu}_1^{c,\mathrm{cts}}) + W_2(\widehat{\mu}_1^{c,\mathrm{cts}}, \widehat{\mu}_1^{c,N}).$$

The first term is the coverage term from Corollary G.3. The second term is the continuous-time fit term, bounded by Theorem D.1 applied to the ideal projected-target flow $U_{\tau}^{\leq K_c}$. The third term is the discretization term from Theorem E.2. Combining the three bounds gives (91). $\square$

**Sanity checks.** The bound (91) has the correct limiting behavior.

- If the learned field is exact on the projected target, i.e.

$$v_{\theta} = v_{\leq K_c}^{\star},$$

then the fit term vanishes.

- If the learned ODE is integrated exactly, i.e. $N \to \infty$, then the discretization term vanishes.

- Therefore, at fixed cutoff $K_c$, a perfect model leaves only the unavoidable coverage floor.

- If, in addition, the cutoff tends to infinity and the structure-function tail vanishes accordingly, then the total error tends to zero.

These are the basic plausibility requirements any correct theory must satisfy.

**What the master bound explains in the paper.** Theorem H.1 gives the error taxonomy that the main text should emphasize everywhere:

- **Coverage** is a hard floor imposed by unresolved small scales.

- **Fit** measures how well the learned neural velocity approximates the ideal projected-target conditional transport field.

- **Discretization** measures how hard the learned transport is to integrate numerically; this is the term improved by straightness.

The EMA theorem from Section F is an additional OOD stabilization result, not a fourth term in the generic master decomposition.

## I  From law-level bounds to the reported one-point metrics

**One-point Wasserstein as a marginal metric.** Let $\mu$ and $\widehat{\mu}$ be laws on fields $u : \mathcal{D} \to \mathbb{R}^m$. For each $x \in \mathcal{D}$, define the one-point marginals

$$\mu_x := (\mathrm{ev}_x)_\# \mu, \qquad \widehat{\mu}_x := (\mathrm{ev}_x)_\# \widehat{\mu}.$$

The reported metric in the main text is

$$W_1^{(1)}(\mu, \widehat{\mu}) := \int_{\mathcal{D}} W_1(\mu_x, \widehat{\mu}_x)\, dx.$$

By Kantorovich–Rubinstein duality, $W_1(\mu_x, \widehat{\mu}_x)$ controls the discrepancy of all pointwise 1-Lipschitz observables at the location $x$.

**Relation to the law-level theory.** The law-level bounds above are stated in $W_2$ on the full field space because this is the natural metric for coupling with $L^2$-based structure functions and Fourier tails. The one-point Wasserstein metric used in the experiments is weaker and more local, but it probes the same underlying conditional law. In particular:

- the *coverage* term controls whether unresolved high-frequency content can be represented at all;

- the *fit* term controls whether the learned continuous-time conditional flow reaches the correct projected target law;

- the *discretization* term controls whether a few ODE steps are sufficient to realize that law in practice.

These are exactly the mechanisms reflected empirically in the mean/spread/Wasserstein tables and in the spectral diagnostics.

**Emphasis of spectra during sampling.** The spectral evolution plots act as a scale-resolved diagnostic of the coverage and discretization story. They show how quickly the sampler reconstructs high-frequency energy up to the grid cutoff. This is why they are the right empirical counterpart of the theory above: they reveal both the finite-resolution ceiling and the fact that straighter rectified transports reach the attainable high-frequency regime with far fewer function evaluations than diffusion-based samplers.

## J   Architecture of the ReFlow Model

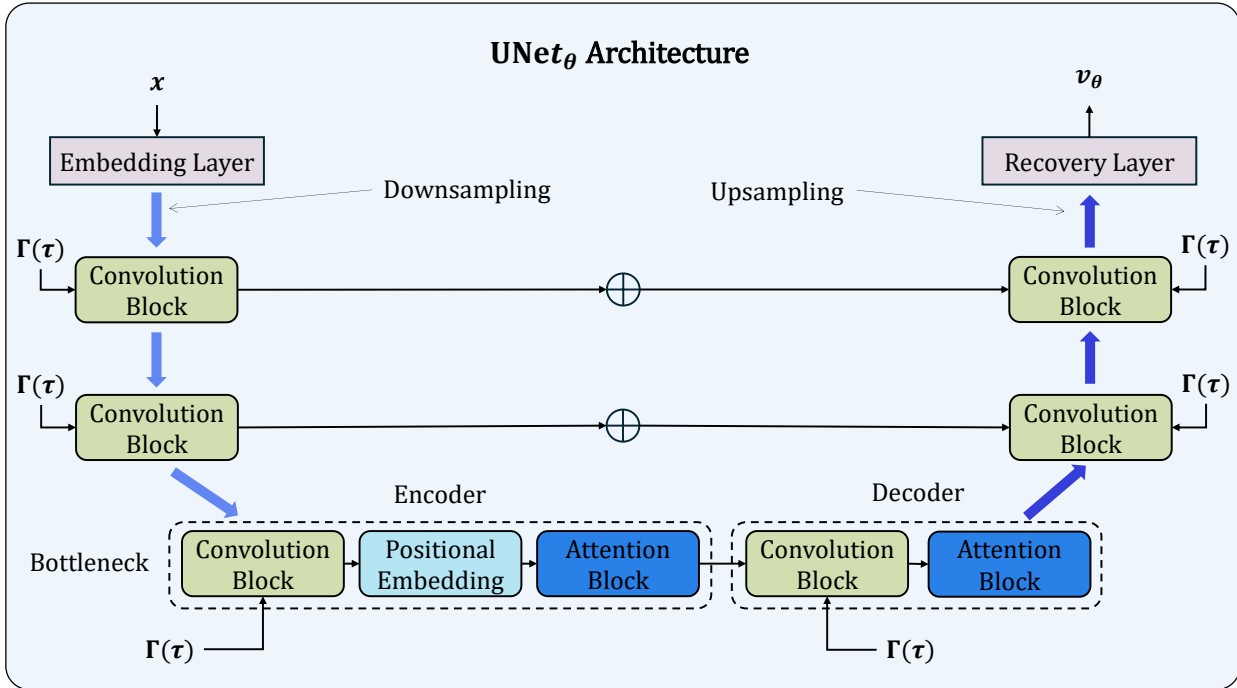

Figure 5: **UNet Backbone Architecture Used in ReFlow.** This schematic illustrates the core UNet-based architecture used within the ReFlow framework, structured across three resolution levels. For clarity, the number of blocks per level is set to one in this illustration. In the actual GenCFD and ReFlow configurations, each block is repeated four times per level. The bottleneck shows an asymmetry between the encoder and decoder sides: the block on the encoder side includes a Convolution Block, Positional Embedding, and Attention Block, while the corresponding block on the decoder side omits the Positional Embedding. The time embedding $\Gamma(\tau)$ is computed *once per sample* from the scalar rectified-flow time $\tau$ via a sinusoidal encoding followed by a shared MLP. Each convolution block then applies its own learned affine projection of this global embedding to produce the per-block scale and shift parameters used in the FiLM modulation (see §J.1). The embedding is therefore *globally shared* across levels, while its effect on features is *block-specific*.

### J.1   Architectural Components

**Motivation.** Multi-scale PDE data contain wide-ranging spatial scales, so a *multi-resolution* encoder–decoder is natural. We incorporate time embeddings to condition each layer on the interpolation fraction $\tau$.

**Architecture Layout.** We employ a UNet-based architecture augmented with attentional layers, typically termed as **UViT** by Saharia et al. (2022). In addition to the MLP-based time embedding $\Gamma(\tau)$, the input signal is lifted into a higher-dimensional embedding space and later projected back into physical space through convolutional layers. The data flows from high to low resolution across a down-sampling stack composed of convolution blocks into a bottleneck layer that encodes the fine-scale features of the inputs across its channel dimension and mixes them via a multi-head attention mechanism.

On the encoder side, exclusively at the bottleneck level, each attention block is preceded by a simple linear positional embedding. This positional encoding helps preserve spatial locality before global mixing through attention. The decoder side does not apply any positional embedding.

The UNet is conditioned at every level through normalized Feature-wise Linear Modulation (FiLM) techniques embedded within the convolution blocks. The conditioning signal modulates the normalization layers via learned scale and shift parameters. This conditioning mechanism is consistently applied across all en-

coder and decoder stages.

The output is symmetrically reconstructed via an up-sampling pipeline. Downsampling is performed using standard convolutional layers with stride, while upsampling can be implemented either via transposed convolutions or by applying a standard convolution followed by a non-learnable pixel shuffle operation, which rearranges elements of the tensor spatially according to a fixed upsampling factor.

Importantly, the architecture described here resembles the one presented in Molinaro et al. (2024), ensuring consistency with prior state-of-the-art practices in high-fidelity generative modeling for scientific computing.

### J.1.1 Convolution Block

Each UNet level employs dedicated convolution blocks that process features while enabling conditioning on the rectification time $\tau$. These blocks consist of two convolutional layers interleaved with group normalization and Swish activations. Between the convolutions, a normalized FiLM layer adaptively scales features using the conditioning embeddings. A residual connection combines the processed features with a projected skip connection, ensuring stable gradient flow. Crucially, these convolution blocks form the fundamental building units across all encoder and decoder stages, not just the bottleneck, with consistent application of temporal conditioning through the FiLM mechanism.

### J.1.2 Normalized FiLM Layer

Each convolution block uses a normalized FiLM layer and applies a convolution layer interleaved with normalization and an activation, then adds a residual skip:

$$\mathbf{z} \;=\; \mathrm{Conv}\big(\mathrm{Norm}(\mathbf{x})\big), \qquad \mathbf{z} \;=\; \sigma\big(\mathbf{z} + \alpha\,\gamma(\tau)\big),$$
$$\mathbf{x}_{\mathrm{out}} \;=\; \mathbf{x} \;+\; \mathrm{Conv}\big(\mathbf{z}\big), \tag{93}$$

where Norm is often `RMSNorm`, GroupNorm or LayerNorm, and $\sigma$ is a nonlinearity (e.g. SiLU). The scalar rectification time is embedded once as $\Gamma(\tau) = \mathrm{MLP}(\mathrm{PE}(\tau)) \in \mathbb{R}^{d_{\mathrm{emb}}}$, with sinusoidal positional encoding PE. Each block then applies its own affine map to $\Gamma(\tau)$, denoted here by $\gamma(\tau)$, to produce the scale/shift modulation used in that block. This temporal conditioning is crucial for guiding the generative process through rectified-flow trajectories.

### J.1.3 Attention Block

The architecture strategically employs multi-head attention exclusively at the bottleneck layer to balance global interaction modeling with computational efficiency. Each attention block processes normalized features through a spatial-channel reformatting, where features are reshaped by a flattening operation. This preserves the channel structure while collapsing the spatial grid into a single flattened dimension, allowing attention to operate across spatial locations. At certain scales, a multi-head attention (MHA) mechanism captures distant flow interactions. We define

$$\mathbf{Q} \;=\; W_Q\,\mathbf{h}, \quad \mathbf{K} \;=\; W_K\,\mathbf{h}, \quad \mathbf{V} \;=\; W_V\,\mathbf{h}, \quad \mathcal{A}(\mathbf{h}) \;=\; \mathrm{softmax}\Big(\tfrac{\mathbf{Q}\mathbf{K}^\top}{\sqrt{d}}\Big)\mathbf{V}, \tag{94}$$

where $\mathbf{h} \in \mathbb{R}^{(H \cdot W) \times d}$ denotes the flattened input sequence, with each of the $H \cdot W$ spatial positions represented as a $d$-dimensional feature vector. In practice, attention is split into multiple heads for better representational power, then recombined. This global mixing is key for PDE flows dominated by far-field couplings (e.g. vortex merging). The MHA operator is defined as

$$\mathrm{MHA}(\mathbf{h}) \;=\; W^O\,\mathrm{Concat}(\mathcal{A}_1(\mathbf{h}), \ldots, \mathcal{A}_n(\mathbf{h})), \tag{95}$$

where $n \in \mathbb{N}$ denotes the number of attention heads and $W^O \in \mathbb{R}^{C \times C}$ is a learned output projection matrix. The attention block applies MHA to a normalized input, then adds a residual connection to preserve the original signal and stabilize training:

$$\mathbf{z} \;=\; \mathrm{MHA}\big(\mathrm{Norm}(\mathbf{x})\big),$$
$$\mathbf{x}_{\mathrm{out}} \;=\; \mathbf{x} \;+\; \mathbf{z}. \tag{96}$$

### J.2 Training Pipeline Details

**Theory and implementation: conditional noise-to-solution interpolation.** Both the theory and the implementation use the same conditional noise-to-solution rectified-flow structure. The input field $u_0$ is the fixed conditioning variable and is *not* transported in rectification time. The transported state is initialized from noise and interpolated toward the target output. In the notation of Section 4.1 and Proposition 5.1, the theory uses the linear chord

$$x_\tau = (1 - \tau)\, \xi + \tau\, u_1, \qquad \xi \sim \mathcal{N}(0, I), \qquad \tau \in [0, 1],$$

with target velocity $u_1 - \xi$, while $u_0$ enters the network only as conditioning information. The implementation follows this same structure, with the optional scalar noise-schedule variant

$$\tilde{u}_\tau = \tau\, u_1 + \sigma(1 - \tau)\, \xi, \qquad \xi \sim \mathcal{N}(0, I),$$

so that the corresponding target direction is $u_1 - \sigma(1-\tau)\xi$ up to the chosen schedule convention. At inference we fix $u_0$, start from pure noise $\xi$, and integrate the learned conditional ODE to $\tau = 1$.

**Trainer Workflow.** A high-level `Trainer` class manages:

1. Batching PDE pairs $(u_0, u_1)$, normalized pixel-wise with statistics extracted over each entire dataset.

2. Sampling $\tau \in [0, 1]$ and random noise $\xi$ according to a given noise scheduler $\sigma(\cdot)$.

3. Partial interpolation and noise addition: $\tilde{u}_\tau := \tau\, u_1 + \sigma(1 - \tau)\, \xi$ (noise-conditioned form; see paragraph above). We have experimented with non-linear interpolation methods, but have not noticed meaningful oscillations in performance.

4. Minimizing the squared deviation between model output and target displacement/noise, optionally with a consistency/EMA update.

5. Throughout training, we monitor validation performance on a separate holdout dataset across all main metrics. Different validation splits have been tested to ensure there is no data leakage that would bias performance metrics.

Table 5: Representative hyperparameters for 2D fluid-flow tasks. See main text for justification of these choices.

| Parameter | Value (Optimal empirical performance) | Notes |
|---|---|---|
| Channels per level | (128, 256, 256) | Number of channels per level. |
| Downsample Ratios | (2, 2, 2) | Downsampling ratio per level. |
| Attention Heads | 8 | Each attention block. |
| Dropout Rate | 0.1 | In ConvBlock for generalization. |
| Noise Schedule | `uniform`, `log-uniform`, `cos-map` | Helps with multi-scale noise. |
| ODE Steps (Sampling) | 4–8 (NFE) | Often $> 15\times$ fewer vs. diffusion. |
| Batch Size | 16 | Tied to GPU capacity. |
| EMA Decay | 0.9999 | Teacher model for consistency. |
| Learning Rate | $3 \times 10^{-4}$ | Cosine decay in code. |

In practice we noticed that the most robust denoising schedule was the `uniform` one. For details on the `cos-map` schedule, we direct the reader to its description in Esser et al. (2024).

### J.3 Summary and Key Takeaways

The `RectifiedFlow` class, combined with a time-aware `UNet2D` architecture, builds a deterministic transport model well-suited to multi-scale PDE tasks. By interpolating between noise and data along nearly-straight paths, it reduces sampling overhead compared to conventional diffusion PDE solvers while retaining the flexibility to model complex turbulent phenomena. Further experimental and theoretical results appear in **SM** Section M. Instead of the multi-step *stochastic* integration in diffusion-based models, `RectifiedFlow` fits a *deterministic* ODE from noise to data. This leverages straighter trajectories in frequency space, enabling fewer network evaluations at inference time.

**ODE Integration for Sampling.** At inference:

$$\frac{d\tilde{u}_\tau}{d\tau} = v_\theta\big(\tilde{u}_\tau, u_0, \tau\big), \quad \tau \in [0,1], \quad \tilde{u}_{\tau=0} = \text{random noise.}$$

We solve this with `odeint`. As few as $8-10$ function evaluations (NFE) typically suffice, contrasting with the tens or hundreds of NFE required by full diffusion samplers. The corresponding wall-clock speedup depends on batch size and hardware and is reported separately in §M.6.

## K Datasets

### K.1 Multi-scale Flow Datasets

Following the dataset generation procedure of Herde et al. Herde et al. (2024), the Richtmyer–Meshkov ensemble is created by imposing a randomized sinusoidal perturbation on a two-fluid interface and driving it with a planar shock via prescribed pressure jumps. For the CloudShock and ShearFlow benchmarks, we follow the GenCFD mesh-generation and solver setup of Molinaro et al. Molinaro et al. (2024), initializing CloudShock with concentric density perturbations and ShearFlow with orthogonal shear jets on adaptive Cartesian grids, and employing the same high-order finite-volume scheme and dissipation settings to ensure consistent numerical fidelity across both datasets.

**Overview.** We experiment on three paired-field benchmarks $(u_0, u_1)$:

- **Richtmyer–Meshkov (RM)**: $64\,000$ samples

- **Cloud-Shock (CS)**: $40\,000$ samples

- **Shear-Layer (SL)**: $79\,200$ samples

All datasets track the $u_0 \to S^t(u_0)|_{t=1}$ solution mapping from inputs to evolved outputs at $t = 1$. The first two datasets monitor the evolution of density ($\rho$), momentum components ($m_x, m_y$), and energy ($E$). The shear-flow dataset monitors the velocity components ($u_x, u_y$).

**Train/Validation Split.** For each dataset, we reserve 80% of the samples for training and 20% for validation.

**Macro–Micro Ensemble Evaluation (CS & SL).** To probe out-of-distribution generalization under small perturbations, we adopt a two-stage ensemble protocol:

1. *Macro-sampling:* select $M_{\mathrm{macro}} = 10$ distinct base initial conditions.

2. *Micro-perturbation:* for each base, generate $M_{\mathrm{micro}} = 1{,}000$ perturbed copies within a small radius.

3. *Metrics:* compute all performance measures (mean-field error $e_\mu$, standard-deviation error $e_\sigma$, average $W_1$, spectral agreement) by first averaging over each micro-ensemble, then averaging across the $M_{\mathrm{macro}}$ cases.

**In-Distribution Evaluation (RM).** For the Richtmyer–Meshkov dataset, we additionally evaluate on a held-out test set drawn from the same distribution (no macro–micro perturbations). This allows us to study how deterministic and probabilistic models scale with training set size under standard, in-distribution conditions.

### K.2 Preprocessing and Data Organization

**Data Processing**

Before training, we compute and store global, channel-wise statistics over the entire training dataset. Concretely, for each physical variable $c$ we calculate

$$\mu_c = \frac{1}{N_{\mathrm{train}}\, H\, W\, [D]} \sum_{i=1}^{N_{\mathrm{train}}} \sum_{x,y,[z]} u_c^{(i)}\big(x, y, [z], t_0\big),$$

$$\sigma_c = \sqrt{\frac{1}{N_{\mathrm{train}}\, H\, W\, [D]} \sum_{i=1}^{N_{\mathrm{train}}} \sum_{x,y,[z]} \Big(u_c^{(i)}\big(x, y, [z], t_0\big) - \mu_c\Big)^2},$$

where $N_{\mathrm{train}}$ is the number of training samples, $H \times W\, [\times D]$ the spatial grid size, and $u_c(x, y, [z], t_0)$ the initial field for channel $c$. We average over all spatial locations to yield a single mean $\mu_c$ and standard deviation $\sigma_c$ per channel. The same statistics are also computed over the final-time fields $u_c(t_f)$, producing $\mu'_c$ and $\sigma'_c$ for the output channels.

During both training and evaluation, each input field $u_c$ is standardized via

$$\tilde{u}_c = \frac{u_c - \mu_c}{\sigma_c + 10^{-16}}, \qquad \tilde{v}_c = \frac{v_c - \mu'_c}{\sigma'_c + 10^{-16}},$$

where $u_c$ and $v_c$ denote the initial and target fields, respectively. We then concatenate $\{\tilde{u}_c\}$ and $\{\tilde{v}_c\}$ along the channel axis before feeding them to the network. Because we always apply the same pre-computed statistics at test time, no information from the evaluation set ever influences these normalization parameters.

**Test-time ensemble perturbation (evaluation only).** We adopt the same testing setup proposed in *GenCFD* Molinaro et al. (2024). To evaluate how well our model captures the *distribution* of solutions $u(t)$ arising from a fixed, chaotic initial field $\bar{u}$, we generate a small perturbed ensemble around $\bar{u}$ and evolve each member with a high-fidelity PDE solver—*but only at evaluation time.* Specifically:

1. **Micro-ensemble generation.** Around the test initial condition $\bar{u}$, draw $M_{\text{micro}}$ perturbed copies $\{\bar{u}_j\}_{j=1}^{M_{\text{micro}}}$ by sampling uniformly in a ball of radius $\varepsilon$ centered at $\bar{u}$.

2. **Reference propagation.** Integrate each perturbed field $\bar{u}_j$ forward to time $t$ using the reference solver, producing end-states $\{u_j(t)\}$.

3. **Empirical conditional law.** Form the empirical measure

$$\hat{P}_t(\,\cdot\mid\bar{u})\;=\;\frac{1}{M_{\text{micro}}}\sum_{j=1}^{M_{\text{micro}}}\delta_{u_j(t)},$$

which approximates the true chaotic conditional law $P_t(\,\cdot\mid\bar{u})$.

When the input distribution itself is non-degenerate, one would first draw $M_{\text{macro}}$ base fields from that distribution and then apply the micro-ensemble procedure to each. For our Dirac initial-condition tests we set $M_{\text{macro}} = 1$.

**Training vs. evaluation.** During training the model only ever sees independent pairs $(u_0, u_1)$—one target per input—and never observes any ensemble. The above ensemble perturbation is used *only* to construct a ground-truth distribution at test time.

## L   Models and Baselines

We tested our method against four other baselines, of which two (GenCFD and its FNO-conditioned variant) are diffusion-based algorithms of the same kind. The other two rely on traditional operator learning, with the UViT baseline also serving as an ablation that quantifies the gain from adding diffusion.

**Shared Backbone.** All UNet-based models (ReFlow, GenCFD, GenCFD ∘ FNO, and UViT) use the *identical* UViT backbone specified in Table 5: channels $(128, 256, 256)$, 3 resolution levels, 4 blocks per level, 8 attention heads — approximately **5M trainable parameters** in every case. The architecture is illustrated schematically in Figure 5.

### GenCFD

GenCFD Molinaro et al. (2024) is an end-to-end conditional score-based diffusion model that directly learns the mapping from an input flow field $u_0$ to the target field $u_1$. It uses the same UViT backbone described in Table 5, with noise levels $\sigma_\tau$ corresponding to the standard deviation of the added Gaussian perturbation, in contrast to our ReFlow model which evolves over a rectified diffusion time $\tau \in [0, 1]$. Common choices for noise schedulers include exponential and tangent formulations. In our case, we consistently adopted an exponential noise schedule, as it yielded the best results across all datasets and metrics reported. In all GenCFD models, we opted for a variance-exploding (VE) formulation throughout. During training we minimize the denoising score-matching loss over $(u_0, u_1)$ pairs sampled uniformly in $\tau \in [0, 1]$, using Adam with initial learning rate $3 \times 10^{-4}$ (cosine decay), batch size 16, and EMA decay 0.9999. We applied denoiser preconditioning and used uniform weighting across noise levels. At inference we solve the corresponding SDE via an Euler–Maruyama scheme with 128 discretization steps, and incorporate reconstruction guidance to condition on $u_0$ so that we recover the conditional posterior measure $p(u_1|u_0)$.

### GenCFD ∘ FNO

In this hybrid approach, a Fourier Neural Operator (FNO) Li et al. (2020a) is first trained under an $\ell_2$ loss to predict $u_1$ from $u_0$. Its output $\hat{u}_1^{\text{FNO}}$ is concatenated channel-wise with $u_0$ and fed into the GenCFD score

network at both train and test time. All other architectural and optimization settings (backbone, scheduler, SDE sampler) remain identical to GenCFD, allowing the FNO to provide fast low-frequency priors while the diffusion stage refines fine-scale turbulent features.

**Custom implementation.** This schematic closely parallels the pipeline of Oommen et al. (2024), but with two key distinctions. First, we fully replace both the diffusion backbone and its denoising stages with GenCFD. Second, and most importantly, we condition not only on the FNO's coarse solution but also on the original high-resolution initial condition, ensuring that fine-scale structures are not lost.

### FNO

Our FNO baseline follows Li *et al.* Li et al. (2020a) and is implemented in PyTorch. Each spatial location of the input field $u_0$ is lifted from $C_{\text{in}}$ to 256 channels via a two-layer MLP with a SiLU activation, then projected down to a 128-channel hidden representation. This is followed by four `FnoResBlock`s: in each block, the hidden features undergo a Conditional LayerNorm (time embedding size 128), SiLU, and two spectral convolutions (`SpectralConv`) with $num\_modes = (24, 24)$, combined through a learnable "soft-gate" residual skip. Finally, a two-layer projection MLP ($128 \rightarrow 256 \rightarrow C_{\text{out}}$, with SiLU in between) produces the output field $\hat{u}_1$. We train for 100k iterations minimizing $\|u_1 - \hat{u}_1\|_2^2$ with Adam (learning rate $10^{-3}$, batch size 16), and perform inference in one forward pass.

### UViT

Our UViT surrogate Saharia et al. (2022) uses the identical UNet2D backbone of Table 5, injecting time embeddings and employing 8-head multi-head attention at the bottleneck. It is trained deterministically with MSE loss on $(u_0, u_1)$, using Adam at $3 \times 10^{-4}$, batch size 16, for 100k iterations (EMA 0.9999). For multi-step forecasting we roll out autoregressively. This ablates the diffusion component while matching all other design choices.

### Baseline Model Summary

Table 6 summarizes key architectural settings for all models. These configurations were selected to reflect the strongest performance achievable within a shared training budget of up to 120,000 iterations, as used in the quantitative comparison reported in Table 1.

**Padding and boundary conditions.** Each dataset required padding choices tailored to its physical boundary conditions. SL and RM, which exhibit periodic boundaries, use circular padding in all convolutional layers. In contrast, CS features non-periodic boundaries and is trained with standard zero padding. These choices apply to all UNet-based architectures except ReFlow, which uniformly uses zero padding across all datasets.

Table 6: Summary of architectural depth and parameter count for each model. All UNet-based models share a consistent backbone design — channels $(128, 256, 256)$, 3 levels, 4 blocks/level, 8 attention heads — yielding $\approx$5M parameters each. The FNO baseline uses a structurally different architecture.

| Model | # Levels | # Blocks | # Params | Noise Schedule | Diffusion Scheme |
|-------|----------|----------|----------|----------------|------------------|
| ReFlow | 3 | 4 | 5M | uniform | rectified continuous time variable $\tau \in [0, 1]$ |
| GenCFD | 3 | 4 | 5M | exponential | variance-exploding |
| GenCFD ∘ FNO | 3 | 4 | 5M | exponential | variance-exploding |
| UViT | 3 | 4 | 5M | — | — |
| FNO | — | 4 | 11M | — | — |

# M Results

## M.1 Performance Metrics

To compare our generated ensembles against the reference Monte Carlo samples, we employ three complementary measures:

- **Mean-field error.** Let $\mu_{\text{ref}}(x)$ and $\mu_{\text{model}}(x)$ denote the pointwise spatial means of the reference and model ensembles, respectively. We measure their discrepancy by the $L^2$-norm

$$e_\mu \;=\; \big\| \mu_{\text{ref}} \;-\; \mu_{\text{model}} \big\|_2.$$

- **Standard-deviation error.** Similarly, let $\sigma_{\text{ref}}(x)$ and $\sigma_{\text{model}}(x)$ be the pointwise standard deviations:

$$e_\sigma \;=\; \big\| \sigma_{\text{ref}} \;-\; \sigma_{\text{model}} \big\|_2.$$

  Both metrics are reported after normalization by the $L^2$ norm of the ground truth.

- **Average 1-Wasserstein distance.** For a fixed initial condition $u_0$, let $\{u_{\text{ref}}^{(j)}(x)\}_{j=1}^N$ and $\{u_{\text{model}}^{(j)}(x)\}_{j=1}^N$ be the ensembles of final-time values. Since these fields are discretized on $M$ spatial points $\{x_i\}_{i=1}^M$, we obtain two empirical 1D distributions at each $x_i$. Denoting their inverse CDFs by $F_{\text{ref}}^{-1}(\cdot\,;x_i)$ and $F_{\text{model}}^{-1}(\cdot\,;x_i)$, the pointwise Wasserstein-1 distance is

$$W_1\big(p_{\text{ref}}(\cdot \mid x_i),\, p_{\text{model}}(\cdot \mid x_i)\big) \;=\; \int_0^1 \big| F_{\text{ref}}^{-1}(q;x_i) \;-\; F_{\text{model}}^{-1}(q;x_i)\big|\, \mathrm{d}q,$$

  and we report the spatial average $\overline{W}_1 = \frac{1}{M}\sum_{i=1}^M W_1\big(p_{\text{ref}}(\cdot \mid x_i),\, p_{\text{model}}(\cdot \mid x_i)\big)$.

**Energy spectra.** To assess how well different scales are captured, we compute the discrete energy spectrum of each velocity field. Given a sample $u(x)$ on a $d$-dimensional periodic grid with spacing $\Delta$, let $\widehat{u}_k$ be its Fourier coefficient at integer wavevector $k \in \mathbb{Z}^d$. We bin modes by their $\ell_1$ radius $|k|_1 = k_1 + \cdots + k_d$, defining

$$E_r \;=\; \frac{\Delta^d}{2} \sum_{\substack{k \in \mathbb{Z}^d \\ |k|_1 = r}} \big\|\widehat{u}_k\big\|^2, \quad r = 0, 1, 2, \ldots$$

and compare the ensemble-average spectra of reference and model on log–log axes.

## M.2 Metric Variability Across Sampling Seeds

To quantify sampling variance, we repeated each evaluation with five independent random-noise seeds. Table 7 reports mean $\pm$ std. The relative standard deviation is below 0.5% for all $e_\mu$ and $e_\sigma$ entries, and below 0.1% for all $\overline{W}_1$ entries, confirming that the results in Table 1 are highly stable with respect to stochastic sampling.

## M.3 Noise Schedule Sensitivity

We evaluated nine inference-time noise schedules — `uniform`, `log-uniform`, `cos-map`, `exponential`, `tangent`, `quadratic`, `sqrt`, `sigmoid`, and `linear-beta` — on both ShearLayer2D and CloudShock2D. The results are shown in Table 8.

The learned rectified velocity field produces trajectories sufficiently straight that the choice of inference-time schedule has no measurable effect on sample quality. This invariance is a direct consequence of the straightening bias: once the velocity field is nearly straight, the time-discretization grid is irrelevant for integration accuracy. This contrasts with score-based diffusion, where schedule choice critically affects performance because the learned score is highly curved near $\tau = 0$.

Table 7: ReFlow metric variability across 5 independent sampling seeds (mean $\pm$ std).

| Dataset | Variable | $e_\mu$ | $e_\sigma$ | $\overline{W}_1$ |
|---------|----------|---------|------------|------------------|
| ShearLayer2D | $u_x$ | $0.0340 \pm 0.0001$ | $0.0716 \pm 0.0002$ | $(1.76 \pm 0.003) \times 10^{-3}$ |
| | $u_y$ | $0.1893 \pm 0.0010$ | $0.0776 \pm 0.0003$ | $(1.41 \pm 0.001) \times 10^{-3}$ |
| CloudShock2D | $\rho$ | $0.0468 \pm 0.0001$ | $0.0999 \pm 0.0002$ | $(1.14 \pm 0.001) \times 10^{-3}$ |
| | $m_x$ | $0.0317 \pm 0.0001$ | $0.0717 \pm 0.0002$ | $(1.23 \pm 0.001) \times 10^{-3}$ |
| | $m_y$ | $0.0392 \pm 0.0001$ | $0.0812 \pm 0.0002$ | $(1.39 \pm 0.001) \times 10^{-3}$ |
| | $E$ | $0.0138 \pm 0.00002$ | $0.0576 \pm 0.0001$ | $(1.50 \pm 0.001) \times 10^{-3}$ |

Table 8: ReFlow performance across 9 inference-time noise schedules. All 9 schedules produce identical metrics; values shown are the common result.

| Dataset | Variable | $e_\mu$ | $e_\sigma$ | $\overline{W}_1$ |
|---------|----------|---------|------------|------------------|
| ShearLayer2D | $u_x$ | 0.1153 | 0.3674 | 0.0022 |
| | $u_y$ | 0.2346 | 0.3820 | 0.0021 |
| CloudShock2D | $\rho$ | 0.1525 | 0.2320 | 0.0011 |
| | $m_x$ | 0.1083 | 0.2034 | 0.0012 |
| | $m_y$ | 0.1039 | 0.2077 | 0.0015 |
| | $E$ | 0.0452 | 0.1852 | 0.0015 |

## M.4 ODE Integrator Comparison

Tables 9 and 10 compare fixed-step Euler, second- and fourth-order Runge–Kutta (RK2, RK4), and the adaptive curvature-aware sampler on ShearLayer2D and CloudShock2D respectively. Times are measured on a single NVIDIA Quadro RTX 6000.

**Why higher-order integrators do not outperform Euler.** We are not integrating an exact physical ODE with a known vector field; we are integrating a *learned* generative ODE whose velocity field $v_\theta$ carries finite approximation error. A higher-order solver more faithfully follows local imperfections of the learned field, which can move samples farther from the data manifold. By contrast, the first-order Euler method has a mild regularizing effect that tends to keep trajectories in high-density regions. The practical gain therefore comes not from solver order but from the *adaptive curvature-aware sampler*, which selectively refines steps where the learned velocity curves — achieving the lowest absolute error at the fewest NFE.

Table 9: ODE integrator comparison on **ShearLayer2D** (OOD). Error = mean relative $L^2$ over $u_x, u_y$; Cost$\times$Err = NFE $\times$ Error.

| Method | Steps | NFE | Error $\pm$ Std | Cost$\times$Err | Time (ms) |
|--------|-------|-----|-----------------|-----------------|-----------|
| Euler | 8 | 8 | $0.2359 \pm 0.053$ | 1.89 | 161 |
| Euler | 16 | 16 | $0.2475 \pm 0.055$ | 3.96 | 355 |
| Euler | 32 | 32 | $0.2541 \pm 0.056$ | 8.13 | 745 |
| RK2 | 8 | 16 | $0.2603 \pm 0.057$ | 4.16 | 355 |
| RK4 | 8 | 32 | $0.2621 \pm 0.058$ | 8.39 | 745 |
| RK4 | 32 | 128 | $0.2616 \pm 0.058$ | 33.49 | 3083 |
| Adaptive-EMA | 10 | 10 | $0.2090 \pm 0.041$ | 2.09 | 448 |
| Adaptive-EMA | 13 | 13 | **0.1762** | **2.29** | 524 |

Across both datasets, RK4 at 32 NFE is strictly worse than Euler at 32 NFE. The adaptive sampler achieves the lowest absolute error at 10–13 NFE, well below the budget of any fixed-step baseline. This is because error

Table 10: ODE integrator comparison on **CloudShock2D** (OOD). Same metrics as Table 9.

| Method | Steps | NFE | Error $\pm$ Std | Cost$\times$Err | Time (ms) |
|---|---|---|---|---|---|
| Euler | 8 | 8 | $0.2211 \pm 0.021$ | 1.77 | 150 |
| Euler | 16 | 16 | $0.2261 \pm 0.021$ | 3.62 | 330 |
| Euler | 32 | 32 | $0.2289 \pm 0.021$ | 7.32 | 694 |
| RK2 | 8 | 16 | $0.2319 \pm 0.021$ | 3.71 | 331 |
| RK4 | 8 | 32 | $0.2323 \pm 0.021$ | 7.43 | 693 |
| RK4 | 32 | 128 | $0.2322 \pm 0.021$ | 29.72 | 2873 |
| Adaptive-EMA | 11 | 11 | $0.2259 \pm 0.017$ | 2.48 | 440 |
| Adaptive-EMA | 13 | 13 | $\mathbf{0.2121 \pm 0.016}$ | **2.76** | 486 |

saturates quickly with NFE once the velocity field is sufficiently straight; additional steps in high-straightness regions yield no further benefit.

## M.5   Relation to ODE-Based Diffusion Samplers

A natural question is whether ReFlow's efficiency can be matched simply by applying a probability-flow ODE (PF-ODE) sampler — such as DPM-Solver or the Heun-based solver of Karras et al. — to the existing GenCFD score model. We address this directly.

PF-ODE methods accelerate sampling of a *fixed* learned score model; they do not change the learned dynamics. Our method trains a *different* conditional transport field $v_\theta$, whose objective promotes straighter trajectories in rectified time, yielding a smoother, less stiff field that can be integrated accurately in very few steps. The efficiency gain comes from *learning a straighter field*, not from choosing an ODE solver.

To verify this, we re-evaluated GenCFD using the PF-ODE with a second-order Heun solver at $4, 8, 16, 32, 64$, and 128 NFE. On ShearLayer2D at 128 NFE, PF-ODE was worse than the standard SDE sampler on both $e_\mu$ and $e_\sigma$, confirming that the stiffness of the score field — not the choice of solver — is the bottleneck for GenCFD. The rectified-flow velocity field learned by ReFlow is intrinsically less stiff, which is why it integrates accurately in 8–10 steps regardless of solver choice.

## M.6   Training Efficiency Observations

To assess training efficiency, we compare ReFlow with GenCFD using a matched parameter count ($\approx$5M) and three-level UViT architecture. Both models were trained for up to 160,000 iterations with a batch size of 16.

Figure 6 summarizes performance over the training trajectory. At approximately 100,000 steps, ReFlow consistently outperforms GenCFD across all key metrics: $e_\mu$, $e_\sigma$, and $\overline{W}_1$ for both $u_x$ and $u_y$. The $y$-axis uses a logarithmic scale to emphasize differences in error magnitude.

To further probe the long-term capabilities of GenCFD, we extended its training to 600,000 iterations. Table 11 summarizes the earliest step at which GenCFD overtakes ReFlow's performance at 150,000 steps. While GenCFD catches up in several cases, it never surpasses ReFlow on $\overline{W}_1$ for $u_y$ within the training horizon considered.

Table 11: Number of training iterations at which GenCFD surpasses ReFlow's performance at 150,000 iterations. "—" indicates GenCFD never surpasses ReFlow within the training horizon.

| Metric | GenCFD Iteration | GenCFD Value | ReFlow @ 150k |
|---|---|---|---|
| $u_x$ Mean Error $(e_\mu)$ | 280,000 | $3.33 \times 10^{-2}$ | $3.45 \times 10^{-2}$ |
| $u_y$ Mean Error $(e_\mu)$ | 400,000 | $1.70 \times 10^{-1}$ | $1.88 \times 10^{-1}$ |
| $u_x$ Std Error $(e_\sigma)$ | 400,000 | $6.48 \times 10^{-2}$ | $7.24 \times 10^{-2}$ |
| $u_y$ Std Error $(e_\sigma)$ | 340,000 | $7.91 \times 10^{-2}$ | $7.97 \times 10^{-2}$ |
| $u_x$ Wasserstein-1 $(W_1)$ | 200,000 | $3.89 \times 10^{-2}$ | $3.95 \times 10^{-2}$ |
| $u_y$ Wasserstein-1 $(W_1)$ | — | — | $4.11 \times 10^{-2}$ |

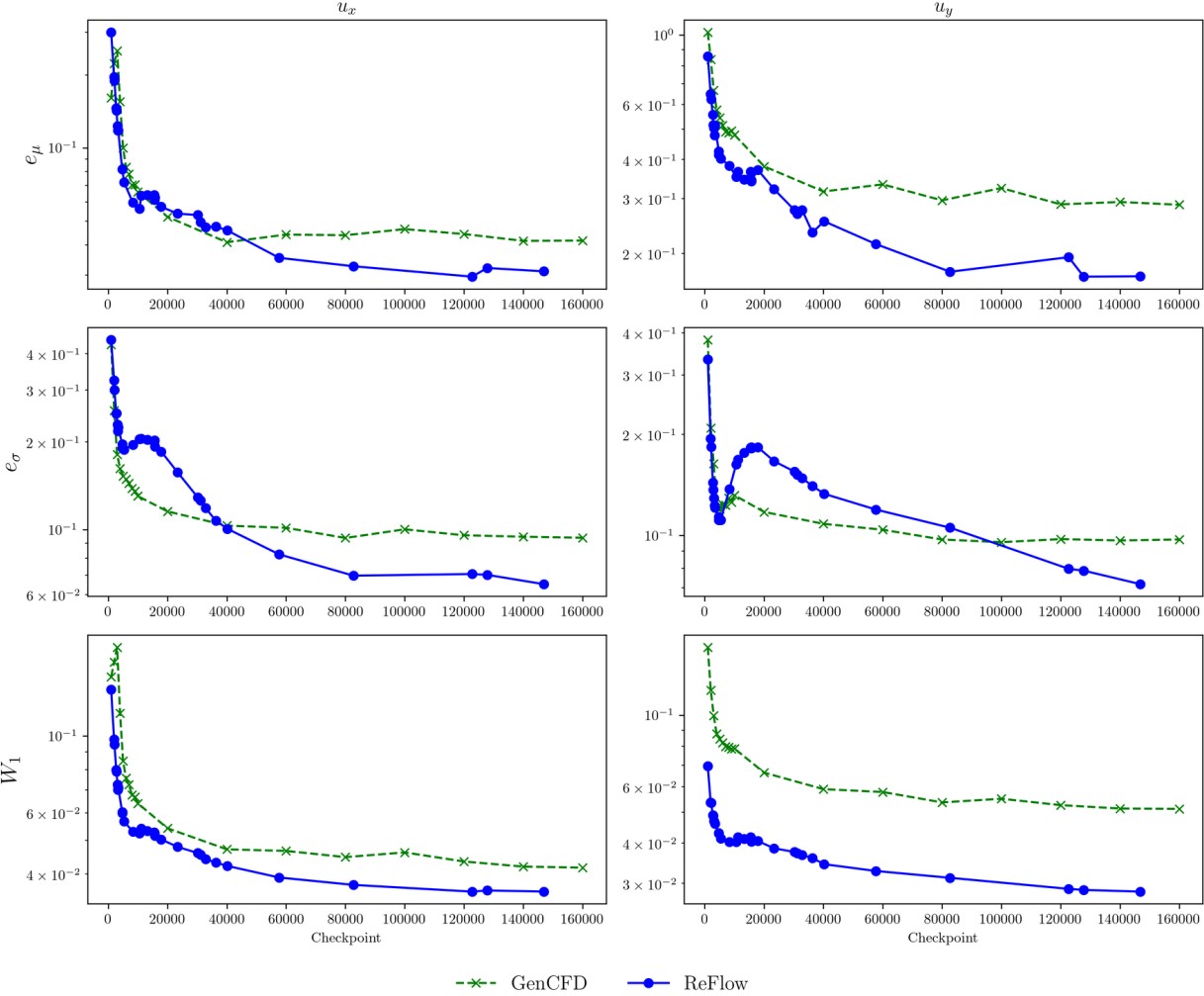

Figure 6: **Training efficiency comparison.** Performance of ReFlow and GenCFD on the Macro-Micro SL ensemble dataset at multiple training checkpoints. Metrics: relative $L^2$ error of the mean $(e_\mu)$ and standard deviation $(e_\sigma)$, and the average 1-point Wasserstein-1 distance $(\overline{W}_1)$, for both velocity components $(u_x, u_y)$. The $y$-axis is logarithmic; the $x$-axis (training iteration) is linear.

## M.7 Experimental Advantages of Straightness

This subsection presents empirical evidence that the rectified trajectories generated by ReFlow enable faster convergence and higher-fidelity reconstructions compared to GenCFD.

**Evaluation settings.** We use the CloudShock dataset throughout; the same results hold across datasets. For each initial condition, both ReFlow and GenCFD are allotted $T = 10$ diffusion steps over normalized diffusion time $\tau \in [0, 1]$. We record outputs at five evenly spaced timesteps $\tau \in \{0.0, 0.25, 0.50, 0.75, 1.0\}$ and analyze: trajectory inpainting of the density field $\rho$; latent-space trajectory visualization via PCA; per-sample average MSE evolution; and evolution of the energy spectrum. All other components (backbone, noise schedule, batch size) are held equal.

### Trajectory Inpainting

We compare ReFlow and GenCFD reconstructions of the density field $\rho$ over five normalized diffusion timesteps $\tau \in \{0.00, 0.25, 0.50, 0.75, 1.00\}$. ReFlow inpaints fine-scale features in far fewer steps, while GenCFD remains overly diffusive. The leftmost panel corresponds to the initial condition $u_0$ upon which both models are conditioned; the following panels show the evolution of the model output as diffusion time progresses.

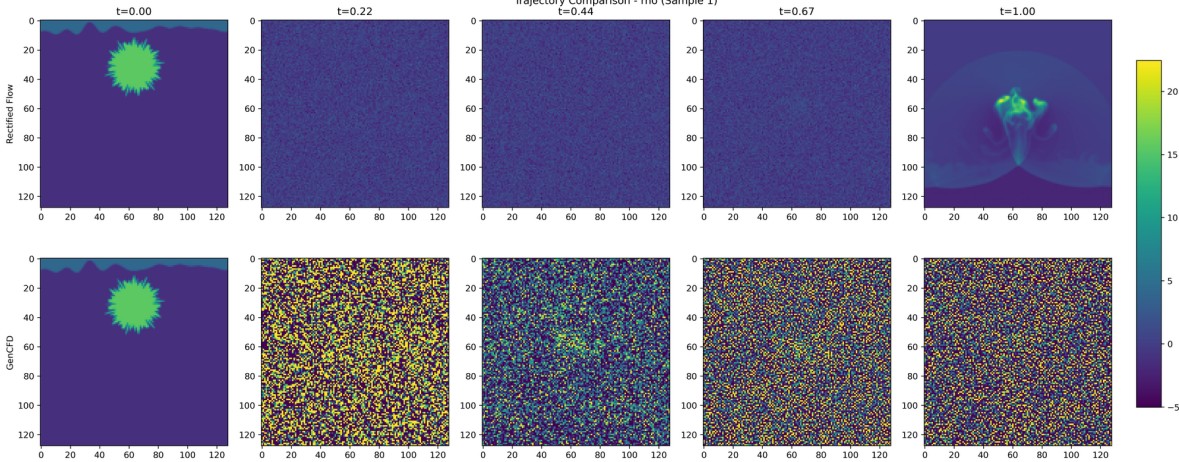

Figure 7: Density trajectories for Sample 1: ReFlow (top) vs. GenCFD (bottom). ReFlow recovers sharp shock fronts by diffusion time $\tau = 1.0$.

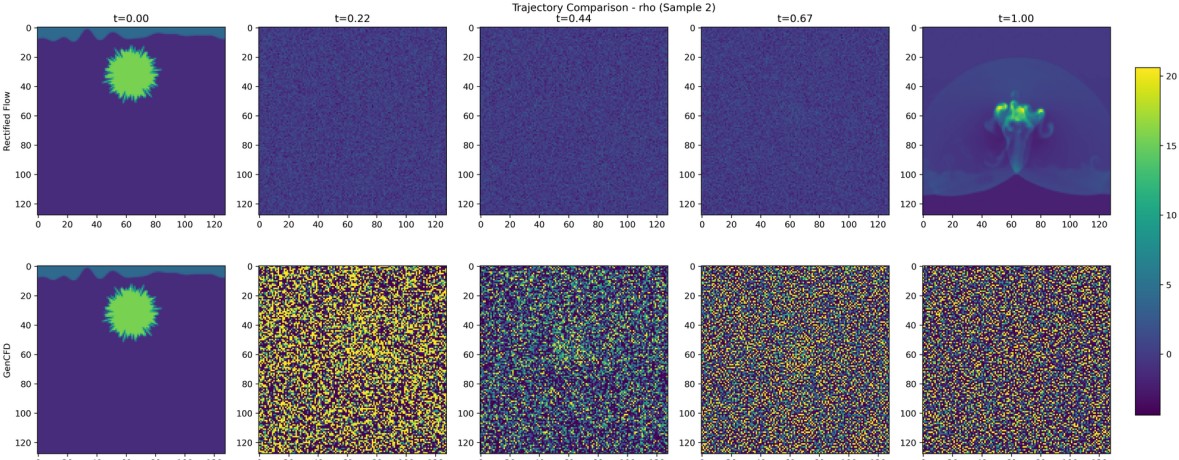

Figure 8: Density trajectories for Sample 2.

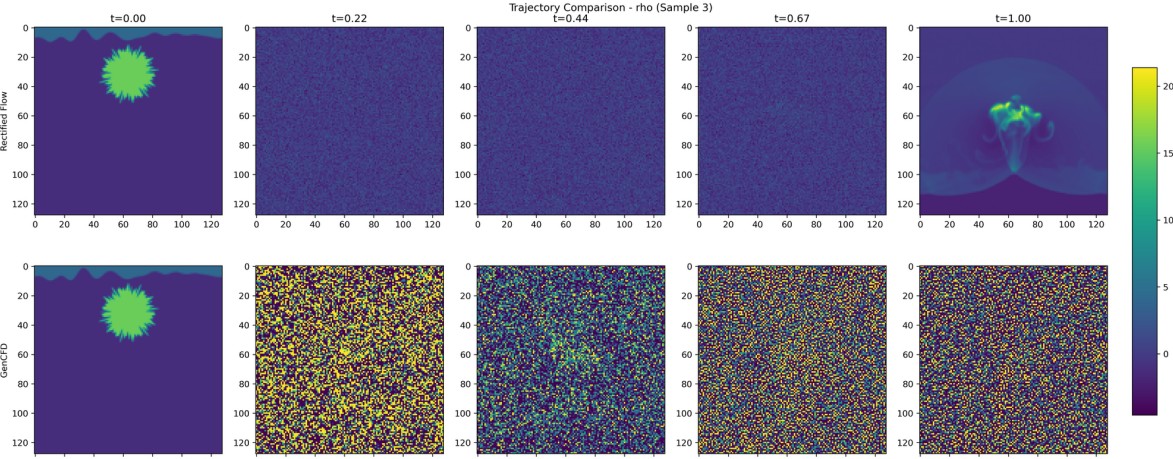

Figure 9: Density trajectories for Sample 3.

**Latent-Space Trajectory Visualization via PCA**

To quantify how directly each model moves through the data manifold, we flatten the $(H \times W \times C)$ field at each timestep into a single feature vector, stack all ReFlow and GenCFD vectors, fit a 3-component PCA, and project each timestep into PC1/PC2/PC3. A straighter path indicates fewer diffusive detours. Because the ReFlow trajectory is much shorter in the projected space, it may visually appear as a tight cluster near the endpoint; this compactness is itself evidence of rectification — the model reaches its target in very few, nearly-collinear steps.

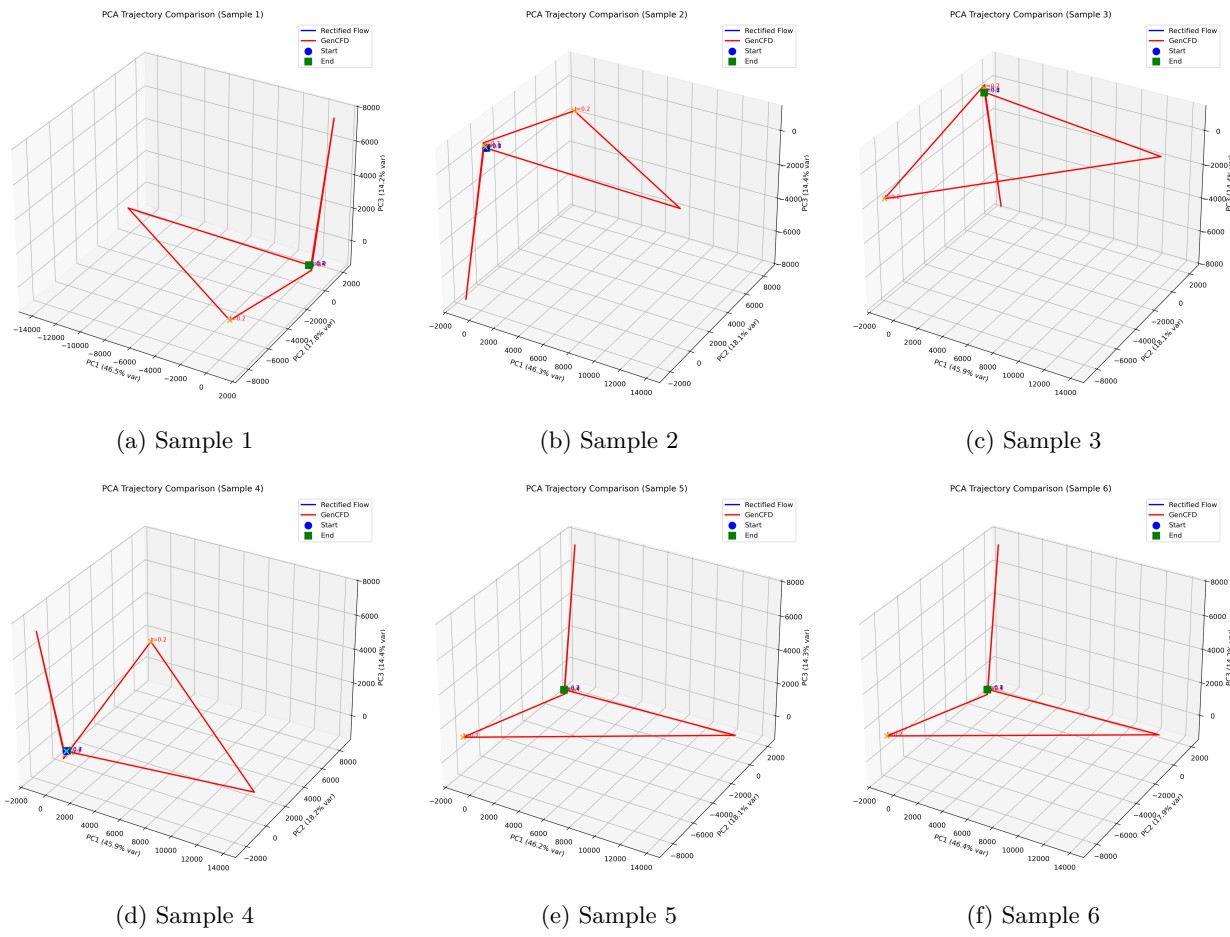

(a) Sample 1      (b) Sample 2      (c) Sample 3

(d) Sample 4      (e) Sample 5      (f) Sample 6

Figure 10: **3D PCA latent-space trajectories (Samples 1–6).** Flattened snapshots at each of $T$ timesteps projected into the top three PCA components. ReFlow (blue) consistently traces a straighter path than GenCFD (red). Circles mark $\tau = 0$, squares mark $\tau = 1$; intermediate timesteps are annotated.

To further illustrate generative dynamics, we visualize single-sample generation trajectories from both models in a 2D PCA space for the RM dataset (Figure 11). For each physical channel ($\rho$, $m_x$, $m_y$, $p$), we apply PCA to the full set of ground truth snapshots to define a common projection basis, then overlay the generative trajectory of a single sample. Kernel density estimates (KDEs) show the empirical data distribution in the same projected space. ReFlow (top) traces a straighter, more compact path over 10 rectified steps; GenCFD (bottom) uses 128 stochastic steps and produces a more diffuse, nonlinear trajectory.

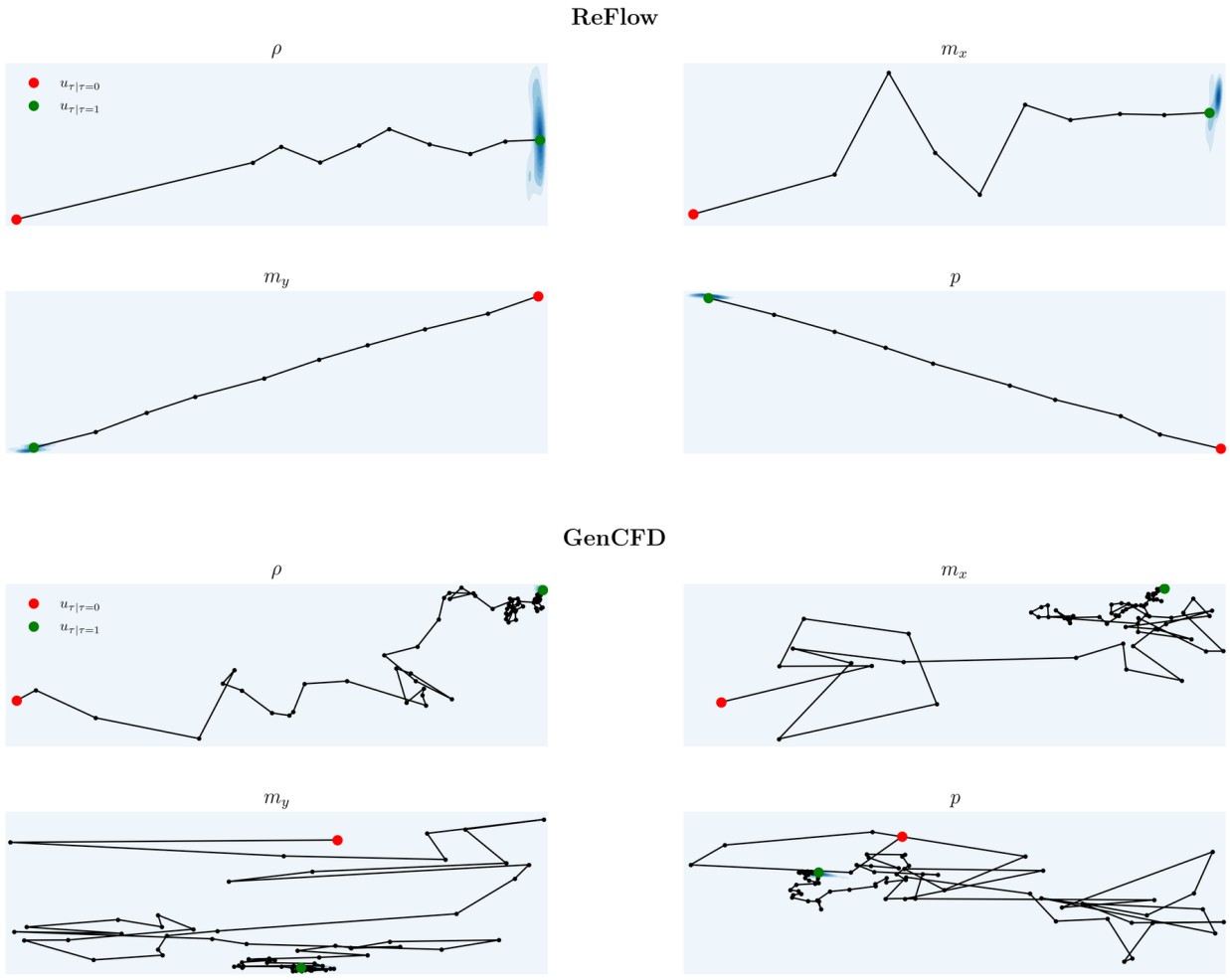

Figure 11: **2D PCA latent-space trajectories.** Flattened velocity snapshots from the RM dataset projected into a 2D PCA space, computed independently per channel ($\rho$, $m_x$, $m_y$, $p$). KDE overlays show the distribution of target fields. ReFlow (top, 10 steps) evolves along straighter, more structured trajectories than GenCFD (bottom, 128 steps).

### Per-Sample Average Error Evolution

We compute the mean-squared error at each timestep $\tau$ and average across all $C$ channels:

$$\overline{\mathrm{MSE}}(\tau) = \frac{1}{C} \sum_{c=1}^{C} \mathrm{MSE}_c(\tau).$$

Figure 12 shows, for Samples 1–6, the log-scaled average MSE (left half of each subfigure) and the normalized error $\overline{\mathrm{MSE}}(\tau)/\overline{\mathrm{MSE}}(0)$ (right half). ReFlow (blue) consistently reduces error faster and to a lower residual than GenCFD (red).

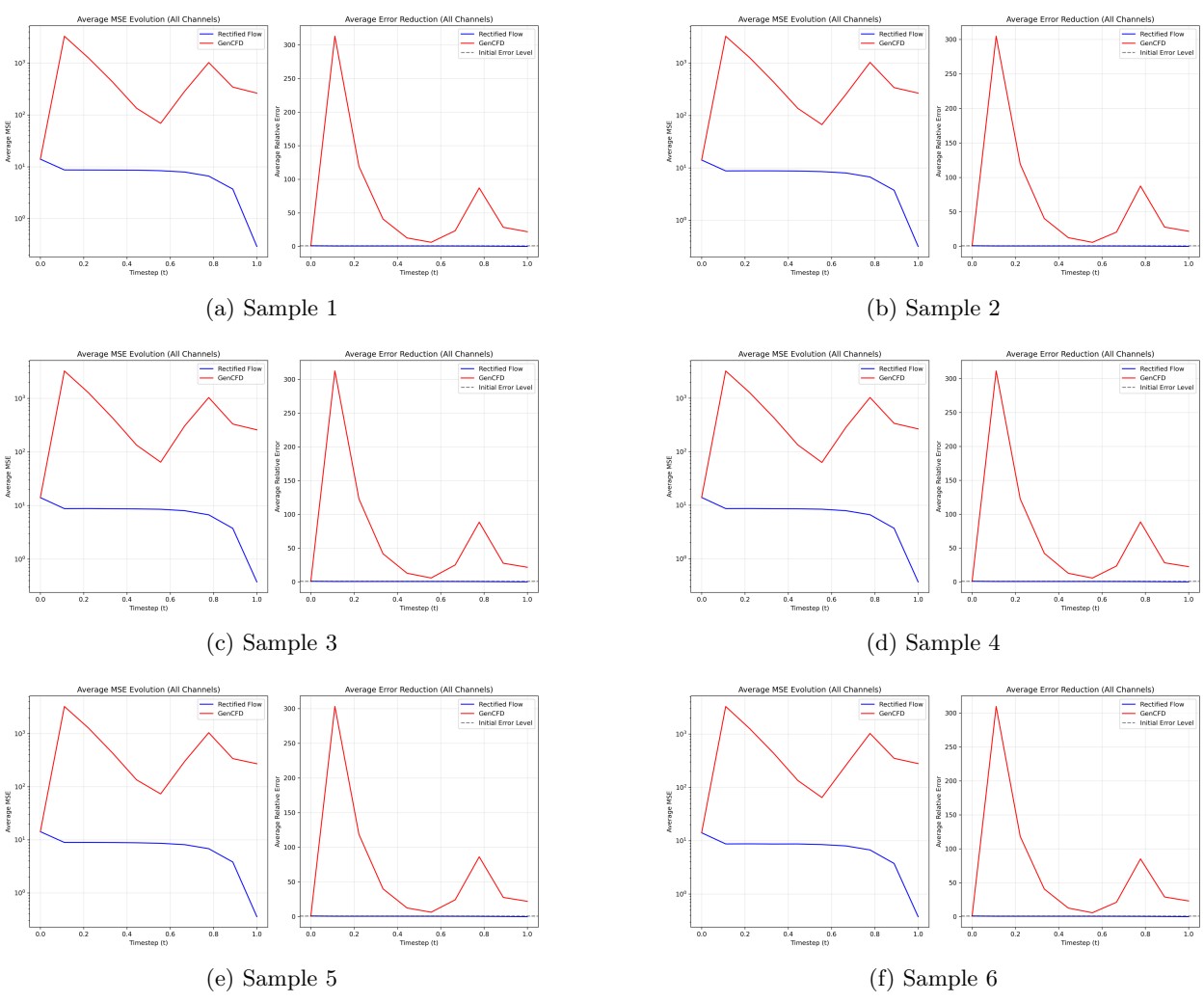

Figure 12: **Per-sample average MSE evolution.** Left half: $\log_{10} \overline{\text{MSE}}(\tau)$; right half: $\overline{\text{MSE}}(\tau)/\overline{\text{MSE}}(0)$. ReFlow (blue) rapidly drives error down and maintains it orders of magnitude below GenCFD (red) across all samples.

**Evolution of the Energy Spectrum**

We examine the 2D radial power spectrum of the energy field $E$ at five normalized diffusion times $\tau \in \{0.00, 0.25, 0.50, 0.75, 1.00\}$. Each figure shows log–log power vs. wavenumber for ReFlow (blue), GenCFD (red), and the ground-truth target (black dashed), with inset log-MSE error annotations. The leftmost panel corresponds to the spectrum of the initial condition $u_0$; subsequent panels track spectral evolution as $\tau$ progresses.

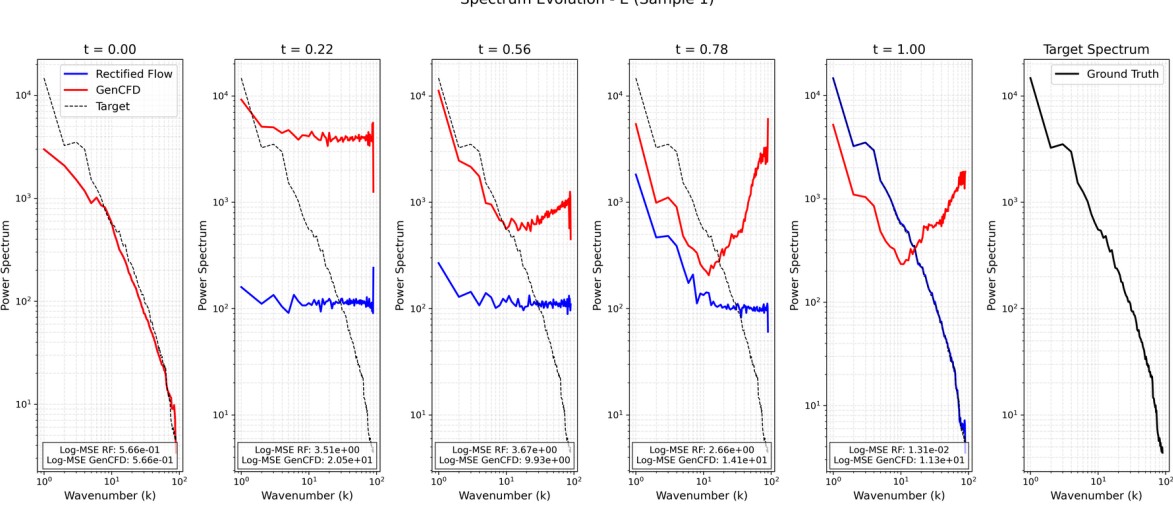

Figure 13: Energy spectrum evolution for Sample 1. ReFlow (blue) outperforms GenCFD (red) in capturing high-wavenumber energy.

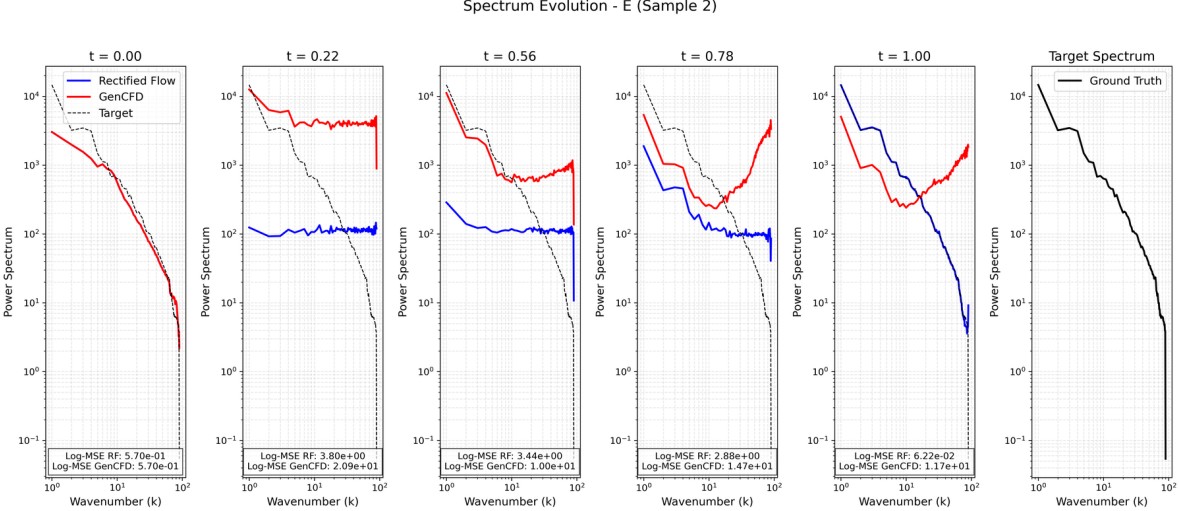

Figure 14: Energy spectrum evolution for Sample 2. ReFlow remains closer to the ground truth (dashed) than GenCFD throughout.

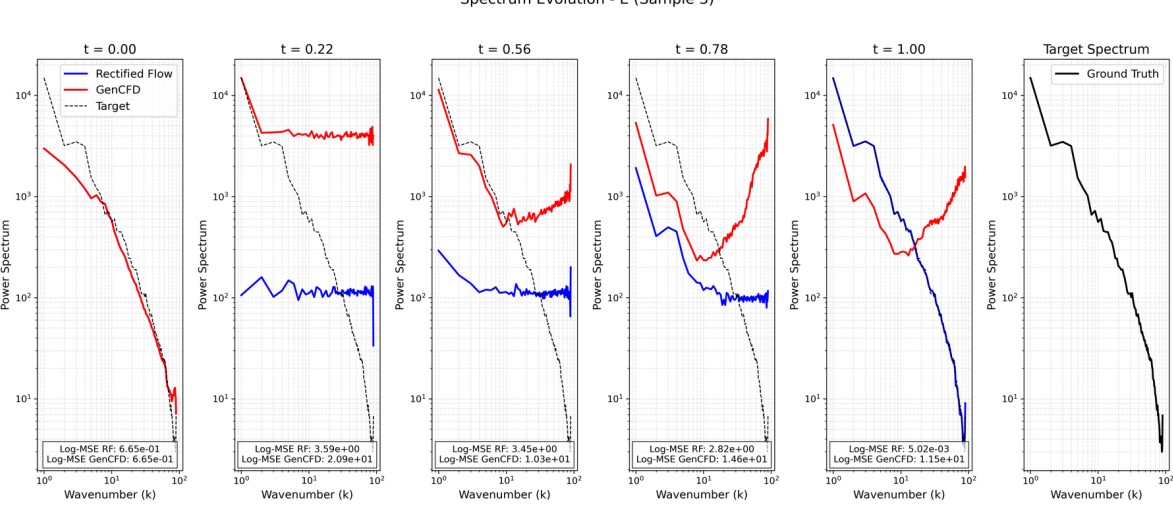

Figure 15: Energy spectrum evolution for Sample 3. Inset log-MSE annotations quantify ReFlow's advantage at each $\tau$.

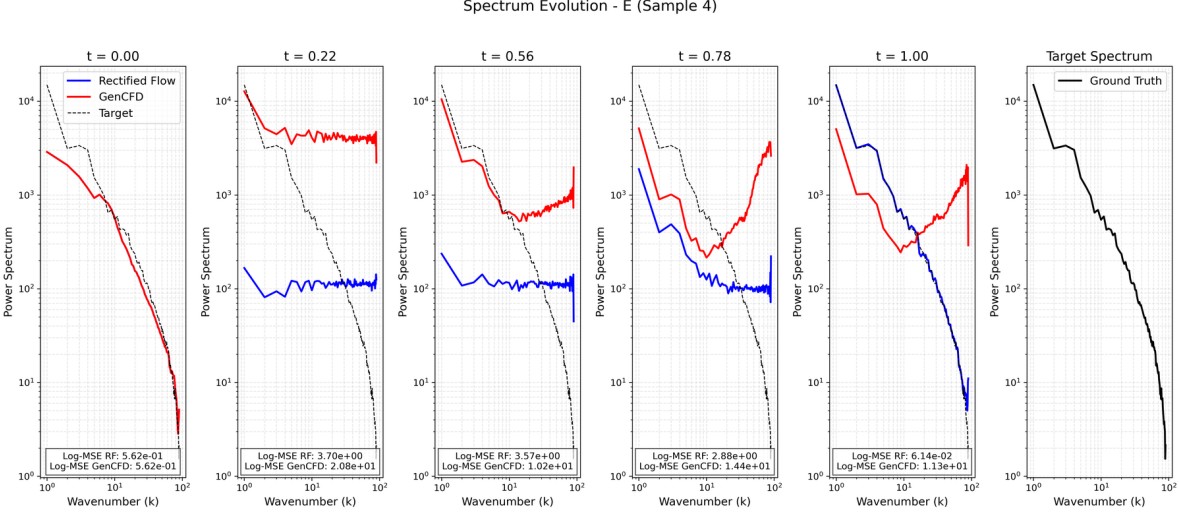

Figure 16: Energy spectrum evolution for Sample 4. ReFlow's spectrum approaches the target as $\tau \to 1.0$.

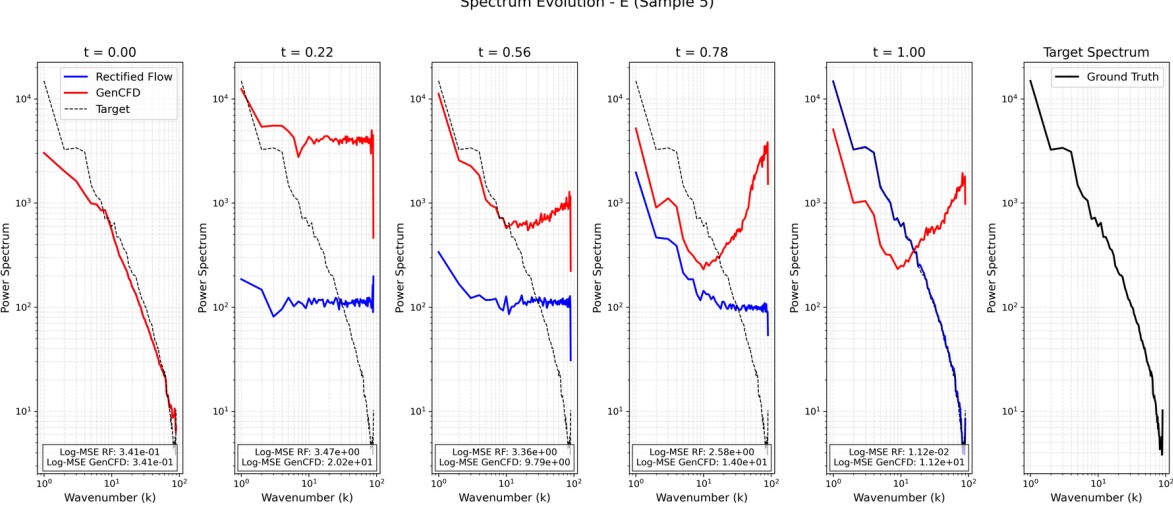

Figure 17: Energy spectrum evolution for Sample 5. ReFlow's superior high-$k$ fidelity is evident throughout.

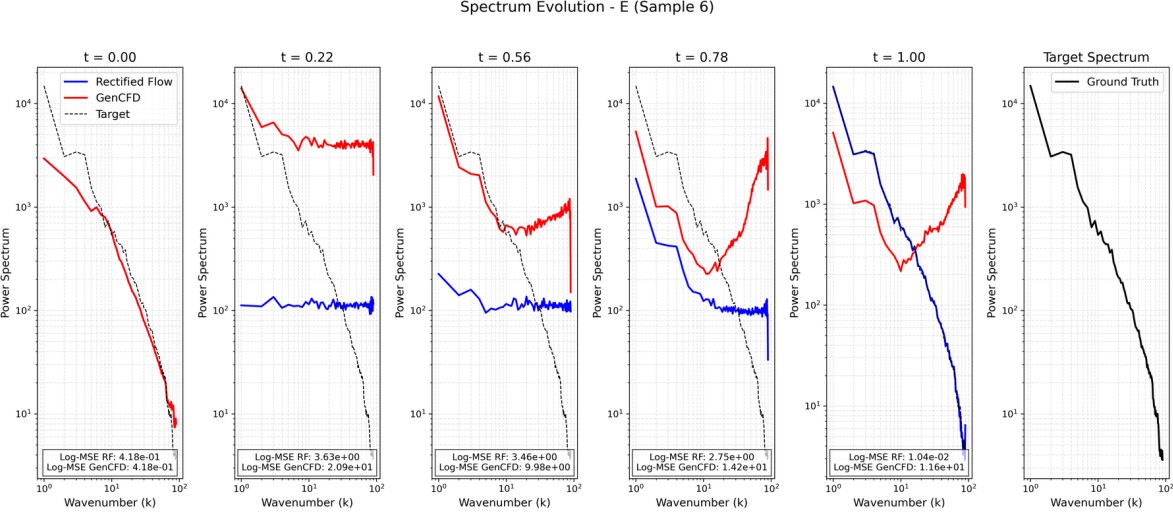

Figure 18: Energy spectrum evolution for Sample 6. ReFlow better captures the ground-truth spectrum's slope across all timesteps.

## N    Computational Considerations

All ReFlow models for 2D fluid-flow tasks use the network configuration in Table 5 and have approximately 5M trainable parameters with identical layer-wise dimensions. We ran each training job for 100,000 iterations on a single NVIDIA Quadro RTX 6000 (Driver 570.124.06, CUDA 12.8) in persistence mode. Resource utilization averaged over the full 100 h training window is reported in Table 12.

Table 12: Aggregate hardware utilization across all training runs.

| Metric | Range | Typical |
|---|---|---|
| GPU power draw (W) | 200–260 | 230 |
| GPU utilization (%) | 30–80 | 55 |
| GPU memory allocated (GB) | 12–17 | 15 |
| GPU temperature (°C) | 40–60 | 50 |
| Process GPU memory usage (%) | 60–80 | 70 |
| Process GPU memory access time (%) | 10–50 | 30 |
| Process CPU threads in use | 7 | 7 |
| System CPU utilization (per core, %) | 60–100 | 80 |
| Process memory in use (GB) | up to 50 | 25 |
| Disk I/O read (MB total) | up to $1.2 \times 10^6$ | $5 \times 10^5$ |
| Disk utilization (%) | 70–75 | 72 |
| Network traffic (bytes total) | $\approx 3 \times 10^{13}$ | $3 \times 10^{13}$ |

Since all datasets are paired with models of identical size and the hardware envelope is effectively the same across runs, any observed performance differences are attributable to the sampling strategy (noise schedule, ODE steps) rather than to model-size or hardware variations. (Network traffic is due to Wandb logging.)

We explored architectures ranging from approximately 5M to 10M trainable parameters; across this range we observed no significant differences in final predictive performance or convergence behavior. All other hyperparameters were held fixed across comparisons, ensuring that differences in accuracy and runtime reflect only the conditioning and sampling strategies under study.

## O    Code and Data Availability

Prior to acceptance, we will make available all three datasets, the complete data-processing scripts, and a fully functional training and evaluation pipeline for ReFlow, including a model checkpoint that can be loaded and tested directly. A more comprehensive, user-friendly code release will follow shortly thereafter.

## P    Additional Visualizations

**Test-time ensemble perturbation (evaluation only).**  Following the protocol of *GenCFD* Molinaro et al. (2024), we construct small micro-ensembles around each test initial condition $\bar{u}$ by sampling $M_{\mathrm{micro}}$ perturbed fields uniformly in an $\varepsilon$-ball centered at $\bar{u}$. Each perturbed member is advanced to evaluation time $t$ with a high-fidelity PDE solver to produce a reference ensemble $\{u_{\mathrm{ref}}^{(j)}(t)\}_{j=1}^{M_{\mathrm{micro}}}$. Our model generates a corresponding ensemble $\{u_{\mathrm{model}}^{(j)}(t)\}_{j=1}^{M_{\mathrm{micro}}}$ from the same perturbed inputs, but without ever observing ensembles during training. We compare these via $e_\mu$, $e_\sigma$, and $\overline{W}_1$ (see §M).

**Out-of-distribution testing (cloud-shock and shear-flow).**  Because the micro-perturbations render each initial field slightly OOD, this evaluation probes generalization under small chaotic perturbations. Our ReFlow ensemble closely tracks the reference uncertainty and multiscale statistics, whereas deterministic baselines collapse to a single trajectory and fail to capture the conditional variability.

**In-distribution scaling (Richtmyer–Meshkov).** For the RM dataset, we additionally study the effect of training set size on a held-out test set from the same distribution. Even deterministic approaches improve with more data, but still underperform on the micro-ensemble test, requiring significantly larger training sets to match ReFlow's accuracy and exhibiting poor calibration under small perturbations.

In this appendix we provide comprehensive qualitative comparisons between all baselines (GenCFD variants, UViT, FNO) and ReFlow across three 2D fluid-flow benchmarks. For each task we show:

- **Mean fields.** Ensemble averages reveal how deterministic models (UViT/FNO) diffuse fine-scale features versus preserving sharp interfaces (ReFlow, GenCFD variants).

- **Uncertainty maps.** Per-pixel standard deviations illustrate whether uncertainty localizes in physically meaningful regions (shocks, shear interfaces) or remains overly diffuse.

- **Random samples.** Representative samples from ReFlow alongside the deterministic FNO and the hybrid GenCFD ∘ FNO, highlighting ReFlow's ability to produce realistic, high-resolution flow fields that preserve physical coherence across scales.

- **Spectral and pointwise comparisons.** (Fig. 22: RM global $E(k)$, local spectra, and pointwise density histograms; Fig. 25: SL global $u_x$ spectra and pointwise $u_x$ distributions; Fig. 29: CS global $m_y$ spectra and pointwise energy distributions.) Combined Fourier-space and histogram panels quantify multiscale fidelity and calibration.

Together, these visualizations underscore ReFlow's ability to maintain sharper physical structures, produce better-calibrated uncertainties, and faithfully reproduce multiscale spectral statistics.

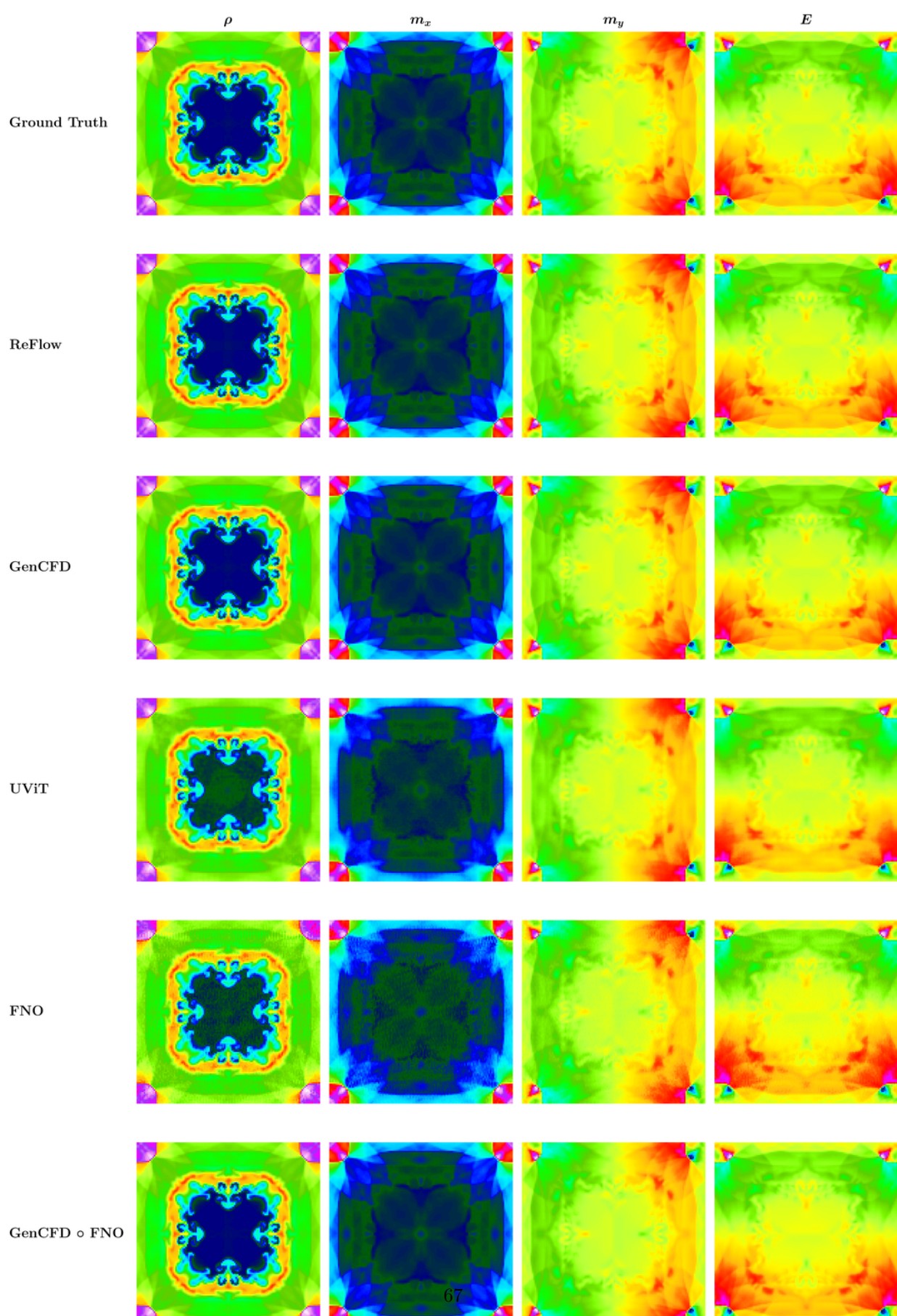

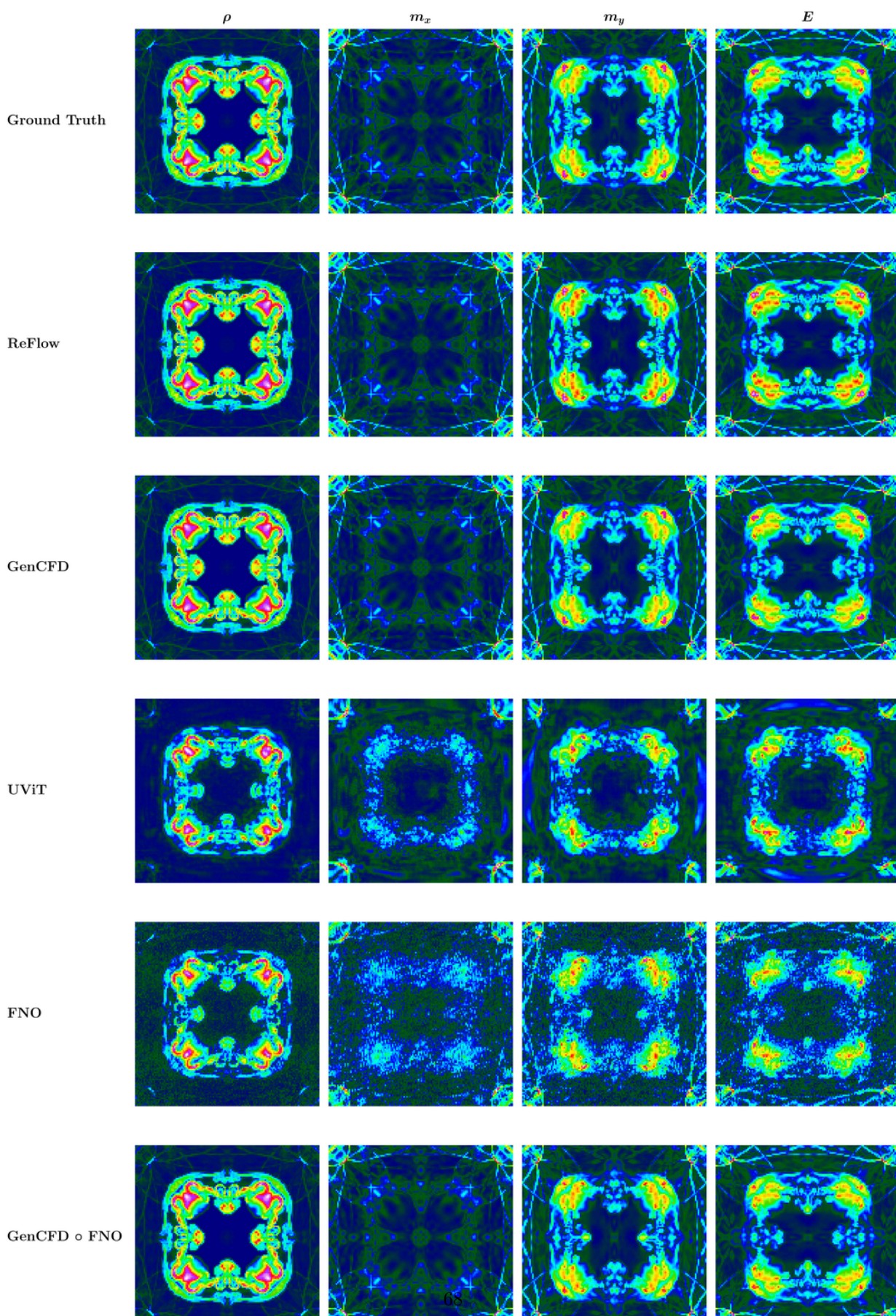

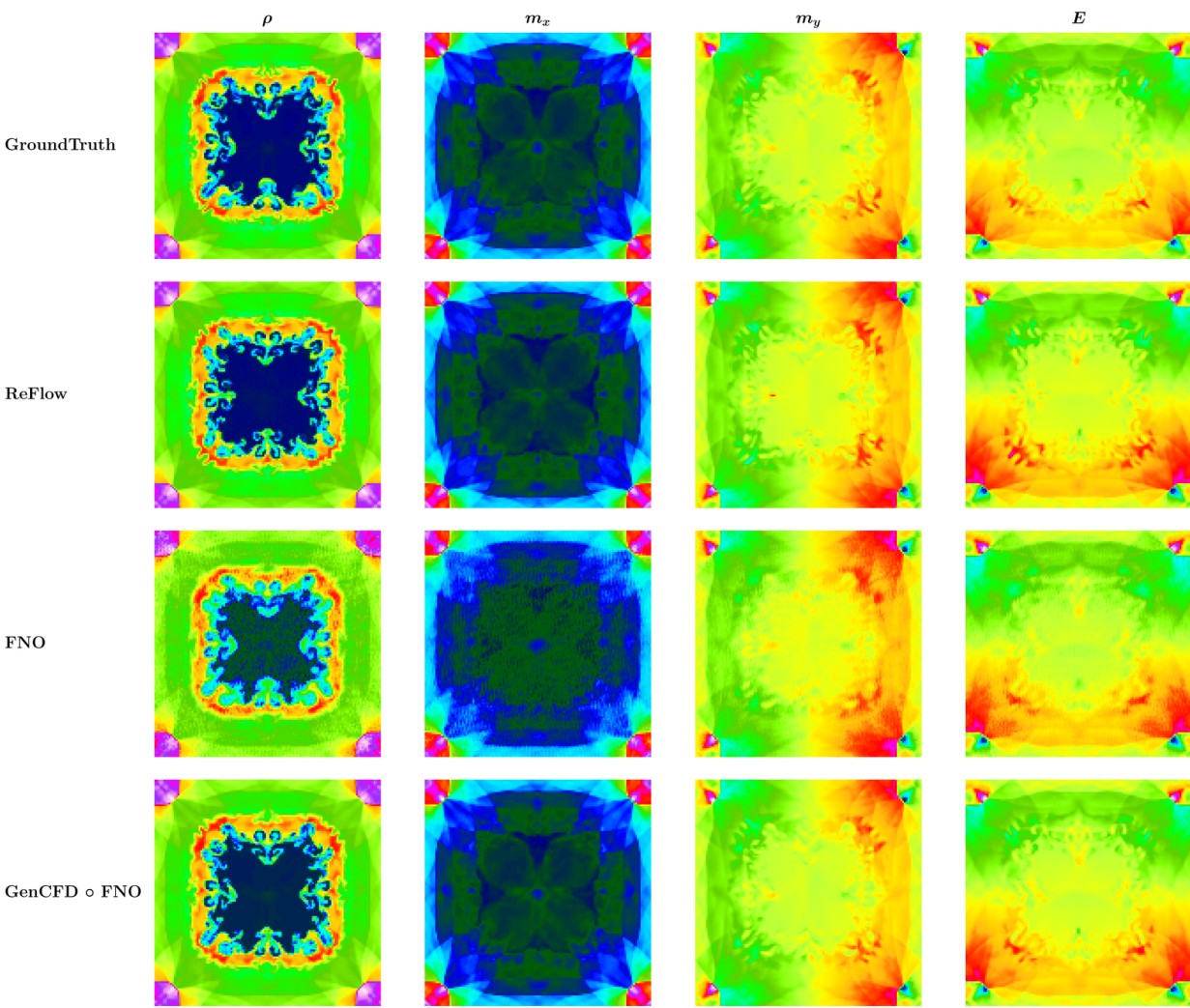

Figure 21: **Richtmyer–Meshkov Individual Sample Comparison**: A representative random test sample from the Richtmyer–Meshkov dataset, visualized across all four physical channels ($\rho$, $m_x$, $m_y$, $p$). All models produce physically plausible results, with only little variation in fidelity to fine-scale structures. ReFlow offers the closest visual match to the ground truth, preserving coherent details and sharp interfaces. Notably, the GenCFD ∘ FNO hybrid enhances spectral richness by reconstructing high-frequency features based on a low-frequency FNO prior, illustrating its capacity to recover sharper modes even when starting from a coarse prediction.

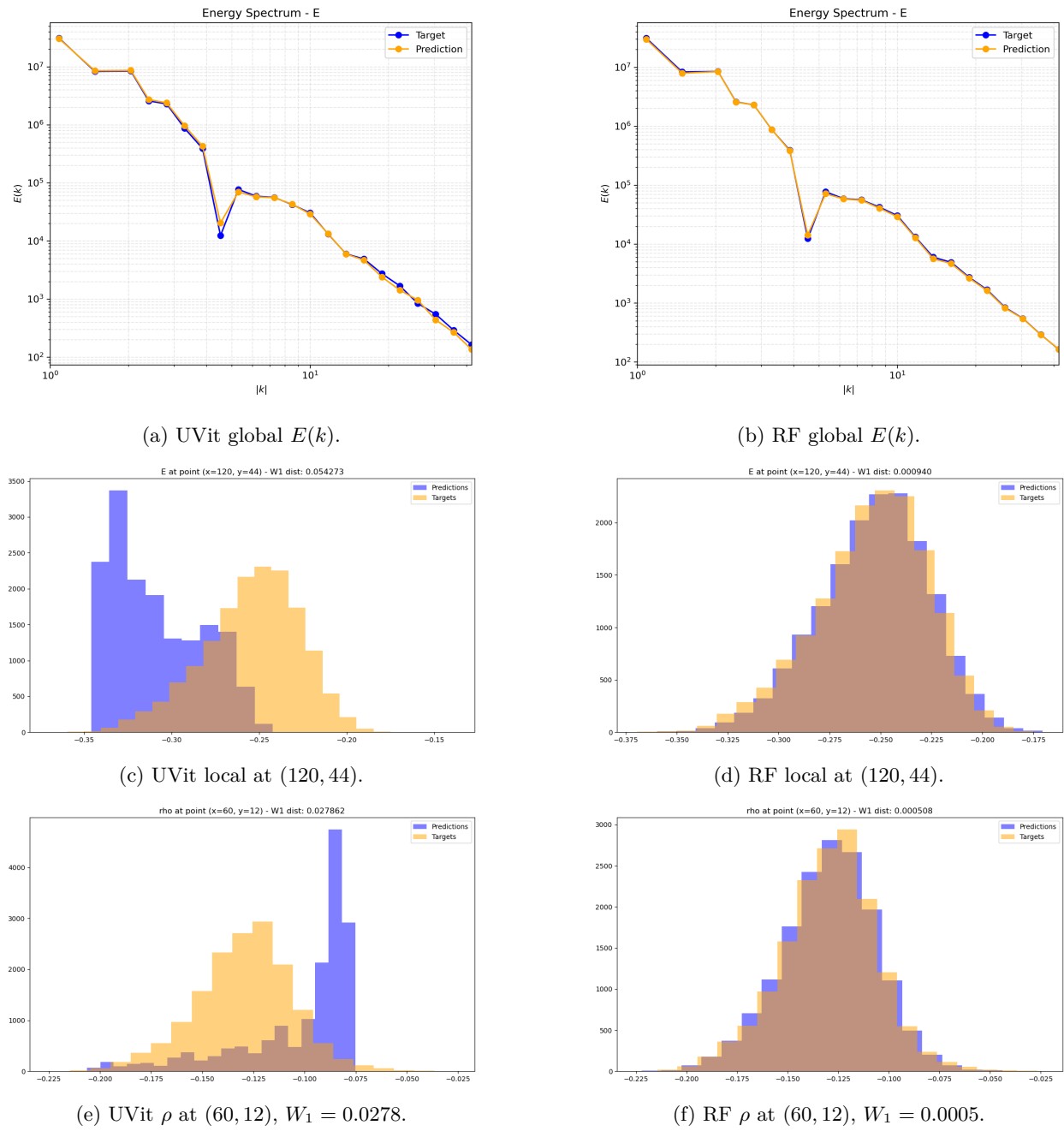

Figure 22: Richtmyer–Meshkov: spectral and pointwise distribution comparisons between UVit and Rectified Flow.

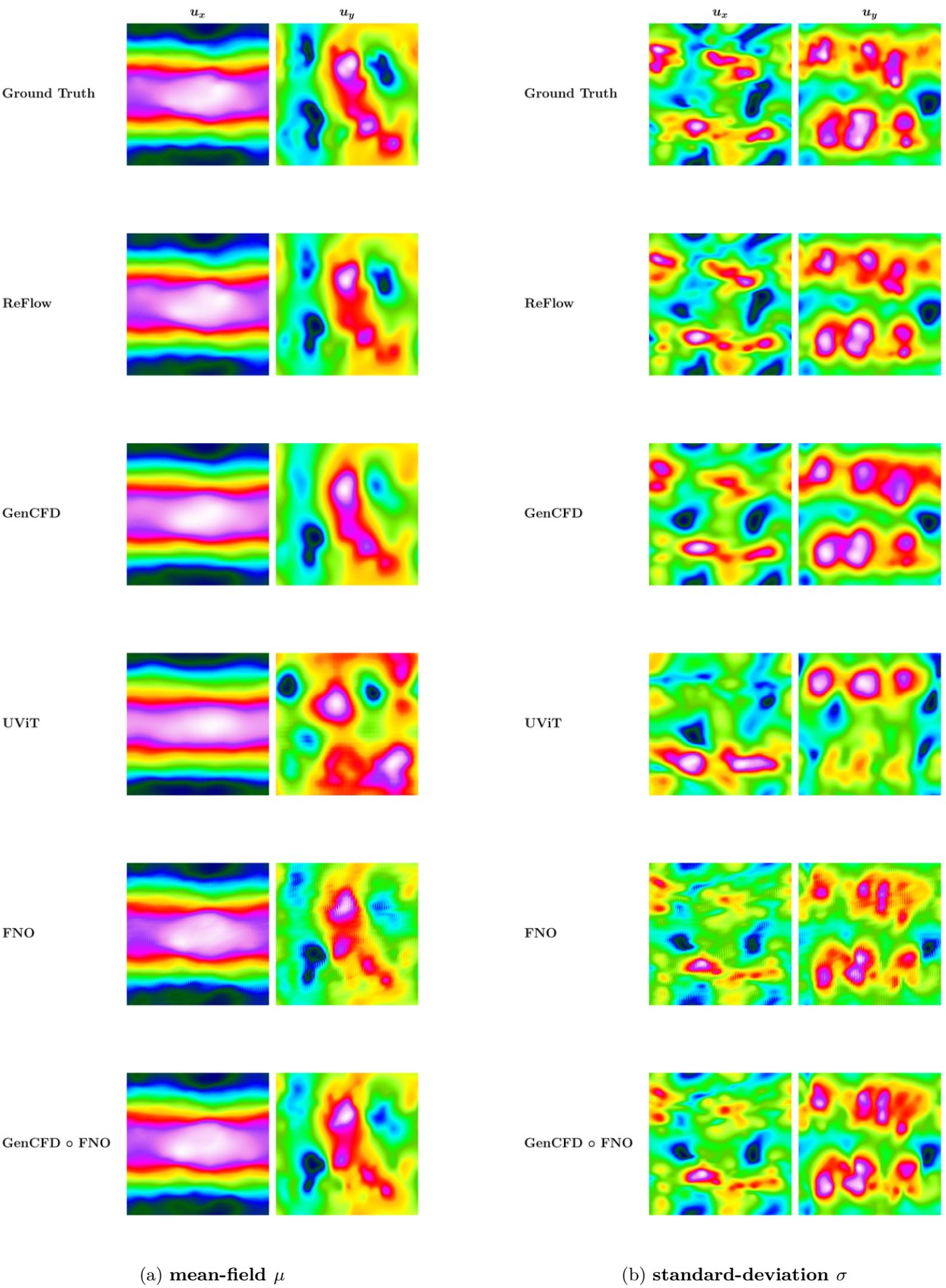

(a) **mean-field** $\mu$                    (b) **standard-deviation** $\sigma$

Figure 23: **Shear-Layer Mean and Standard Deviation Comparison**: Visualization of mean-field ($\mu$) and standard-deviation ($\sigma$) predictions for the Shear-Layer dataset across both velocity components ($u_x$, $u_y$). For each model, the left half shows the predicted mean fields and the right half shows per-pixel standard deviations. The top row contains the reference ground truth. Despite the reduced channel complexity of this

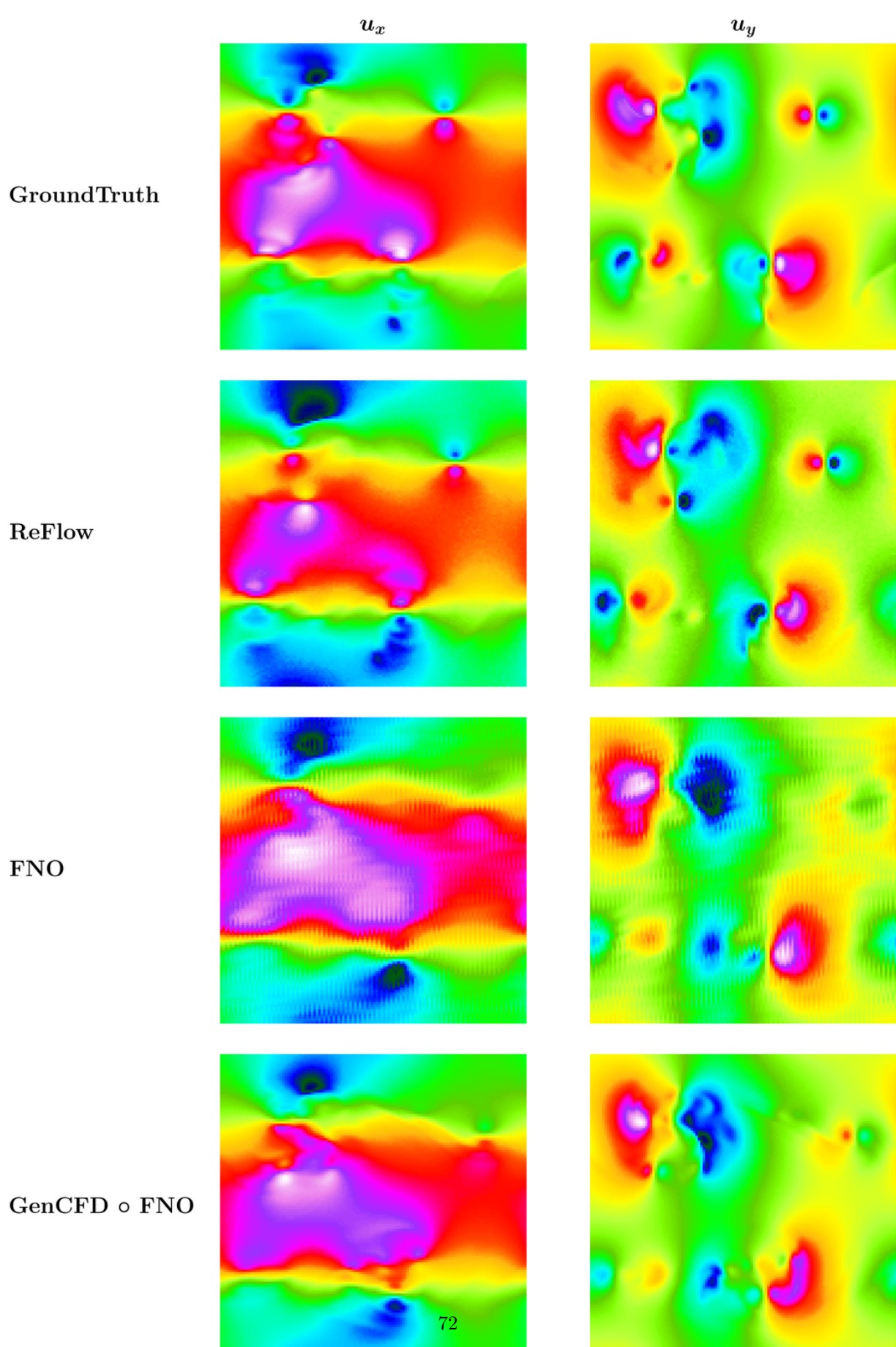

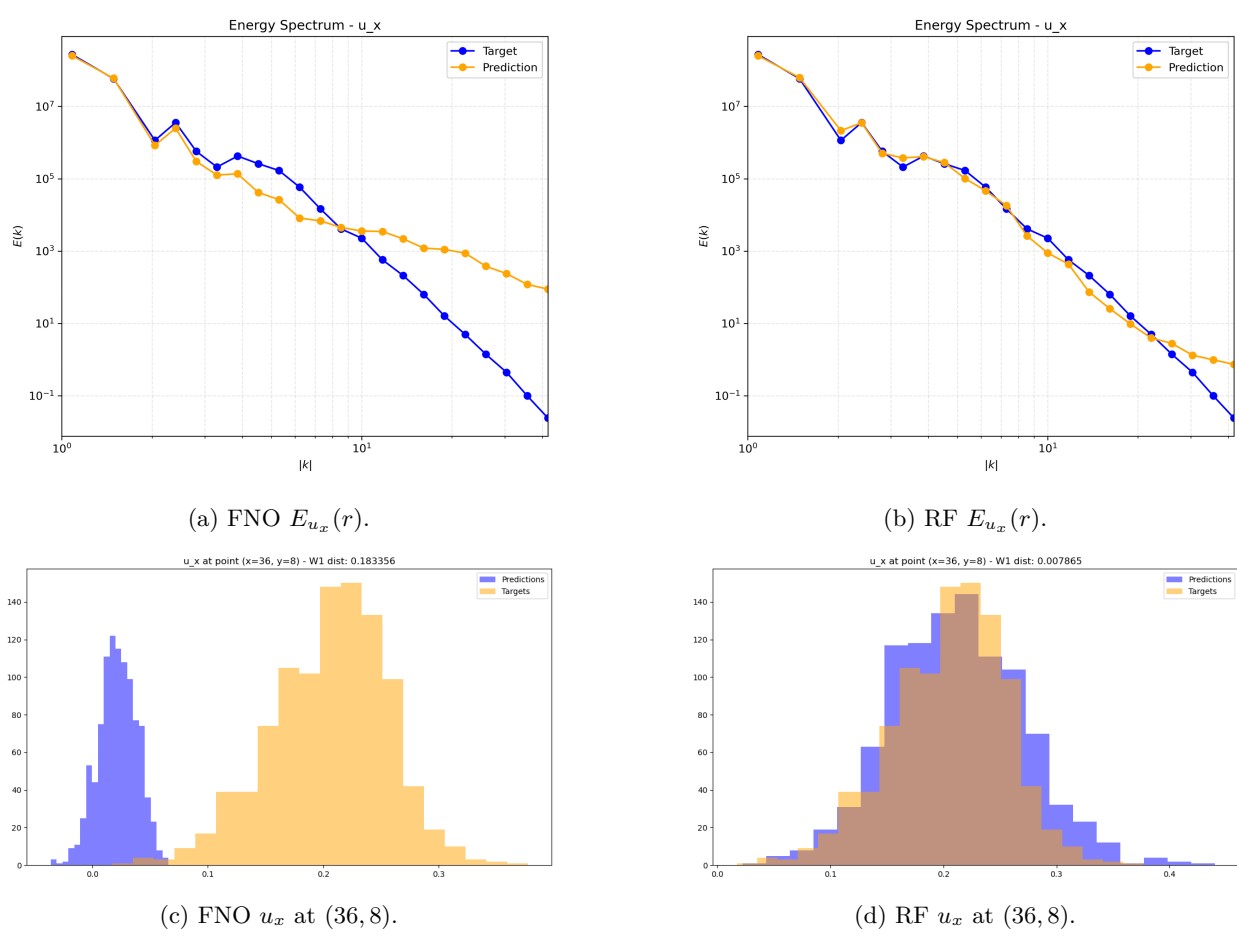

(a) FNO $E_{u_x}(r)$.

(b) RF $E_{u_x}(r)$.

(c) FNO $u_x$ at $(36, 8)$.

(d) RF $u_x$ at $(36, 8)$.

Figure 25: Shear-layer: global $u_x$ spectra and pointwise $u_x$ distributions.

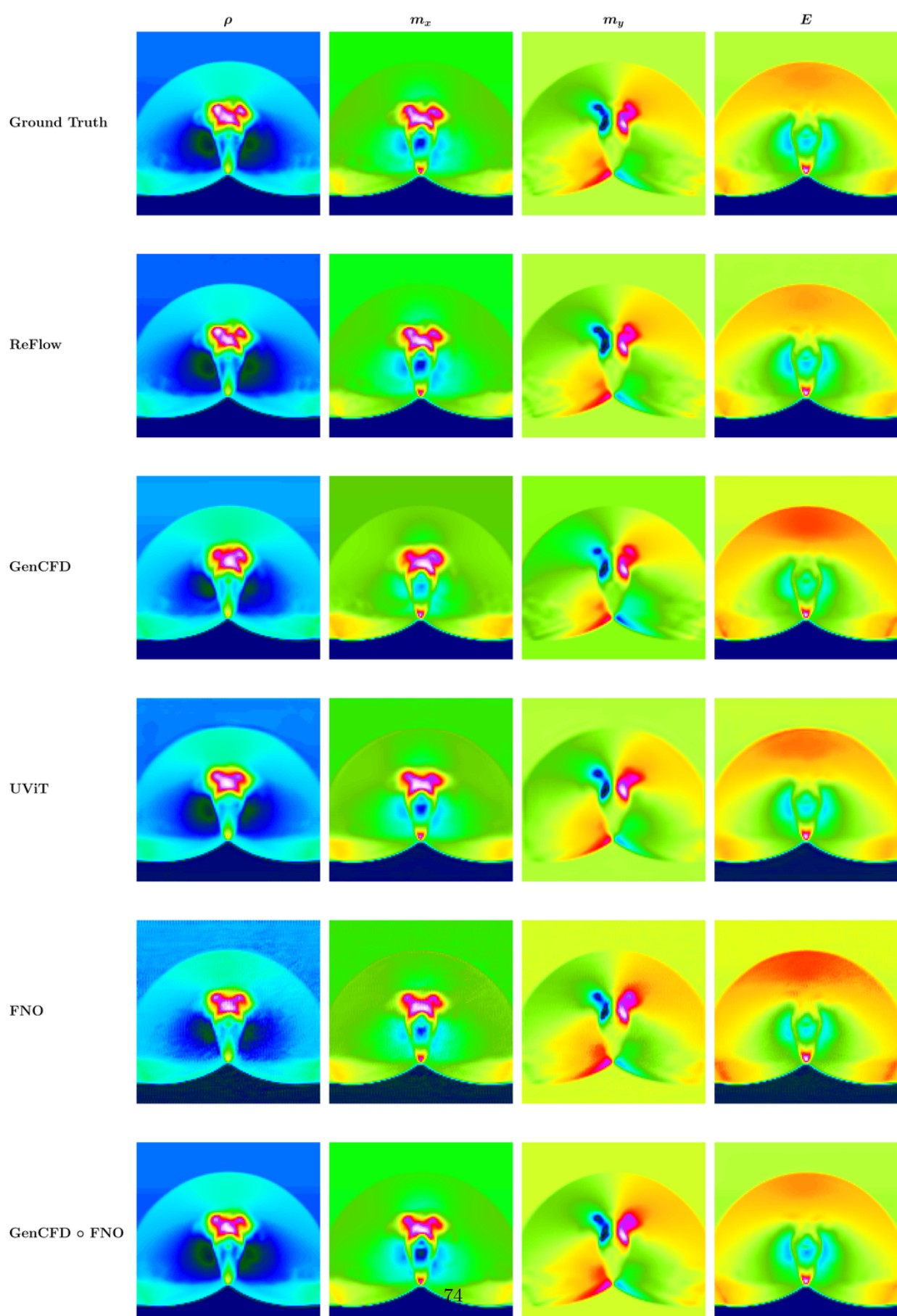

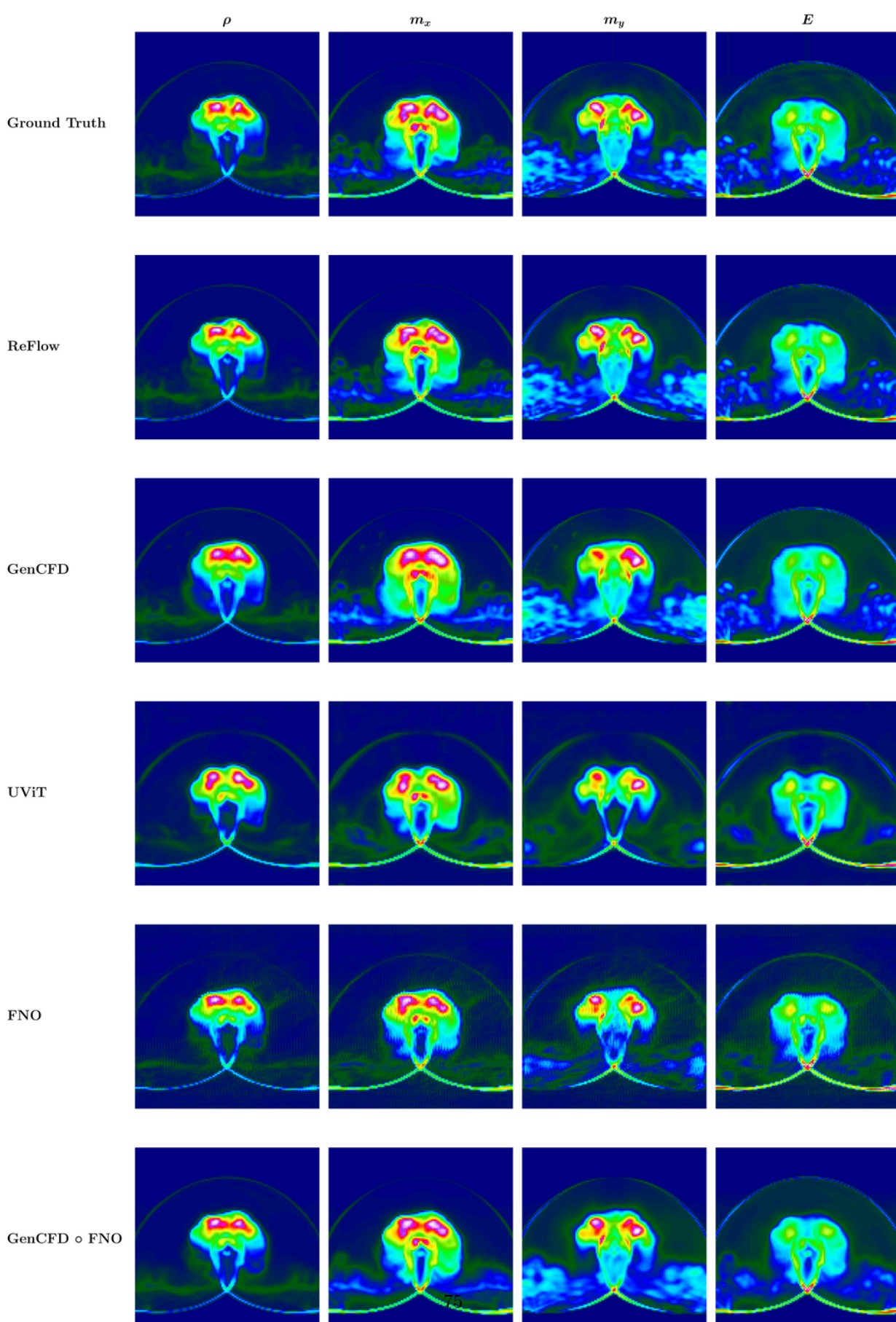

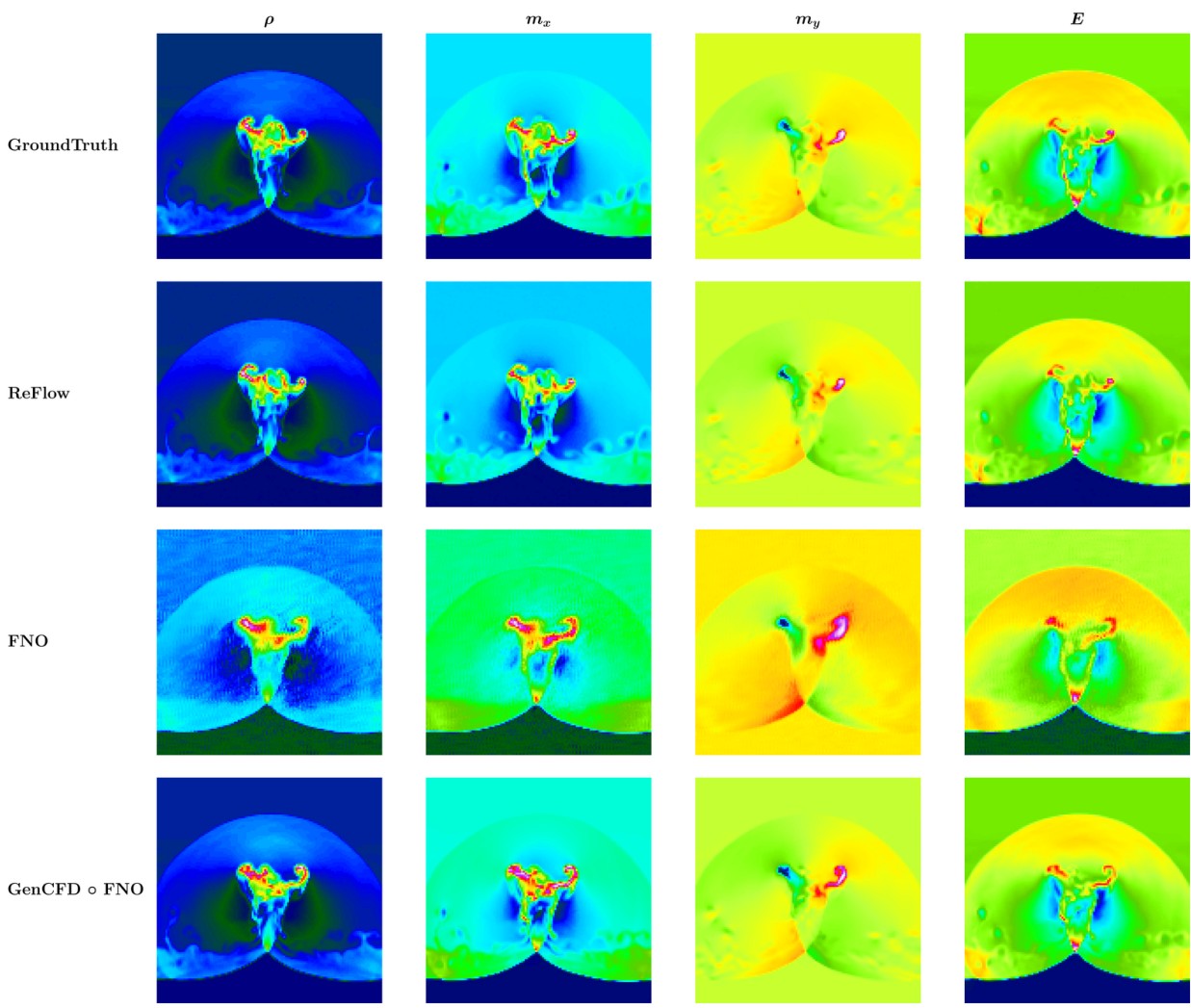

Figure 28: **Cloud–Shock Individual Sample Comparison**: A representative random test sample from the Cloud–Shock dataset, displayed across all physical channels. While all models produce plausible results, ReFlow and the hybrid GenCFD ∘ FNO most closely align with the ground truth, particularly in preserving fine-scale detail. The spectral reconstruction effect of the hybrid model is especially evident, recovering high-frequency structures from the coarse FNO prediction and enhancing the overall visual fidelity.

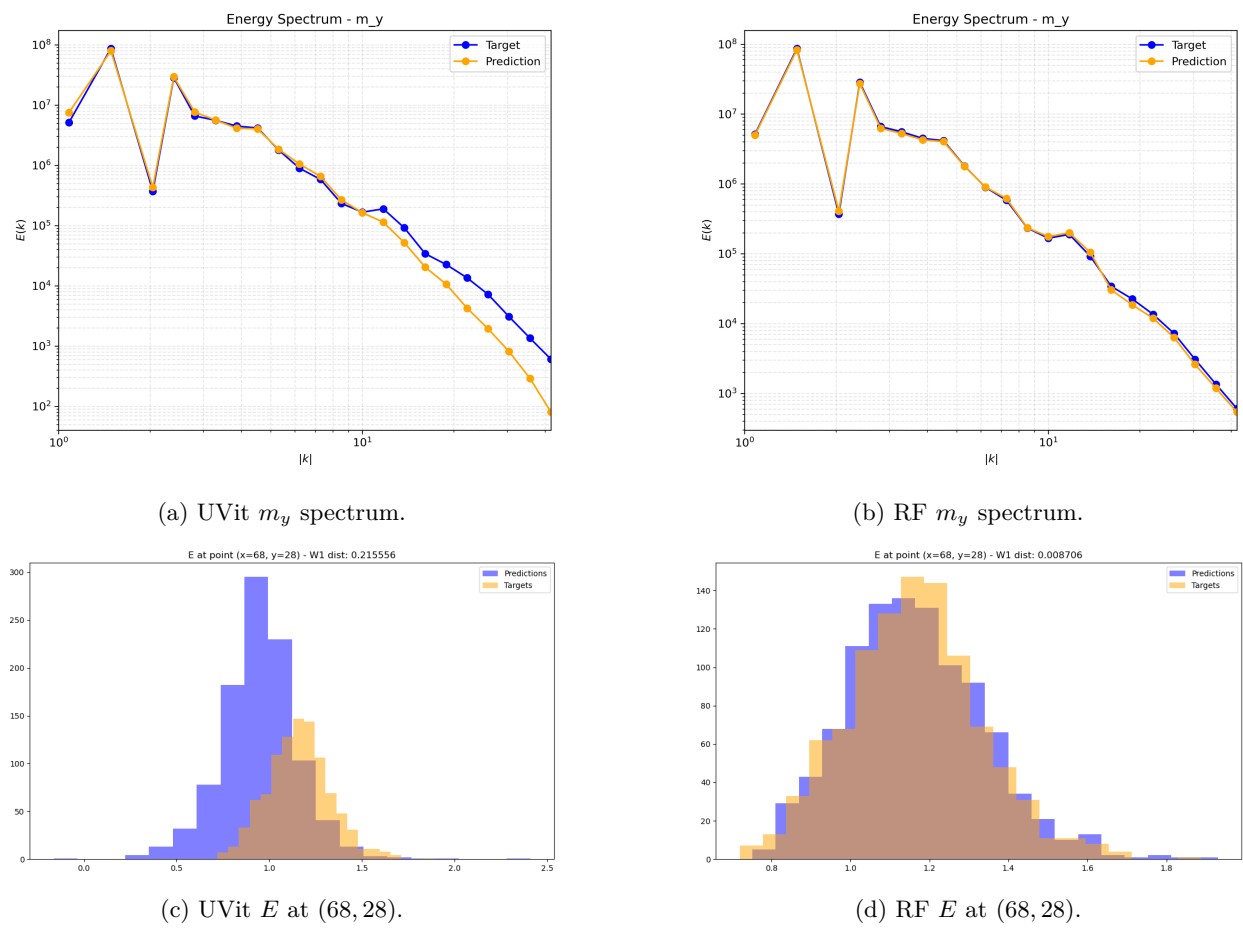

(a) UVit $m_y$ spectrum.

(b) RF $m_y$ spectrum.

(c) UVit $E$ at $(68, 28)$.

(d) RF $E$ at $(68, 28)$.

Figure 29: Cloud-shock: spectral and pointwise energy comparisons.

