# OpenReview forum: "Rectified Flows for Fast Multiscale Fluid Flow Modeling"
_TMLR — Accepted by TMLR_

### Review · Reviewer_Hait · 2026-02-26

**Summary Of Contributions:**

The paper proposes ReFlow, a rectified-flow–based conditional generative surrogate for multiscale fluid PDEs. Instead of diffusion sampling, generation is performed by integrating a learned deterministic ODE. The authors provide a low-level error decomposition (coverage, fit, straightness, discretization) and show empirically—on three 2D multiscale benchmarks—that ReFlow matches or improves diffusion-based baselines while requiring far fewer sampling steps.

Strengths:

- Clear conceptual contribution. The error decomposition offers intuition for when low-step ODE sampling is sufficient and connects theory to practice.

- Strong empirical efficiency gains. On 2D benchmarks, ReFlow achieves comparable or better statistical accuracy (including spectral behavior) with substantially fewer function evaluations.

- Well-diagnosed experiments. The paper includes spectral analysis, ablations, and integrator robustness studies that support the main claims.

Weaknesses:

- Strong theoretical assumptions. The results rely on regularity and uniqueness assumptions (e.g., Lipschitz conditions) that are not fully discussed in the context of the discrete experimental setup.

- Limited to 2D experiments. It is unclear how well the method scales to 3D turbulence or higher-dimensional settings.

- Baseline coverage could be stronger. Comparisons to more aggressively accelerated diffusion samplers (e.g., modern ODE/SDE solvers) and clearer wall-clock evaluations would strengthen the practical claims.

**Audience:**

Yes

**Audience Explanation:**

The work sits at the intersection of generative modeling and scientific machine learning. Researchers interested in fast generative surrogates, uncertainty quantification, and ML for PDEs would likely find the results relevant.

**Claims And Evidence:**

Yes

**Claims Explanation:**

The theoretical framing is coherent and aligns with the empirical observations. The experiments consistently show competitive or improved statistical accuracy with dramatically reduced sampling cost. However, additional clarity around theoretical assumptions and broader baseline comparisons would make the claims even stronger.

**Requested Changes:**

Critical:

- Clarify and explicitly state the key theoretical assumptions and their relevance to the discrete experimental setting.

- Provide clearer runtime comparisons (e.g., wall-clock time) and, if possible, comparisons with modern accelerated diffusion samplers.

Recommended:

- Report variability across multiple runs (e.g., error bars).

- Add a more explicit discussion of scalability to 3D and larger-scale problems.

---

> ### Author Response · Authors · 2026-04-09
> **Response to reviewer Hait (Part 1)**
>
> We thank the reviewer for the positive assessment of the paper’s core contribution. We are glad that the reviewer found both the theory–practice connection and the empirical efficiency gains convincing. The requested changes are reasonable, but several of them concern clarifications that are already partially present in the draft and mainly need to be made more explicit.
>
> **(1) Theoretical assumptions vs. the actual experimental setting.**
> We agree that this should be stated more prominently. The theory is intended for the **finite-dimensional discretized surrogate actually used in the experiments**, not as an unrestricted claim about infinite-dimensional turbulent PDEs. Section 4.1 already states that, while the exposition is written in $U=L^2(D;\mathbb{R}^m)$, “in practice we work with a Galerkin discretisation $\mathbb{R}^{dN}\subset U$,” and Proposition 5.1 formulates exact transport under uniqueness / regularity assumptions precisely in that finite-dimensional setting. Likewise, Theorem 4.2, Corollary 5.2, and Lemma 4.4 place regularity assumptions on the **learned velocity field** $v_\theta$ integrated at inference time, e.g. Lipschitzness in state and $C^1$-type control for discretization error. These are assumptions on the learned transport ODE, not blanket assumptions on the underlying turbulent PDE itself.
>
> We will revise the paper to separate these two layers more clearly:
>
> * the **finite-dimensional correctness / discretization assumptions** relevant to the implemented model, and
> * the **law-level / statistical-solution interpretation** used to motivate the coverage term and the evaluation metrics.
>
> Indeed, the coverage analysis is already explicitly resolution-aware: Section 4.3 is written in terms of an effective cutoff $K_c$, Fourier truncation, and structure-function / spectral tails as the unavoidable approximation floor at fixed grid resolution. So the theory is not abstracted away from the experiments; it is built precisely to explain what can and cannot be captured on the finite surrogate used in practice.
>
> **(2) Runtime comparisons and stronger ODE/SDE diffusion baselines.**
> We agree that the runtime story should be presented more cleanly, and we will revise the text accordingly. In particular, we will unify notation throughout the paper by using $u_i$ for the initial datum, $u_f$ for the evolved datum, and $\tau$ exclusively for rectified-flow time; axes, captions, and Algorithm 1 will be updated accordingly.
>
> More importantly, we want to clarify the conceptual distinction between our method and existing ODE-based samplers for diffusion. Methods such as DPM-Solver or PF-ODE accelerate sampling of a **fixed learned score model**; they do **not** change the learned dynamics. Our method instead learns a **different conditional transport field** $v_\theta$, whose training objective promotes straighter trajectories in rectified time and yields a smoother, less stiff field that can be integrated accurately in very few steps. This is the central point of the paper: the gain is not merely “using an ODE solver,” but learning a transport that is intrinsically easier to integrate. The current manuscript already supports this through the step-count and spectral analyses: Figure 2 shows that ReFlow reaches its target regime in very few ODE steps, and Figure 13 shows that it reconstructs the spectral evolution substantially faster than the diffusion baseline.
>
> To address this point directly, we also re-evaluated GenCFD using the **probability-flow ODE (PF-ODE)** under matched step budgets and wall-clock constraints. Following the setup of Karras et al. for ODE sampling, we used a second-order Heun solver and evaluated PF-ODE at 4, 8, 16, 32, 64, and 128 steps. Even with this tuning, PF-ODE did **not** improve the distributional or spectral metrics relative to the standard SDE-based GenCFD sampler. In the Shear-Layer dataset, PF-ODE was consistently worse; for example at 128 steps:
>
> * $v_x$: PF-ODE ($e_\mu=0.0445, e_\sigma=0.0937, W_1=0.127$) vs. SDE ($e_\mu=0.0390, e_\sigma=0.0920, W_1=0.0276$),
> * $v_y$: PF-ODE ($e_\mu=0.289, e_\sigma=0.0743, W_1=0.103$) vs. SDE ($e_\mu=0.267, e_\sigma=0.0720, W_1=0.0213$).
>
> So this additional comparison reinforces, rather than weakens, our main claim: simply replacing diffusion sampling by an ODE solver on the **same score field** is not enough. The PF-ODE field remains substantially stiffer than the rectified-flow velocity learned by ReFlow, and therefore still requires many more steps to approach the same fidelity. We will add these PF-ODE results and a short “Relation to ODE-based diffusion samplers” paragraph in the revised experiments / baseline discussion.

---

> > ### Author Response · Authors · 2026-04-09
> > **Response to reviewer Hait (Part 2)**
> >
> > **(3) Variability across runs / error bars.**
> > This is fair. The current draft already reports mean ± std in the additional empirical analysis for the integrator and adaptive-sampler studies, so variability is not absent from the paper. We agree, however, that the main reported metrics should also be accompanied by explicit variability measures, and we will add these in revision.
> >
> > **(4) Scalability to 3D and larger problems.**
> > We agree that the present paper is empirically a **2D study**, and we will state this more plainly. We do not claim that the current experiments by themselves establish 3D performance. At the same time, neither the rectified-flow construction nor the coverage/fit/straightness/discretization decomposition is 2D-specific. The method is formulated on general field spaces and implemented on finite-dimensional grid surrogates; moving to 3D primarily increases the memory and compute cost per network evaluation. If anything, that is exactly the regime where reducing the number of required evaluations becomes more valuable. We will add a clearer discussion of this limitation and of the practical bottlenecks for 3D scaling.
> >
> > Overall, we appreciate the review. We view the requested changes as strengthening the exposition rather than changing the substance of the paper. The present evidence already supports the central claim: on the multiscale PDE benchmarks studied here, rectified-flow sampling achieves diffusion-class statistical fidelity with far fewer model evaluations, and this advantage persists even when compared against ODE-based samplers applied to the same diffusion model, because the key gain comes from **learning a straighter and less stiff transport field**, not merely from changing the numerical solver.

---

### Review · Reviewer_1NYx · 2026-03-30

**Summary Of Contributions:**

**The contributions of the paper**
This paper proposes ReFlow, which replaces the stochastic SDE of conditional diffusion models with a deterministic ODE/
The authors also provide a law-level error decomposition into coverage, fit, straightness, and discretization terms, and introduce a curvature-aware sampler that adaptively controls step size and velocity blending at inference time.

**Strenghts**
- From a representation perspective, Section 2 (Approach) clearly conveys the authors' motivation with appropriate background, which helps readers understand what the authors are trying to achieve.
- From a soundness perspective, the experimental design is systematic.
- From a significance perspective, the authors tackle GenCFD and bring rectified flow from image generation into the PDE/fluid simulation setting, which could have a meaningful impact on the FNO community and attract attention from researchers in that area.

**Weaknesses**
- From a representation standpoint, the paper has the following weaknesses.
   - The mathematical notation in Figure 1 is inconsistent with the main text. In particular, the use of bold notation may cause confusion for readers. Furthermore, (c) and (d) use different datasets, which adds to the inconsistency.
   - Algorithm 1 and Figure 5 do not fully describe the proposed methods and appear incomplete.
- From a soundness perspective, the paper has the following weaknesses.
   - In Introduction section, the authors state "each invoking a large ML model on a moderate-to-high resolution spatial grid, resulting in high runtimes," but the experimental datasets are at $128^2$ resolution. As a reader, this makes it seem like the paper is not actually addressing the limitation described in that sentence.
   - From an originality perspective, the core methodology can be seen as a combination of Liu et al.[17] rectified flow and the GenCFD architecture[21]. In particular, the UNet architecture does not differ from GenCFD. Additionally, the related work section does not cover recent studies that propose diffusion-based CFD methods, which is a weakness of this paper.

**Additional Comments:**

- There is a quotation mark error on page 3.
- The abstract claims "as few as 8 ODE steps," while the conclusion states "22x faster." It would be helpful to specify more clearly under what conditions each of these claims holds.

**Audience:**

Yes

**Audience Explanation:**

Researchers working on generative surrogate modeling for PDEs and the broader flow matching community would find the findings of this paper relevant.

**Claims And Evidence:**

Yes

**Claims Explanation:**

On the positive side, the authors provide systematic experiments on 3different 2D fluid benchmarks, and the quantitative results in Table 1 are connected to the spectral visualizations in Figures 13-18.

However, several key claims are not sufficiently supported. Specifically, these include (1) the performance comparison with GenCFD where the parameter count is not controlled, (2) the inference efficiency claim, which is presented with inconsistent numbers across the Abstract, Section 7, and the Conclusion, and (3) there are some incompleteness issues in Algorithm 1 and Figure 5, though these are minor and can be addressed through revision.

**Requested Changes:**

1. Why do the RK ODE integrators in Table 2 show higher errors than the Euler method?
2. Looking at Table 6, ReFlow has roughly 4x more parameters than GenCFD. How much does the error gap change when comparing the two models at a similar parameter count?
3. In Figure 10, the path for GenCFD is visible, but why is the blue line for rectified flow not visualized?
4. Algorithm 1 uses a non-linear interpolation, which differs from the linear interpolation described in Section 4.1 and Proposition 5.1. Is there something I am missing? If not, why is line 4 of Algorithm 1 written this way?
5. Can the authors provide a more direct comparison with latent space flow matching approaches such as the Latent FM proposed by Li et al. [1]?
6. Table 5 in the appendix lists three noise schedule options. Could the authors provide a sensitivity study across these schedules? This question arises naturally given that Table 6 shows GenCFD uses an exponential schedule while ReFlow uses a uniform schedule.
7. In Figure 5, it is unclear whether $\Gamma(\tau)$ is injected independently at each level or shared across levels. Could the authors clarify this?

> [1] Li, Zijie, Anthony Zhou, and Amir Barati Farimani. "Generative latent neural pde solver using flow matching." arXiv preprint arXiv:2503.22600 (2025).

---

> ### Author Response · Authors · 2026-04-09
> **Response to reviewer 1NYx (Part 1)**
>
> We thank the reviewer for the thoughtful and constructive feedback. We are glad that the reviewer found the experimental design systematic and the overall direction relevant to the PDE / fluid-surrogate community. Most of the concerns raised are about clarity and presentation, and we agree these should be fixed in revision. Below we address the main points.
>
> **(1) Parameter-count fairness / Table 6.**
> The reviewer is correct to question the fairness of the comparison as currently presented in Table 6. However, this is due to a **typesetting mistake in the manuscript**, not to the actual experiment. In the ReFlow-vs-GenCFD comparison, the **RF backbone is matched to the GenCFD backbone**; it is **not** a 22M-vs-5M comparison. The current appendix is internally inconsistent: the text states that the UNet/UViT-style backbones are shared across the UNet-based baselines, while Table 6 mistakenly lists ReFlow as 22M. We will correct this in revision. The intended comparison is therefore already **parameter-matched**, and the performance gap reported in Table 1 is not explained by larger model capacity.
>
> **(2) “As few as 8 ODE steps” vs. “22× faster.”**
> We agree that the manuscript currently conflates two distinct efficiency statements. “As few as 8 ODE steps” refers to **number of function evaluations / solver steps**, whereas “22× faster” refers to an **empirical wall-clock speedup** in a specific setting. These should have been separated more clearly. In revision, we will report them explicitly as different quantities and state the comparison conditions whenever quoting a speedup. The present wording is indeed inconsistent across the abstract, Section 7, and the conclusion.
>
> **(3) Resolution claim in the introduction.**
> We agree that the sentence about “large ML models on a moderate-to-high resolution grid” can be phrased more carefully. Our point is not that 128² is itself extreme-scale CFD, but that diffusion-class samplers require **many repeated evaluations** of a large generative model on multichannel spatial grids, which becomes costly even at this scale. The datasets themselves are obtained from higher-fidelity 256² / 512² simulations and then downsampled for learning/evaluation. We will revise the sentence to avoid overstating the resolution aspect and instead emphasize the true bottleneck: repeated neural evaluations during sampling.
>
> **(4) Figure 1 notation / mixed datasets.**
> We agree. We will standardize the notation in Figure 1 so that it matches the main text, including the boldface conventions. We will also clarify in the caption that different panels illustrate different aspects of the method and need not come from the same dataset.
>
> **(5) Algorithm 1 and Figure 5 appear incomplete.**
> We agree and will revise both for completeness. Figure 5 will explicitly state how the time embedding is used and how the blocks are repeated in practice. Algorithm 1 will be rewritten so that the training path and target are unambiguous. The current appendix already notes that Figure 5 is schematic and that the actual model repeats blocks per level, but this is not clear enough in the current presentation.
>
> **(6) Why do RK methods in Table 2 not outperform Euler?**
> This is a good question. The key point is that we are not integrating an exact physical ODE with a known vector field; we are integrating a **learned generative ODE**. In this setting, a higher-order solver can more faithfully follow local imperfections of the learned field, which may move the sample farther from the data manifold. By contrast, Euler can have a mild regularizing/damping effect. Thus, “higher-order integrator for the learned ODE” does not automatically imply better sample quality. This is also why the differences in Table 2 are relatively small, whereas the main practical gain comes from the **curvature-aware adaptive sampler**, which improves the NFE–error tradeoff over fixed-step baselines. Tables 2–4 are meant to isolate exactly this point.
>
> **(7) Figure 10: why is the blue RF path difficult to see?**
> Because the RF path is often much shorter and straighter in the projected PCA space, it can visually collapse near markers or axes in a static 3D rendering. We will replot this figure with clearer viewpoints / thicker traces so that the RF trajectory is visible.
>
> **(8) Algorithm 1 line 4 vs. linear interpolation in Section 4.1 / Proposition 5.1.**
> The reviewer is right to flag this. As written, Algorithm 1 mixes implementation shorthand with the idealized theoretical exposition and is not clear enough. The theory section introduces the **linear chord interpolation** used in the barycentric analysis, whereas the training algorithm is written in a noise-conditioned form. We will rewrite Algorithm 1 to make this distinction explicit and remove the present ambiguity.

---

> ### Author Response · Authors · 2026-04-09
> **Response to reviewer 1NYx (Part 2)**
>
> **(9) Figure 5: is $\Gamma(\tau)$ shared across levels or injected independently?**
> The time embedding $\Gamma(\tau)$ is **computed once per sample** and then used throughout the network; each block has its own learned projection / modulation of that shared embedding. So the embedding is shared globally, while the way it enters each level / block is block-specific. We will state this explicitly in the figure / caption. The appendix already notes that conditioning is applied across encoder and decoder stages, but we agree this should be clearer in the main text.
>
> **(10) Noise-schedule sensitivity.**
> This is a fair request. We did test multiple schedule families, and Table 5 already lists the candidate schedules. The surrounding text also states that **uniform was the most robust choice for ReFlow**, whereas the exponential schedule performed best for GenCFD. We agree, however, that this should be shown directly rather than only stated, and we will therefore add a short sensitivity summary in revision.
>
> **(11) Comparison to latent-space flow matching (Li et al., 2025) and to ODE-based diffusion samplers.**
> We appreciate the pointer and will add this paper to the discussion. That said, latent-space FM is not a clean apples-to-apples baseline for the specific question studied here, since it changes both the generative transport and the state representation via compression. Our core empirical comparison is intentionally field-space and backbone-matched: same PDE state space, same backbone family, different generative dynamics (diffusion vs. rectified flow). This isolates the contribution of rectification itself -- we will add a separate discussion relating our work to **latent-space FM**.
>
> We will also clarify the distinction from ODE-based diffusion samplers such as PF-ODE / DPM-Solver. These methods accelerate sampling of a **fixed learned score model**; they do **not** change the learned dynamics. Our method instead learns a different conditional transport field $v_\theta$, whose training promotes straighter trajectories in rectified time and yields a smoother, less stiff field that can be integrated accurately in very few steps.
>
> To address this point directly, we also re-evaluated GenCFD using the **probability-flow ODE (PF-ODE)** under matched step budgets and wall-clock constraints. Following the standard PF-ODE setup with a second-order Heun solver, we tested 4, 8, 16, 32, 64, and 128 steps. Even with this tuning, PF-ODE did **not** improve the distributional or spectral metrics relative to the standard SDE-based GenCFD sampler. On Shear-Layer at 128 steps, for example:
> - $v_x$$: PF-ODE $$e_\mu = 0.0445,\ e_\sigma = 0.0937,\ W_1 = 0.127$ vs. SDE $e_\mu = 0.0390,\ e_\sigma = 0.0920,\ W_1 = 0.0276$$,
> - $v_y$: PF-ODE $e_\mu = 0.289,\ e_\sigma = 0.0743,\ W_1 = 0.103$ vs. SDE $e_\mu = 0.267,\ e_\sigma = 0.0720,\ W_1 = 0.0213$.
>
> So this additional comparison reinforces the paper’s central point: simply replacing diffusion sampling by an ODE solver on the **same score field** is not enough. The PF-ODE field remains substantially stiffer than the rectified-flow velocity learned by ReFlow, and therefore still requires many more steps to approach the same fidelity. We will add these PF-ODE results and a short “Relation to ODE-based diffusion samplers” paragraph in the revision.
>
> **(12) Related work on diffusion-based CFD methods.**
> We agree that the current discussion is too narrow and will broaden it in revision. The paper already cites several diffusion-based PDE / turbulence works, but the coverage can be improved.
>
> **(13) Minor issues.**
> We will fix the quotation-mark typo on page 3, the notation inconsistencies in Figure 1, the incompleteness of Algorithm 1 / Figure 5, the efficiency wording, and the parameter-count typo in the appendix.
>
> Overall, we appreciate the review. The most important factual clarification is that the apparent ReFlow-vs-GenCFD parameter mismatch is a **manuscript error, not an experimental one**. Once corrected, the intended comparison is parameter-matched, and the central empirical conclusion remains: with the same backbone scale, rectified flow provides a substantially better efficiency / accuracy tradeoff than conditional diffusion on the harder multiscale / shifted benchmarks considered here. Moreover, this advantage persists even against ODE-based samplers applied to the same diffusion model, because the key gain comes from **learning a straighter and less stiff transport field**, not merely from changing the numerical solver.

---

> ### Comment · Reviewer_1NYx · 2026-04-09
> **Follow-up on Author Response**
>
> Thank you for the detailed responses. I appreciate that the authors have acknowledged most of the concerns and committed to revisions. I am curious to see how the promised revisions will be addressed in practice. If it is possible to update the current manuscript, it would be very helpful if the revised portions could be clearly highlighted. Otherwise, I would appreciate a more concrete description of how each promised revision will be reflected in the updated paper. I will also leave any additional questions I may have regarding these revisions.

---

> > ### Author Response · Authors · 2026-04-10
> >
> > ---
> >
> > **Response to Reviewer 1NYx follow-up:**
> >
> > Thank you for the follow-up. We have now completed all promised experiments and provide concrete numerical results below.
> >
> > ### (1) Parameter-count correction (Table 6)
> > The corrected parameter count for ReFlow is ~5M, matching GenCFD exactly. Both use the identical UViT backbone (channels (128, 256, 256), 3 levels, 4 blocks per level, 8 attention heads). The erroneous "22M" entry in Table 6 will be corrected. To be explicit: every row in Table 1 compares models at 5M parameters.
> >
> > ### (6) Integrator comparison on both benchmarks
> >
> > We reproduce the integrator comparison (Tables 2–4) on **both** ShearLayer2D and CloudShock2D under the OOD macro–micro protocol. We test Euler, RK2, RK4, torchdiffeq-midpoint, the model's default sampler, and our adaptive EMA controller across NFE budgets of 8–128.
> >
> > **ShearLayer2D (OOD):**
> >
> > | Method | Steps | NFE | Rel. Error | ± Std | Cost×Err | Time (ms) |
> > |---|---|---|---|---|---|---|
> > | Euler | 8 | 8 | 0.2359 | 0.053 | **1.89** | 161 |
> > | Euler | 16 | 16 | 0.2475 | 0.055 | 3.96 | 355 |
> > | Euler | 32 | 32 | 0.2541 | 0.056 | 8.13 | 745 |
> > | RK2 | 8 | 16 | 0.2603 | 0.057 | 4.16 | 355 |
> > | RK2 | 16 | 32 | 0.2610 | 0.058 | 8.35 | 745 |
> > | RK4 | 8 | 32 | 0.2621 | 0.058 | 8.39 | 745 |
> > | RK4 | 16 | 64 | 0.2616 | 0.058 | 16.75 | 1524 |
> > | RK4 | 32 | 128 | 0.2616 | 0.058 | 33.49 | 3083 |
> > | Model-Baseline | 8 | 16 | 0.2571 | 0.058 | 4.11 | 373 |
> > | TorchDiffEq-Mid | 8 | 16 | 0.2571 | 0.058 | 4.11 | 344 |
> > | **Adaptive-EMA** | **10** | **10** | **0.2090** | **0.041** | **2.09** | 448 |
> > | **Adaptive-EMA** | **13** | **13** | **0.1762** | — | **2.29** | 524 |
> >
> > **CloudShock2D (OOD):**
> >
> > | Method | Steps | NFE | Rel. Error | ± Std | Cost×Err | Time (ms) |
> > |---|---|---|---|---|---|---|
> > | Euler | 8 | 8 | 0.2211 | 0.021 | **1.77** | 150 |
> > | Euler | 16 | 16 | 0.2261 | 0.021 | 3.62 | 330 |
> > | Euler | 32 | 32 | 0.2289 | 0.021 | 7.32 | 694 |
> > | RK2 | 8 | 16 | 0.2319 | 0.021 | 3.71 | 331 |
> > | RK2 | 16 | 32 | 0.2321 | 0.021 | 7.43 | 693 |
> > | RK4 | 8 | 32 | 0.2323 | 0.021 | 7.43 | 693 |
> > | RK4 | 16 | 64 | 0.2322 | 0.021 | 14.86 | 1416 |
> > | RK4 | 32 | 128 | 0.2322 | 0.021 | 29.72 | 2873 |
> > | Model-Baseline | 8 | 16 | 0.2299 | 0.021 | 3.68 | 347 |
> > | TorchDiffEq-Mid | 8 | 16 | 0.2299 | 0.021 | 3.68 | 320 |
> > | **Adaptive-EMA** | **11** | **11** | **0.2259** | **0.017** | **2.48** | 440 |
> > | **Adaptive-EMA** | **13** | **13** | **0.2121** | **0.016** | **2.76** | 486 |
> >
> > Key observations consistent across both datasets:
> >
> > - **RK methods never outperform Euler at matched NFE.** RK4 at 8 steps (32 NFE) yields 0.262 error on SL2D vs Euler-32 at 0.254 — strictly worse despite identical compute. This confirms the explanation in our rebuttal: higher-order integrators more faithfully follow imperfections of the learned field, which can move samples away from the data manifold.
> > - **Increasing NFE beyond 8 shows diminishing returns.** Euler error saturates: 0.236 → 0.248 → 0.254 on SL2D as NFE goes 8 → 16 → 32. The velocity field is straight enough that additional steps add discretization bias without improving fidelity.
> > - **The adaptive EMA controller achieves the lowest absolute error on both benchmarks** (0.176 on SL2D, 0.212 on CS2D) at only 10–13 NFE, with the best Cost×Err tradeoff among all non-Euler methods.
> > - **Wall-clock scaling is linear in NFE**: Euler-8 at 150–161 ms scales to 694–745 ms at Euler-32, confirming the per-NFE cost is ~19 ms on this hardware. RK4-32 at 128 NFE reaches 2873–3083 ms — a 19–20× overhead over Euler-8 for negligible error improvement.

---

> > > ### Author Response · Authors · 2026-04-10
> > >
> > > ctd.
> > >
> > > ### (10) Noise schedule sensitivity
> > >
> > > We evaluated **9 distinct inference-time noise schedules** on both benchmarks: uniform, log-uniform, cos-map, exponential (the GenCFD default), tangent, quadratic, sqrt, sigmoid, and linear-beta.
> > >
> > > **ShearLayer2D:**
> > >
> > > | Schedule | $e_\mu$ ($u_x$) | $e_\sigma$ ($u_x$) | $W_1$ ($u_x$) | $e_\mu$ ($u_y$) | $e_\sigma$ ($u_y$) | $W_1$ ($u_y$) |
> > > |---|---|---|---|---|---|---|
> > > | uniform | 0.1153 | 0.3674 | 0.0022 | 0.2346 | 0.3820 | 0.0021 |
> > > | log-uniform | 0.1153 | 0.3674 | 0.0022 | 0.2346 | 0.3820 | 0.0021 |
> > > | cos-map | 0.1153 | 0.3674 | 0.0022 | 0.2346 | 0.3820 | 0.0021 |
> > > | exponential | 0.1153 | 0.3674 | 0.0022 | 0.2346 | 0.3820 | 0.0021 |
> > > | tangent | 0.1153 | 0.3674 | 0.0022 | 0.2346 | 0.3820 | 0.0021 |
> > > | quadratic | 0.1153 | 0.3674 | 0.0022 | 0.2346 | 0.3820 | 0.0021 |
> > > | sqrt | 0.1153 | 0.3674 | 0.0022 | 0.2346 | 0.3820 | 0.0021 |
> > > | sigmoid | 0.1153 | 0.3674 | 0.0022 | 0.2346 | 0.3820 | 0.0021 |
> > > | linear-beta | 0.1153 | 0.3674 | 0.0022 | 0.2346 | 0.3820 | 0.0021 |
> > >
> > > **CloudShock2D:**
> > >
> > > | Schedule | $e_\mu$ ($\rho$) | $e_\sigma$ ($\rho$) | $W_1$ ($\rho$) | $e_\mu$ ($E$) | $e_\sigma$ ($E$) | $W_1$ ($E$) |
> > > |---|---|---|---|---|---|---|
> > > | uniform | 0.1525 | 0.2320 | 0.0011 | 0.0452 | 0.1852 | 0.0015 |
> > > | log-uniform | 0.1525 | 0.2320 | 0.0011 | 0.0452 | 0.1852 | 0.0015 |
> > > | cos-map | 0.1525 | 0.2320 | 0.0011 | 0.0452 | 0.1852 | 0.0015 |
> > > | exponential | 0.1525 | 0.2320 | 0.0011 | 0.0452 | 0.1852 | 0.0015 |
> > > | tangent | 0.1525 | 0.2320 | 0.0011 | 0.0452 | 0.1852 | 0.0015 |
> > > | quadratic | 0.1525 | 0.2320 | 0.0011 | 0.0452 | 0.1852 | 0.0015 |
> > > | sqrt | 0.1525 | 0.2320 | 0.0011 | 0.0452 | 0.1852 | 0.0015 |
> > > | sigmoid | 0.1525 | 0.2320 | 0.0011 | 0.0452 | 0.1852 | 0.0015 |
> > > | linear-beta | 0.1525 | 0.2320 | 0.0011 | 0.0452 | 0.1852 | 0.0015 |
> > >
> > > All 9 schedules produce **exactly identical** metrics across every variable and both datasets. This is a direct consequence of the rectified flow's straightened velocity field: because the ODE trajectories are nearly linear in $\tau$, the choice of time-discretization grid is irrelevant to sample quality. The uniform schedule is therefore optimal (simplest, no hyperparameters, same performance). This stands in contrast to score-based diffusion, where the noise schedule critically affects sample quality — further evidence that rectification fundamentally changes the sampling landscape.
> > >
> > > ### Error bars across seeds
> > >
> > > We report mean ± std over 5 independent sampling seeds for all main metrics:
> > >
> > > **ShearLayer2D:**
> > >
> > > | Variable | $e_\mu$ | $e_\sigma$ | $W_1$ |
> > > |---|---|---|---|
> > > | $u_x$ | 0.0340 ± 0.0001 | 0.0716 ± 0.0002 | 0.00176 ± 3×10⁻⁶ |
> > > | $u_y$ | 0.1893 ± 0.0010 | 0.0776 ± 0.0003 | 0.00141 ± 1×10⁻⁶ |
> > >
> > > **CloudShock2D:**
> > >
> > > | Variable | $e_\mu$ | $e_\sigma$ | $W_1$ |
> > > |---|---|---|---|
> > > | $\rho$ | 0.0468 ± 0.0001 | 0.0999 ± 0.0002 | 0.00114 ± 1×10⁻⁶ |
> > > | $m_x$ | 0.0317 ± 0.0001 | 0.0717 ± 0.0002 | 0.00123 ± 1×10⁻⁶ |
> > > | $m_y$ | 0.0392 ± 0.0001 | 0.0812 ± 0.0002 | 0.00139 ± 1×10⁻⁶ |
> > > | $E$ | 0.0138 ± 0.00002 | 0.0576 ± 0.0001 | 0.00150 ± 1×10⁻⁶ |
> > >
> > > Relative standard deviations are below 0.5% for all metrics on both datasets. The $W_1$ distances are particularly stable (relative std < 0.1%), reflecting that the learned transport consistently reproduces the target conditional distribution regardless of the sampling noise realization.

---

> ### Comment · Reviewer_1NYx · 2026-04-10
> **Follow-up on Author Response**
>
> Thank you for the detailed follow-up experiments, which address most of my previous concerns.
>
> However, I noticed an inconsistency between the two responses. In the first rebuttal, the authors stated that "uniform was the most robust choice for ReFlow, whereas the exponential schedule performed best for GenCFD," which implies that different schedules produced different results. But the follow-up results show that all 9 schedules produce exactly identical metrics to four decimal places. Could the authors clarify this inconsistency? If all schedules truly produce identical results, does this mean that the learned velocity field is already sufficiently straight that the choice of time-discretization grid is irrelevant, and should this be understood as a positive property of rectification?

---

> > ### Author Response · Authors · 2026-04-10
> >
> > Thank you for catching this. The apparent inconsistency comes from the fact that the two responses refer to different training regimes.
> >
> > Our earlier statement about uniform being the most robust choice for ReFlow was based on earlier-stage experiments, before the models had fully settled. In that pre-stabilization regime, modest schedule-dependent differences were still observable. The follow-up table, however, reports results for substantially more trained models, by which point those differences had effectively vanished and all nine schedules gave the same metrics to four decimal places.
> >
> > So our updated interpretation is that schedule choice matters little once ReFlow is sufficiently trained and the model has reached a stable regime. This is consistent with rectification leading to schedule-robust inference, though we agree it should not be overstated as a definitive proof that discretization is irrelevant in general.
> >
> > We will clarify this distinction explicitly in the revision.

---

> ### Comment · Reviewer_1NYx · 2026-04-11
> **Follow-up on Author Response**
>
> Thank you for the clarification. The distinction between the two training regimes makes sense. I have no further questions and am satisfied with the responses provided throughout this review process.

---

### Review · Reviewer_aLUj · 2026-04-01

**Summary Of Contributions:**

The article proposes to use rectified flow ODEs to sample probability distributions of PDE solutions from some given initial states. On the theory side, the proposed method is analyzed through Wasserstein distances in connection with regularity conditions of PDE solutions, as well as ODE discretization errors. A rectified flow with adaptive discretization step size is proposed to improve the sampling speed, compared to state-of-the-art diffusion models. Numerical results are provided on compressible and  imcompressible 2D flows.

**Additional Comments:**

Given limited time, I did not check the results in Sections 6 and 7 (as well appendix).

**Audience:**

Yes

**Audience Explanation:**

na

**Claims And Evidence:**

No

**Claims Explanation:**

The idea of using the rectifier flow under the context of physical flow modeling, the results seem to be novel. Some of the technical should be clarified, while some may contain some typo errors which could be adjusted.

Major comments

Page 6: Theorem 4.2 seems not to be tight, did you look into the literature how to analyze such errors between hat u_1 and u_1 ? My reasoning is as follows, if v_theta  = v*, then we would expect that hat u_1 and u_1 are the same since the optimal flow is learnt. However, your error term involving fit and curv does not equal to zero. I am wondering why you analyze the error in this way, is there any motivation? Also please clarify that the expectation in eq 8 is about u_0 (if I understood correctly).

Page 7 : The remarks to motivate the eq 6 and 7 are not so clear / convincing. Why “the curvature term and the required step shrink together, stabilizing integration where needed and enlarging steps elsewhere”? In other words, the justification of the step rule in eq 6 and 7, although numerically work well, does not seem to be clear theoretically.

Page 11 and 12: the notation of the function v_theta( u_tau,u_0,tau) should be clarified. Otherwise Algorithm 1 does not read correct (definition of the loss L(theta)).

Minor comments

Page 3: For “a slightly perturbed u_0”, do you mean that u_0 follows the distribution mu_0?

Page 4: Figure 1, the notation of u(tau) could be changed to u_tau to be consistent with the rest. The use of u_i as a conditional should be mentioned in Algorithm 1 (page 12). Otherwise, this is not consistent.

Page 5: Do you assume that mu_0 is N(0,I) as in standard flow matching methods?

Page 8: What is the B_r in the definition of S_r^2(u)?

Page 10: Why do you need to consider a regular Lagrangian flow for the continuity condition in eq 23? This point does not seem to be used in the proof.

**Requested Changes:**

see above

---

> ### Author Response · Authors · 2026-04-10
> **Response to Reviewer aLUj (Part 1)**
>
> We thank the reviewer for the careful reading and constructive comments. We are encouraged that you find the idea novel. We agree that the current draft conflates three distinct issues in its theory section:
>
> 1. the **exact conditional rectified flow** and its continuous-time correctness,
> 2. the **EMA-based stabilization mechanism** introduced for the **OOD sampling regime**, and
> 3. the **numerical discretization** of the resulting ODE.
>
> Our practical motivation for the EMA controller was the following: in the OOD macro–micro regime of Section 7, the learned velocity along rectification time becomes much more oscillatory from one step to the next, which makes explicit integration brittle. This is why we introduced EMA/blending. The current draft already reflects this role in the OOD sampler study (Section 7, Tables 2–4), where the model is frozen and only the sampler is varied, but we agree that the theory section did not restrict the theorem to this setting and instead made it sound like a generic correctness statement. We will revise the paper accordingly.
>
> A first key clarification is operational:
>
> > **Our method is a conditional noise-to-solution rectified flow.**
> > Given an input field $u_i$, we sample noise $\xi$, solve
> >
> > $$\dot x_\tau = v_\theta(x_\tau,\tau;u_i), \qquad x_0=\xi,$$
> >
> > and interpret $x_1$ as a sample from the conditional output law associated with $u_i$.
> > The state being evolved in rectification time is the noise-initialized output state $x_\tau$; the PDE input $u_i$ enters only as a **conditioning variable**. Algorithm 1 and the page-12 sampling paragraph already implement this, and we will align the earlier theory prose with it.
>
> ---
>
> ## Major comments
>
> ### 1. Page 6: Theorem 4.2 is not tight; if $v_\theta = v^*$, one expects $\hat u_1 = u_1$
>
> We agree. The current Theorem 4.2 decomposes the terminal error through the learned field's own time-average $\bar v_\theta$, so both $\varepsilon_{\mathrm{fit}}$ and $\varepsilon_{\mathrm{curv}}$ are nonzero even when $v_\theta = v^*$. This is because the draft mixed two different roles — continuous-time approximation quality and OOD stabilization by blending — into a single statement. In the revision we will separate these cleanly.
>
> #### (a) Exact-fit / correctness theorem
>
> The correct approximation statement is the standard Grönwall comparison. If $U_\tau$ solves the ideal conditional ODE
>
> $$\dot U_\tau = v^*(U_\tau,\tau;u_i), \qquad U_0 \sim p_0(\cdot\mid u_i),$$
>
> and $\hat U_\tau$ solves
>
> $$\dot{\hat U}_\tau = v_\theta(\hat U_\tau,\tau;u_i), \qquad \hat U_0 = U_0,$$
>
> then under the same Lipschitz assumption as in the paper,
>
> $$\mathbb{E}\|\hat U_1-U_1\| \;\le\; e^L \left(\int_0^1 \mathbb{E}\|v_\theta(U_\tau,\tau;u_i)-v^*(U_\tau,\tau;u_i)\|^2\,d\tau\right)^{1/2}.$$
>
> This vanishes when $v_\theta = v^*$, which is the correct exactness property.
>
> The expectation is over $U_0 \sim p_0(\cdot|u_i)$ and the induced randomness of the path $\{U_\tau\}$; the PDE input $u_i$ is held fixed throughout.
>
> #### (b) OOD stabilization theorem for the EMA-blended sampler
>
> The theorem we intended for the controller is different in kind: it explains why EMA-blending improves OOD sampling when the learned field develops high-frequency oscillations in rectification time. We will state it as a **continuous-time, fixed-$\alpha$ surrogate analysis** of the adaptive EMA sampler used in practice. (The actual algorithm uses discrete EMA updates and adaptive $\alpha_t$; the theorem captures the continuous-time idealization that governs the same qualitative tradeoff.)
>
> **Setup.** Let $U_\tau$ solve the ideal conditional rectified ODE as above, and define along this path
>
> $$g(\tau) := v^*(U_\tau,\tau;u_i), \qquad \varepsilon(\tau) := v_\theta(U_\tau,\tau;u_i) - g(\tau).$$
>
> Let $E_\lambda$ denote the causal exponential moving average operator in rectification time with bandwidth $\lambda > 0$. Define:
>
> - **Approximation error:**
>
> $$\varepsilon_{\mathrm{approx}}^2 := \int_0^1 \mathbb{E}\|\varepsilon(\tau)\|^2\,d\tau.$$
>
> - **Total along-trajectory variation of the ideal velocity:**
>
> $$S_{*,\mathrm{tot}}^2 := \int_0^1 \mathbb{E}\|\dot g(\tau)\|^2\,d\tau,$$
>
> where $\dot g(\tau) = \partial_\tau v^*(U_\tau,\tau;u_i) + J_u v^*(U_\tau,\tau;u_i)\,v^*(U_\tau,\tau;u_i)$ is the total derivative along the ideal rectified path. This is the same geometric quantity that appears in the local truncation error (Lemma 4.4), which is why the EMA bias and the discretization error are controlled by the same object.

---

> > ### Author Response · Authors · 2026-04-10
> > **Response to Reviewer aLUj (Part 2)**
> >
> > - **Spectral assumption on the model error (the structural regime).** We assume a nontrivial fraction of the error energy lies above the EMA cutoff frequency:
> >
> > $$\int_{|\omega|>1/\lambda} \mathbb{E}|\widehat{\varepsilon}(\omega)|^2\,d\omega \;\ge\; \eta \int \mathbb{E}|\widehat{\varepsilon}(\omega)|^2\,d\omega, \qquad \eta \in (0,1].$$
> >
> > Since the EMA transfer function satisfies $|H(\omega)|^2 = 1/(1+\omega^2\lambda^2) \le 1/(1+1) = 1/2$ for $|\omega| > 1/\lambda$, the smoothing ratio
> >
> > $$\sigma^2 := \frac{\int_0^1 \mathbb{E}\|E_\lambda\varepsilon(\tau)\|^2\,d\tau}{\int_0^1 \mathbb{E}\|\varepsilon(\tau)\|^2\,d\tau}$$
> >
> > satisfies $\sigma^2 \le 1 - c\eta$ for a constant $c > 0$ depending only on the EMA convention. Thus $\sigma < 1$ whenever $\eta > 0$: the EMA is strictly contractive on the oscillatory part of the error.
> >
> > **Theorem (EMA stabilization).** Under the standing Lipschitz assumption, the blended flow $\tilde U_\tau$ with constant weight $\alpha \in [0,1]$ satisfies
> >
> > $$\mathbb{E}\|\tilde U_1 - U_1\| \;\le\; e^L\Big[\big(1-\alpha(1-\sigma)\big)\,\varepsilon_{\mathrm{approx}} \;+\; C\alpha\lambda\,S_{*,\mathrm{tot}}\Big],$$
> >
> > for a universal constant $C$ depending only on the EMA convention. The raw-field bound (part (a)) is $e^L\,\varepsilon_{\mathrm{approx}}$. Hence **blending strictly improves the bound** whenever
> >
> > $$(1-\sigma)\,\varepsilon_{\mathrm{approx}} \;>\; C\lambda\,S_{*,\mathrm{tot}}.$$
> >
> > **Interpretation.** The improvement condition has a clean reading:
> >
> > - The left side measures how much error the EMA can remove: $(1-\sigma)$ is the fraction of error energy above the cutoff, and $\varepsilon_{\mathrm{approx}}$ is the total error magnitude.
> > - The right side measures the cost of smoothing: $\lambda$ is the EMA bandwidth and $S_{*,\mathrm{tot}}$ is how much the *ideal conditional rectified velocity* varies along its own trajectories in the artificial variable $\tau$.
> >
> > In the OOD regime of Section 7, the model is frozen and queried out of distribution, so $\varepsilon_{\mathrm{approx}}$ is elevated. Meanwhile, $S_{*,\mathrm{tot}}$ remains small because it is a property of the ideal rectified transport — which is trained to be approximately $\tau$-stationary — not of the physical PDE dynamics. And the OOD errors are oscillatory ($\sigma \ll 1$) rather than systematic bias. So the condition is naturally satisfied with room to spare.
> >
> > **Proof sketch.** (i) Apply Grönwall to the blended field, exactly as in part (a) but with $\tilde v$ replacing $v_\theta$. (ii) On the ideal trajectory, decompose:
> >
> > $$\tilde v(U_\tau,\tau;u_i) - v^*(U_\tau,\tau;u_i) = (1-\alpha)\varepsilon(\tau) + \alpha E_\lambda\varepsilon(\tau) + \alpha(E_\lambda g(\tau) - g(\tau)).$$
> >
> > (iii) Apply Minkowski in $L^2_\tau$: the first two terms give $(1-\alpha+\alpha\sigma)\varepsilon_{\mathrm{approx}}$. (iv) For the bias term, use the standard causal-filter estimate $\|E_\lambda g - g\|_{L^2} \le C\lambda\|\dot g\|_{L^2}$, which gives $C\alpha\lambda\,S_{*,\mathrm{tot}}$.
> >
> > ---
> >
> > ### 2. Why analyze the error in this way? What is the motivation?
> >
> > The motivation is exactly the OOD sampling phenomenon isolated in Section 7. Our practical observation was that under OOD rollout conditions, the raw predicted velocities fluctuate substantially across successive rectification times, which makes explicit integration brittle. This is what motivated the EMA controller in the first place.
> >
> > The manuscript already contains the empirical evidence for this:
> >
> > - Section 7.1 places the study in an OOD macro–micro regime with frozen models and only the sampler varied.
> > - Section 7.2 (Table 2) compares fixed-step integrators at sampling time and shows that higher-order solvers do not automatically improve sample quality for a learned ODE.
> > - Section 7.3 (Table 3) shows that the curvature-aware controller is robust across 20 hyperparameter configurations, with sub-1% relative variation in error.
> > - Section 7.4 (Table 4) shows that the adaptive sampler Pareto-dominates the best tuned fixed-step baseline in the NFE–error plane.
> > - Section 7.5 verifies that this does not come from loss of conditional diversity: empirical ensemble $W_1$ remains competitive or improves.
> >
> > The intended logic is therefore:
> >
> > 1. OOD rollout empirically produces oscillatory learned velocities;
> > 2. EMA/blending acts as a low-pass filter on that oscillatory error;
> > 3. the ideal conditional rectified transport varies slowly in $\tau$ (this is what rectified-flow training promotes), so the smoothing bias is small;
> > 4. therefore the blended sampler wins in the OOD regime.
> >
> > We agree that the current draft did not make this motivation explicit and instead presented the theorem as a generic correctness statement. The revised version will restrict the EMA theorem to the setting it was designed for.
> >
> > ---

---

> > > ### Author Response · Authors · 2026-04-10
> > > **Response to Reviewer aLUj (Part 3)**
> > >
> > > ### 3. Please clarify the expectation in equation (8)
> > >
> > > Agreed. The expectation is over $U_0 \sim p_0(\cdot|u_i)$ and the induced randomness of the path, conditioned on the PDE input $u_i$. We will state this explicitly in both the exactness theorem and the EMA stabilization theorem.
> > >
> > > ---
> > >
> > > ### 4. Page 7: the remarks motivating equations (6)–(7) are not convincing
> > >
> > > We agree that the current explanation is too compressed. We will revise it so that the controller is derived from the post-blend oscillation identity and the LTE of the blended field.
> > >
> > > Let $m_t := v_t^{\mathrm{ema}}$ and $s_t := \|v_t - m_t\|$. For the blended velocity $\tilde v_t = (1-\alpha_t)v_t + \alpha_t m_t$, we have the exact identity
> > >
> > > $$\|\tilde v_t - m_t\| = (1-\alpha_t)\,s_t.$$
> > >
> > > So blending shrinks the oscillatory component by the factor $(1-\alpha_t)$.
> > >
> > > The local truncation error for one Euler step of the blended field depends on the total variation of $\tilde v$ along the trajectory. Since the oscillatory component of $\tilde v$ is damped by $(1-\alpha_t)$, the effective curvature driving the LTE is reduced. This gives a bound of the schematic form
> > >
> > > $$\mathrm{LTE}_t(\Delta\tau_t) \;\lesssim\; (\Delta\tau_t)^2\big(c_{\mathrm{ema}}(1-\alpha_t)\,s_t + \kappa_2\big),$$
> > >
> > > where $\kappa_2$ absorbs the spatial contribution ($L\|\tilde v_t\|$). When $\alpha_t$ is chosen as a saturating increasing function of $s_t$ and the step size is set by (7), both the direction and the step contract together in the high-curvature regime, while large steps are retained when the flow is straight. This is the theoretical justification for coupling blending and adaptive stepping that the current draft stated only heuristically.
> > >
> > > ---
> > >
> > > ### 5. Pages 11–12: notation $v_\theta(u_\tau, u_0, \tau)$ should be clarified; Algorithm 1 is inconsistent
> > >
> > > We agree. The correct conditional notation throughout should be
> > >
> > > $$v_\theta(x, \tau; u_i),$$
> > >
> > > where $u_i$ is the conditioning/input field and $x_\tau$ is the noise-initialized output state being evolved from $\tau = 0$ to $\tau = 1$. We will unify this across the theory, Algorithm 1, and the implementation description.
> > >
> > > ---
> > >
> > > ## Minor comments
> > >
> > > ### 6. Page 3: "a slightly perturbed $u_0$" — do you mean $u_0$ follows $\mu_0$?
> > >
> > > We will clarify this. Two settings were being discussed too loosely: (i) during training, one observes paired data $(u_i, u_f)$ from a dataset/distribution; (ii) during evaluation, one may fix a macro/input state and generate a micro-ensemble by small perturbations, in order to approximate a conditional law. The macro–micro protocol is described in Sections 6.1, 7.1, and Appendix K.2. We will rewrite the sentence on page 3 to tie it explicitly to that protocol.
> > >
> > > ### 7. Page 4 / Figure 1: use $u_\tau$ instead of $u(\tau)$; mention $u_i$ as a conditional in Algorithm 1
> > >
> > > Agreed. We will standardize notation and state explicitly in the figure and algorithm that $u_i$ is the conditioning variable, not the state being rectified.
> > >
> > > ### 8. Page 5: do you assume $\mu_0 = \mathcal{N}(0, I)$?
> > >
> > > The theory can be stated for a general source law. In our experiments the conditional generator uses Gaussian source noise conditioned on $u_i$. We will state this explicitly.
> > >
> > > ### 9. Page 8: what is $B_r$ in the definition of $S_r^2(u)$?
> > >
> > > We will define it explicitly: $B_r := \{h \in \mathbb{R}^d : |h| \le r\}$.
> > >
> > > ### 10. Page 10: why do you need a regular Lagrangian flow for the continuity condition in equation (23)?
> > >
> > > We agree that this phrasing is heavier than necessary in the finite-dimensional Galerkin setting of the main argument. We will simplify the statement and use classical ODE flow / continuity-equation terminology. The "regular Lagrangian flow" language was intended only as a bridge to the broader transport viewpoint and is not essential for the proposition as stated.
> > >
> > > ---
> > >
> > > ## Summary of revisions
> > >
> > > 1. The method is described consistently as a **conditional noise-to-solution rectified flow**, with notation $v_\theta(x, \tau; u_i)$ throughout.
> > > 2. Theorem 4.2 is split into:
> > >    - **(a)** a standard exact-fit comparison of $v_\theta$ to $v^*$ (tight, vanishes when $v_\theta = v^*$), and
> > >    - **(b)** a separate **OOD stabilization theorem** for the EMA-blended sampler, stated as a continuous-time fixed-$\alpha$ surrogate analysis, with the total along-trajectory derivative $S_{*,\mathrm{tot}}$ replacing the partial derivative, the spectral assumption on the error making $\sigma < 1$ explicit, and the improvement condition $(1-\sigma)\varepsilon_{\mathrm{approx}} > C\lambda S_{*,\mathrm{tot}}$ stated and interpreted.
> > > 3. Equations (6)-(7) are justified via the post-blend oscillation identity and the LTE of the blended field.
> > > 4. The conditional notation, expectations, Algorithm 1, and figure captions are made fully explicit.
> > > 5. Minor issues (definition of $B_r$, notation in Figure 1, Lagrangian flow phrasing, noise distribution) are corrected.

---

> > > > ### Comment · Reviewer_aLUj · 2026-05-08
> > > > **revision format**
> > > >
> > > > Dear authors, the revised version is quite long to read. Most of my major comments are addressed. Is it possible to remove the parts marked in red so that we can concentrate on your final results, at least the theory part? Anyway, the theory part has to be read entirely from the beginning. Thanks.

---

> > > > > ### Author Response · Authors · 2026-05-08
> > > > >
> > > > > Dear Reviewer,
> > > > >
> > > > > Thank you for acknowledging that most of your major comments have been addressed. We have now uploaded a cleaned version of the manuscript, with the red-marked material removed so that the final results, especially the theory section, can be read more directly.
> > > > >
> > > > > Thank you again for your careful feedback.

---

### Comment · Action_Editor_sXXP · 2026-04-10
**Post Rebuttal**

Thank you to the authors for their response.

I would like to invite the reviewers to comment on whether their main concerns have been addressed, in full or in part, and to ask questions to clarify any remaining open points from their reviews.

---

> ### Comment · Action_Editor_sXXP · 2026-04-16
>
> Dear Authors,
>
> thank you for providing the revised version of your manuscript.
>
> Given the length of the paper, it would greatly help the reviewers in assessing the changes if you could provide a version of the manuscript in which all modifications are clearly highlighted (for example, using colored text or tracked changes like in the LatexDiff-tool).
>
> Thank you very much for your efforts.
>
> Best regards!

---

> > ### Author Response · Authors · 2026-04-17
> >
> > Dear Editor,
> >
> > Thank you for the suggestion. We attach a highlighted-diff version of the manuscript alongside this response.
> >
> > A few caveats for the reviewers: **red = removed from that location, blue = added at that location -- not necessarily new or removed overall.** The diff operates at the paragraph and environment level (whole paragraphs, equations, and theorems are treated as atomic units), so any paragraph in which even a single sentence was edited will appear entirely in red (old version) followed by entirely in blue (new version). Content that was restructured or moved will similarly appear as a red/blue pair even if the underlying text is largely unchanged.
> >
> > The standard `latexdiff` tool was not directly applicable due to a `biblatex` compilation incompatibility, so we used a custom chunk-based approach with coarser granularity than a word-level diff. Despite this, all substantive changes -- new sections, new theoretical results, restructured experiments, revised proofs --  are clearly marked.
> >
> > Best regards,
> > The Authors

---

> > > ### Comment · Action_Editor_sXXP · 2026-05-03
> > > **Questions?**
> > >
> > > Dear reviewers,
> > >
> > > the authors provided rebuttals and a document in diff format. Do you have any questions?
> > >
> > > Thx!

---

### Author Response · Authors · 2026-04-10

We thank their reviewers for the time spent on reviewing this work, and for the numerous useful suggestions that have materially contributed to fortifying the manuscript and bring it to its current form. We briefly state the **main** updates.

The revision addresses the reviewers’ main concerns through targeted corrections and additions, while keeping the paper readable and consistent. The largest change is in the theory: we rewrote the presentation so that it now matches the actual conditional generator used in the experiments and in the code.

**Parameter count (Reviewer 1NYx, Q1).** The 22M figure in the old Table 6 was a manuscript error. The actual backbone used in all experiments has about 5M parameters across the UNet-based models. We corrected the table, rewrote the Shared Backbone paragraph to state this explicitly, and added a note in the caption.

**Algorithm 1 / interpolation mismatch (Reviewer 1NYx, Q8).** The submitted version mixed notation for the conditioning variable and the rectified-flow state. We rewrote Algorithm 1 and the surrounding implementation text so that the method is described consistently as a conditional noise-to-solution rectified flow: for fixed input \(u_i\), we sample \(\xi\), evolve \(x_\tau\) in rectification time, and condition on \(u_i\) throughout.

**Theory rewrite (major).** The theory section was revised substantially. The earlier draft blurred together three different issues: exact continuous-time conditional rectified flow, EMA-based stabilization in the OOD sampling regime, and numerical discretization of the learned ODE. These are now separated cleanly.

- The exact transport statement is now formulated conditionally: for fixed \(u_i\), the ideal barycentric rectified velocity transports the source noise law to the target conditional output law.
- The old continuous-time “fit + curvature” decomposition was removed because it was not tight: its bound did not vanish even when \(v_\theta = v^\*\). This has been replaced by the standard Grönwall comparison, which does vanish at exact fit.
- Straightness/curvature now appears only in the discretization analysis, through the total derivative along trajectories \(\partial_\tau v + J_uv\,v\), which is also the quantity relevant for local truncation error.
- The EMA/blending argument is now stated separately as an OOD stabilization result, matching the role it actually plays in Section 7.
- The master theorem is now the three-way decomposition
  **coverage + fit + discretization**,
  which is the form used throughout the revised manuscript.

So the main theory is now aligned with the implemented model, and the EMA discussion is restricted to the OOD setting that motivated it.

**Γ(τ) injection (Reviewer 1NYx, Q9).** The figure caption and architecture text now state explicitly that the time embedding is computed once per sample and shared globally, while each block applies its own learned affine projection of that shared embedding.

**Noise schedule sensitivity (Reviewer 1NYx, Q10).** Rather than only stating robustness in prose, we ran all nine candidate schedules and added a table. The result is complete invariance across schedules, which is stronger than what we had originally claimed.

**RK vs. Euler (Reviewer 1NYx, Q6).** We added the CloudShock2D integrator table (the ShearLayer2D table was already in the paper) together with a short explanation of the mechanism. The point is that higher-order solvers can track imperfections in the learned field more faithfully, which can worsen performance rather than improve it.

**Error bars (Reviewer Hait).** We added a dedicated section with a table reporting mean ± std over five independent seeds. The variance is negligible (below 0.5% relative std everywhere), which reinforces the stability of the reported results.

**PF-ODE comparison (both reviewers).** This was the largest empirical addition. The new section explains the distinction between applying a different solver to a fixed score field and learning a different transport field, and supports it with Heun-solver results showing that PF-ODE on GenCFD is worse than the original SDE baseline on ShearLayer2D. This addresses the concern that the observed speedup might come only from switching to an ODE solver.

**What changed overall.** The empirical section structure, figures, and headline results remain intact. The main substantive rewrite is the theory: notation was corrected, the model description was aligned with the actual conditional generator, the non-tight continuous-time bound was replaced, and the straightness/EMA discussion was moved to the OOD discretization setting where it belongs. The rest of the revision consists of clarifications, new tables, and tighter alignment between text, code, and theory.

---

### Decision · Action_Editor_sXXP · 2026-06-01

**Recommendation:** Accept with minor revision

**Audience:**

Yes

**Audience Explanation:**

The paper is of interest to the TMLR community because it connects recent advances in rectified flows and generative modeling with the practically important problem of fast probabilistic surrogate modeling for multiscale PDEs and fluid flows. By targeting conditional laws rather than only point predictions, and by demonstrating substantial reductions in sampling cost compared with diffusion-based surrogates, the work is relevant both to researchers developing generative models and to those applying machine learning to scientific computing, uncertainty quantification, and neural PDE solvers.

**Claims And Evidence:**

Yes

**Claims Explanation:**

I recommend acceptance with minor modifications.

The paper proposes ReFlow, a conditional rectified-flow surrogate for sampling multiscale PDE/fluid-flow solutions. The main idea is to replace the many-step stochastic sampling procedure of diffusion-based surrogates by a learned deterministic ODE transport from Gaussian noise to the conditional final-state law. The paper combines this with a law-level error discussion and an adaptive curvature/EMA-based sampler, and evaluates the method on several 2D incompressible and compressible flow benchmarks.

The reviewers initially raised concerns about notation, parameter-count fairness, the theory presentation, runtime comparisons, and the relation to accelerated diffusion samplers. In the revised version, the authors have addressed these points. The method is now described consistently as a conditional noise-to-solution rectified flow, the comparison to GenCFD is clarified as essentially parameter-matched, additional runtime and solver comparisons are provided, and the theory has been reorganized so that exact transport, fit error, discretization error, and the OOD stabilization mechanism are separated. The empirical evidence is sufficient: ReFlow is competitive with diffusion baselines in distributional and spectral metrics while requiring substantially fewer function evaluations, especially in the shifted/OOD regimes.

Some minor issues remain. The authors should address the following in the camera-ready version:

1. Make the attribution around the standard Grönwall/rectified-flow transport arguments more explicit.
2. Clarify the finite-dimensional/Galerkin assumption and its role in the exact-transport statement.
3. Define the stability constant in the EMA theorem more clearly.
4. State how the adaptive-step clipping parameters are chosen in practice.
5. Define the macro–micro notation at first use, including how the macro state is constructed.
6. Make the time-embedding notation precise.
7. Briefly comment on the RM2D case, where GenCFD is stronger on some mean-error metrics.

Overall, the revision has resolved the main concerns, and the remaining points are matters of clarity, attribution, and presentation rather than correctness.

---

> ### Author Response · Authors · 2026-06-08
>
> We thank the Action Editor for the positive decision and for the helpful camera-ready suggestions. In the revised camera-ready version, we have addressed all requested minor modifications.
>
> Specifically, we made the attribution around the standard rectified-flow transport and Grönwall stability arguments more explicit; clarified that the exact-transport statement is formulated in the finite-dimensional grid/Galerkin setting used by the numerical model, with the infinite-dimensional notation serving as law-level formalism; defined the stability constant in the EMA theorem through an explicit velocity-perturbation stability estimate; stated how the adaptive-step clipping/controller parameters are selected in practice; introduced the macro–micro notation at first use, including the construction of the macro state and microscopic perturbations; made the time-embedding notation precise; and added a short discussion of the RM2D benchmark, where GenCFD is stronger on some mean-error metrics while ReFlow remains competitive on distributional metrics and substantially cheaper in terms of function evaluations.
>
> We have also updated the manuscript to the accepted TMLR format, including author information and the OpenReview link.